# MINIMAX-OPTIMAL AGGREGATION FOR DENSITY RATIO ESTIMATION

**Lukas Gruber    Markus Holzleitner    Sepp Hochreiter    Werner Zellinger**
ELLIS Unit Linz and LIT AI Lab, Institute for Machine Learning
Johannes Kepler University
Linz, Austria
{gruber,zellinger}@ml.jku.at

## ABSTRACT

Density ratio estimation (DRE) is fundamental in machine learning and statistics, with applications in domain adaptation and two-sample testing. However, DRE methods are highly sensitive to hyperparameter selection, with suboptimal choices often resulting in poor convergence rates and empirical performance. To address this issue, we propose a novel model aggregation algorithm for DRE that trains multiple models with different hyperparameter settings and aggregates them. Our aggregation provably achieves minimax-optimal error convergence without requiring prior knowledge of the smoothness of the unknown density ratio. Our method surpasses cross-validation-based model selection and model averaging baselines for DRE on standard benchmarks for DRE and large-scale domain adaptation tasks, setting a new state of the art on image and text data.

## 1 INTRODUCTION

Let $\{x_m\}_{m=1}^M$ and $\{x'_n\}_{n=1}^N$, with $M, N \in \mathbb{N}$, be two sets of samples independently drawn from probability density functions $p$ and $q$, respectively. The goal of Density Ratio Estimation (DRE) is to learn the density ratio $\beta(x) := \frac{p(x)}{q(x)}$ from these samples. According to the fundamental lemma of mathematical statistics, a statistical test with maximal power has a unique representation as a threshold decision on the density ratio (Neyman & Pearson, 1933). It is therefore natural that DRE plays a central role in machine learning and statistics (Sugiyama et al., 2012a). DRE is used in a wide range of applications, including distribution shift detection (Shimodaira, 2000; Dinu et al., 2023; Della Vecchia et al., 2025), divergence estimation (Nguyen et al., 2007; 2010; Rhodes et al., 2020), anomaly detection (Smola et al., 2009; Hido et al., 2011), energy-based modeling (Ceylan & Gutmann, 2018; Gutmann & Hyvärinen, 2012; Tu, 2007; Grover & Ermon, 2018), and generative modeling (Mohamed & Lakshminarayanan, 2016; Grover et al., 2019; Kim et al., 2024; Sanokowski et al., 2025). Unlike supervised learning, DRE is performed without access to the true density ratio values $\beta(x)$ for the combined sample set $\{x_m\}_{m=1}^M \cup \{x'_n\}_{n=1}^N$.

However, the accuracy of density ratio estimators is highly sensitive to hyperparameter choices, which has already been observed in Kanamori et al. (2009a;b). For example, a series of works (Sugiyama et al., 2007; 2008; 2010) investigates kernel bandwidth selection for Kernel Mean Matching (KMM) (Huang et al., 2006), which cannot be done by straight-forward cross-validation (CV). In practice, one often ends up with a sequence of models trained with distinct hyperparameter settings, with the goal of selecting a model that achieves fast convergence rates.

In this work, we address the problem of optimally aggregating all models in such a sequence. In contrast to related approaches, e.g., cross-validation as used in (Kanamori et al., 2009a; Bickel et al., 2009; Menon & Ong, 2016; Nguyen et al., 2010; Rhodes et al., 2020; Choi et al., 2021; Srivastava et al., 2023; Zellinger, 2025; Kato & Teshima, 2021), aggregation can outperform the best model in such a sequence of models, see Figure 1.

Informally, our method trains one model $f_k$ for each hyperparameter setting, and subsequently optimizes the aggregation $\sum_k \alpha_k f_k$ s.t. minimax-optimal error rates are achieved. This is done by optimizing an upper bound on the Bregman divergence between the aggregated estimator and the

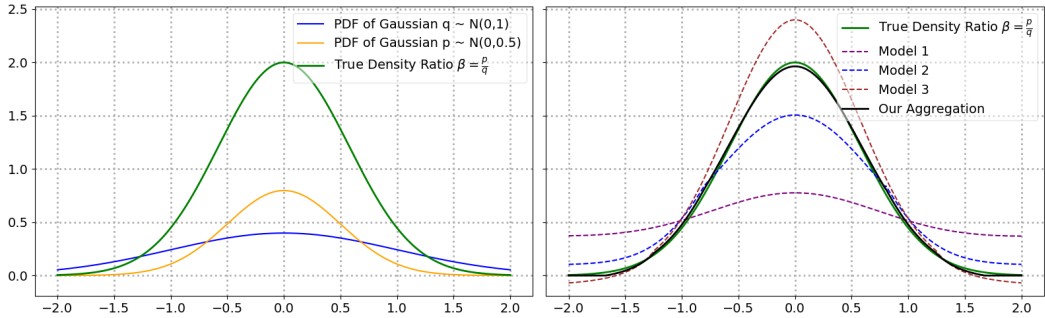

Figure 1: Left: Density ratio $\beta = \frac{p}{q}$ between Normal distribution $p = \mathcal{N}(0, 0.5)$ and $q = \mathcal{N}(0, 1)$. Right: Procedures for selecting a single Model 1–3 (dashed) cannot approximate $\beta$ well. In contrast, our aggregation method (solid) achieves a good approximation.

true density ratio $\beta$. More details in Eq. 5 that follows (Zellinger et al., 2023), combining (Menon & Ong, 2016) with (Marteau-Ferey et al., 2019; Bach, 2010). Our approach is summarized in Algorithm 1 that conveniently computes an analytic solution of this upper bound. This solution satisfies minimax-optimal convergence rates for optimization in reproducing kernel Hilbert spaces (RKHS), following the theory for self-concordant loss functions (Bach, 2010; Marteau-Ferey et al., 2019; Beugnot et al., 2021). Our approach can be applied to DRE methods which don't allow parameter tuning by CV (such as KMM), and is computationally more efficient than standard n-fold CV (see Ablation 2 in Table 1 and Figure 2 (Left)). We conduct experiments on DRE benchmarks, as well as on large-scale model selection benchmarks for deep domain adaptation (text, image, time-series), training over 9,000 neural networks and performing approximately 500 hyperparameter selection tasks. Results underpin the empirical effectiveness of our approach, which sets a new state of the art for model selection on MiniDomainNet and Amazon Reviews.

Our main contributions are:

- We propose a novel theory-grounded aggregation method for approaching hyperparameter choice issues in DRE.
- We prove minimax-optimal error convergence (for $M, N \to \infty$) for a broad class of DRE methods (Sugiyama & Kawanabe, 2012) when applied in combination with our method.
- We conduct experiments on benchmarks for DRE and large-scale domain adaptation tasks on diverse data modalities (text, image and sensory data) involving over 9000 neural networks and 500 hyperparameter selection tasks; validating outperformance of related approaches.
- Our method improves cross-validation in density ratio estimation in terms of (a) computational efficiency (see Figure 2 (Left), (b) the possibility of recovering density ratios not present in the given sequence of models (e.g., Figure 1) and (c) applicability (e.g., to parameter choice issues of KMM, Ablation 2 in Table 1).
- Our method achieves better performance than recent and advanced model averaging and ensembling baselines for DRE.

## 2 BACKGROUND

It was first observed by Sugiyama et al. (2012b) that a broad class of methods for DRE use the objective

$$f_{\mathcal{H}} := \arg\min_{f \in \mathcal{H}} B_F(\beta, g(f)) \qquad (1)$$

and derive the desired density ratio estimator by $\beta_{\mathcal{H}} := g(f_{\mathcal{H}})$, where $g : \mathbb{R} \to \mathbb{R}$ is a strictly increasing function (depending on the specific DRE method as in Example 1) and

$$B_F\left(\beta, \widehat{\beta}\right) := \mathbb{E}_{x \sim q}[F(\beta(x)) - F(\widehat{\beta}(x)) - F'(\widehat{\beta}(x))[\beta(x) - \widehat{\beta}(x)]]$$

denotes the Bregman divergence with prescribed generator $F : \mathbb{R} \to \mathbb{R}$. For our theoretical results, we assume $f \in \mathcal{H}$ to be in a RKHS, which is not required for our algorithm and our experiments.

For example, as can be seen in Table 1, experiments were conducted for both, methods in RKHS and neural networks in a given model class. Consider the following example (cf. Sugiyama et al. (2012b); Menon & Ong (2016); Kato & Teshima (2021); Zellinger et al. (2023); Kim et al. (2024); Zellinger (2025)):

**Example 1.**

1. *As shown in Kanamori et al. (2012b) the kernel unconstrained least squares importance fitting procedure (KuLSIF) (Kanamori et al., 2009a) is realized by using $F(x) = (x-1)^2/2$ and $g(x) = x$ in equation 1 such that $B_F(\beta, g(f)) = \frac{1}{2} \|\beta - g(f)\|_{L^2(q)}$.*

2. *Menon & Ong (2016) use $F(x) = x^{3/2}$ and $g(x) = e^{2x}$ in equation 1 to obtain the exponential function approach (Exp) as applied for AdaBoost (Freund et al., 1996).*

3. *The square loss approach (SQ) in Menon & Ong (2016) uses $F(x) = \frac{1}{2x+2}$ and $g(x) = \frac{-1+2x}{2-2x}$.*

4. *To realize the logistic regression (Nelder & Wedderburn, 1972) (LR) approach as used in Bickel et al. (2009) $F(x) = x\log(x) - (1+x)\log(1+x)$ and $g(x) = e^x$ are set in equation 1.*

## 3 RELATED WORK

This work is along the lines of research that investigates methods that achieve finite-sample error rates for approximations of equation 1. Such error rates were shown for KuLSIF in Kanamori et al. (2012b); Gizewski et al. (2022); Nguyen et al. (2023) and extended to a class of DRE methods with self-concordant loss functions by Zellinger et al. (2023); Gruber et al. (2024). Their work demonstrates that fast error rates can be achieved when the optimal hyperparameter configuration is employed in Eq. 1. However, to achieve fast error rates when estimating the finite-sample approximation in Eq. 1 it is crucial to have an appropriately chosen hyperparameter setting. DRE is a complex problem and different studies highlight its complexity. For example, problems such as density-chasm (Rhodes et al., 2020) arise when the underlying densities exhibit significant dissimilarity Srivastava et al. (2023); Zellinger (2025). To mitigate these issues, Choi et al. (2021) propose leveraging normalizing flows to achieve simpler and closer approximations of the target densities.

Many applications exist. For example, recent research has explored deep learning-based DRE methods in positive-unlabeled learning (Kiryo et al., 2017; Kato & Teshima, 2021). Another active area of research focuses on integrating DRE techniques into deep generative models. For example, Kim et al. (2023) utilize DRE to enhance diffusion-based generative models. Subsequently, Kim et al. (2024) extend the Bregman framework to incorporate time-dependent corrections and introduce novel Bregman divergences. DRE methods have also been applied to diverse tasks beyond generative modeling. These include conditional density estimation (Schuster et al., 2020), mutual information estimation (Poole et al., 2019; Belghazi et al., 2018; Song & Ermon, 2020), importance weighting (Sugiyama et al., 2012a; Fang et al., 2020; Zhang et al., 2023), model-based enrichment estimation (Busia & Listgarten, 2023) and unsupervised representation learning (Thomas et al., 2020).

Although aggregation methods for averaging and ensembling density ratios have been used (Zhu & Tan, 2013; Chandra et al., 2018; Rhodes et al., 2020; Fragoso et al., 2018; Hushchyn & Ustyuzhanin, 2021; Yu et al., 2021; Wu & Benkeser, 2024), none of these methods can guarantee fast convergence rates for a finite sequence of suboptimal models, even when the sample size $M+N$ grows to infinity, see Ablation 3 in Table 1 and Fig. 2 (Right) for practical consequences.

## 4 PROBLEM

**Notation** Following Menon & Ong (2016), equation 1 can be equivalently formulated as binary classification with loss function $\ell : \mathcal{Y} \times \mathbb{R} \to \mathbb{R}$ and $\mathcal{Y} := \{-1, 1\}$, such that $\ell(1, f(x))$ and $\ell(-1, f(x))$ measure the error of a classifier $f(x)$ predicting whether $x$ is originating from $p$ or $q$. For this, we only need the assumption that the combined sample

$$\mathbf{z} := \{(x_m, 1)\}_{m=1}^M \cup \{(x_n', -1)\}_{n=1}^N \tag{2}$$

is an i.i.d. sample from the distribution $\rho$ constructed as follows (Zellinger et al., 2023): it is defined as a probability measure on $\mathcal{X} \times \mathcal{Y}$ with conditionals $\rho(x|y=1) := p(x)$, $\rho(x|y=-1) := q(x)$, and, marginal $\rho_{\mathcal{Y}}$ defined as Bernoulli measure such that the probability for both events $y = 1, y =$

---

**Algorithm 1** DRE Aggregation

---

**Input:** $K$ different hyperparameter setups, dataset $\mathbf{z} = \{(x_m, 1)\}_{m=1}^M \cup \{(x'_n, -1)\}_{n=1}^N$, see (2) .

**Output:** Model $\widehat{\beta}(x) = g\left(\sum_{k=1}^K \widehat{\alpha}_k f_k(x)\right)$ with optimal coefficients $\widehat{\boldsymbol{\alpha}} = (\widehat{\alpha}_1, \ldots, \widehat{\alpha}_K) \in \mathbb{R}^K$.

**Step 1:** Train one model $\beta_k$ for each hyperparameter setup (e.g., using a method in Example 1) and denote by $f_k = g^{-1}(\beta_k)$ the associated binary classifier (see Example 2).

**Step 2:** Compute aggregation weights $\widehat{\boldsymbol{\alpha}} = \widehat{\mathbf{G}}^{-1}\widehat{\mathbf{r}}$ with empirical Gram matrix $\widehat{\mathbf{G}}$ and inner product vector $\widehat{\mathbf{r}}$ defined by

$$\widehat{\mathbf{G}} = \left(\frac{1}{M+N}\sum_{(x,y)\in\mathbf{z}}\left(\nabla^2\ell(y, f_1(x))\right)f_k(x)f_j(x)\right)_{k,j=1}^K \quad \widehat{\mathbf{r}} = \left(\frac{1}{M+N}\sum_{(x,y)\in\mathbf{z}}\left(\nabla^2\ell(y, f_1(x))\right)f_k(x)y\right)_{k=1}^K$$

with second derivative $\nabla^2\ell(y, f(x))$ w.r.t. $f$ (computed analytically) as in equation 11 , see Appendix B for an example.

**Return:** Aggregation $\widehat{\beta}(x) = g\left(\sum_{k=1}^K \widehat{\alpha}_k f_k(x)\right)$.

---

$-1$ is $\frac{1}{2}$. Then equation 1 is equivalent to the minimization (Menon & Ong, 2016)

$$f_{\mathcal{H}} := \arg\min_{f\in\mathcal{H}} \mathcal{R}(f) \tag{3}$$

with expected risk

$$\mathcal{R}(f) := \int_{\mathcal{X}\times\mathcal{Y}} \ell(y, f(x))d\rho(x, y)$$

and the density ratio estimator $\beta_{\mathcal{H}} := g(f_{\mathcal{H}})$ can be recovered from the classifier $f_{\mathcal{H}}$. For simplicity, we assume that the problem is well-specified, i.e., $f_{\mathcal{H}} = \arg\min_{f:\mathcal{X}\to\mathbb{R}} \mathcal{R}(f)$. For instance, all the methods discussed in Example 1 can be optimized through risk minimization equation 3 by using the loss functions in the following example (Menon & Ong, 2016).

**Example 2.**

1. *KuLSIF satisfies equation 3 by* $\ell(-1, x) = \frac{1}{2}x^2$, $\ell(1, x) = -x$.
2. *Exp can be obtained by* $\ell(-1, x) = e^x$, $\ell(1, x) = e^{-x}$.
3. *SQ realize equation 3 with* $\ell(-1, x) = (1+x)^2$, $\ell(1, x) = (1-x)^2$.
4. *For LR equation 3 holds with* $\ell(-1, x) = \log(1 + e^x)$, $\ell(1, x) = \log(1 + e^{-x})$.

**Problem** Given density ratio models $\beta_1, \ldots, \beta_K : \mathcal{X} \to \mathbb{R}$, each trained with a different hyperparameter setting, the goal is to find a model $\widehat{\beta} : \mathcal{X} \to \mathbb{R}$ with minimal error $B_F(\beta, \widehat{\beta})$.

## 5  APPROACH

We approach this problem by an aggregation of models

$$\widehat{\beta} := g\left(\sum_{k=1}^K \alpha_k f_k\right) \tag{4}$$

with $f_k := g^{-1}(\beta_k)$ being the binary classifier associated with the density ratio estimator $\beta_k$, and the aggregation weights $\alpha_1, \ldots, \alpha_K$ which we compute as follows.

First, we follow Menon & Ong (2016) and Marteau-Ferey et al. (2019) to bound the Bregman divergence in equation 1 by a norm (cf. Zellinger et al. (2023) for this strategy) s.t.

$$B_F\left(\beta, \widehat{\beta}\right) - B_F(\beta, g(f_{\mathcal{H}})) = 2\left(\mathcal{R}(f) - \mathcal{R}(f_{\mathcal{H}})\right) \leq 2 \cdot \left\|g^{-1}(\widehat{\beta}) - f_{\mathcal{H}}\right\|^2 \tag{5}$$

where the norm $\|.\|$ depends on $F$ and $g$. Our approach is to choose the aggregation weights $\alpha_1, \ldots, \alpha_k$ that minimize this upper bound

$$\min_{\alpha_1, \ldots, \alpha_K \in \mathbb{R}} \left\| \sum_{k=1}^{K} \alpha_k f_k - f_{\mathcal{H}} \right\|^2 \tag{6}$$

which has three immediate advantages: (a) an analytic solution, (b) computational improvement compared to n-fold CV (see Fig. 2, Left) and (c) minimax-optimal rates (see Theorem 1).

**Analytic solution** A necessary optimality condition for (6) leads to the solution

$$L(\alpha) := \left\| \sum_{k=1}^{K} \alpha_k f_k - f_{\mathcal{H}} \right\|^2 = \left\langle \sum_{k=1}^{K} \alpha_k f_k - f_{\mathcal{H}}, \sum_{k=1}^{K} \alpha_k f_k - f_{\mathcal{H}} \right\rangle$$

$$= \sum_{k,j=1}^{K} \alpha_k \alpha_j \langle f_k, f_j \rangle - 2 \sum_{k=1}^{K} \alpha_k \langle f_k, f_{\mathcal{H}} \rangle + \langle f_{\mathcal{H}}, f_{\mathcal{H}} \rangle$$

$$\frac{\partial L(\alpha)}{\partial \alpha_k} = 2 \left( \sum_{k=1}^{K} \alpha_k \langle f_k, f_j \rangle - \langle f_k, f_{\mathcal{H}} \rangle \right) = 0$$

$$\boldsymbol{\alpha} = \mathbf{G}^{-1} \mathbf{r}$$

with $\mathbf{G} = (\langle f_k, f_j \rangle)_{k,j=1}^{K}$, $\mathbf{r} = (\langle f_k, f_{\mathcal{H}} \rangle)_{k=1}^{K}$. After discretization and using binary labels $\{-1, 1\}$ as specified in equation 2 for minimizer $f_{\mathcal{H}}$, we arrive at Algorithm 1.

**Relation to model selection methods** In the continuous case (i.e., with access to $p, q$), our aggregation strategy is at least as good as model selection:

$$\min_{\alpha_1, \ldots, \alpha_K \in \mathbb{R}} \left\| \sum_{k=1}^{K} \alpha_k f_k - f_{\mathcal{H}} \right\|^2 \leq \min_{\alpha_1, \ldots, \alpha_k \in \{0,1\}} \left\| \sum_{k=1}^{K} \alpha_k f_k - f_{\mathcal{H}} \right\|^2 \leq \min_{k \in \{1, \ldots, K\}} \|f_k - f_{\mathcal{H}}\|^2 \tag{7}$$

However, as we have only access to finite samples from $p, q$, we need to investigate the accuracy of the discretization in more detail.

## 6 RESULTS

### 6.1 THEORETICAL RESULT

In the following we show that our aggregation approach in Eq. 4 achieves minimax-optimal error rates when Eq. 1 is optimized in RKHS $\mathcal{H}$ with norm $\|.\|_{\mathcal{H}}$ and Tikhonov penalty $\|f\|_{\mathcal{H}}^2$ s.t.

$$f^{\lambda} := \arg\min_{f \in \mathcal{H}} \left[ B_F(\beta, g(f)) + \frac{\lambda}{2} \|f\|_{\mathcal{H}}^2 \right], \tag{8}$$

where $\lambda > 0$ is the regularization parameter which we aim to choose. All DRE methods in Example 1 fit into the optimization problem in equation 8, see Zellinger et al. (2023); Gruber et al. (2024). We fix a sequence $(\lambda_k)_{k=1}^{K}$ of regularization parameters and refine our aggregation equation 4 by

$$\widehat{\beta} := g\left( \sum_{k=1}^{K} \alpha_k f^{\lambda_k} \right).$$

To prove that our Algorithm 1 achieves minimax-optimal convergence rates, we introduce typical assumptions in learning theory, such as on data, the regularity of the underlying problem and the previously introduced loss function $\ell$.

**Assumption 1.** *The loss function $\ell : \mathcal{Y} \times \mathbb{R} \to \mathbb{R}$ which is used for separating $\{(x_m, 1)\}_{m=1}^{M}$ from $\{(x'_n, -1)\}_{n=1}^{N}$ has an associated link function $\psi : [0, 1] \to \mathbb{R}$ with the following properties:*

- *$\psi$ is invertible and the associated conditional Bayes risk $G(u) := u\,\ell(1, \psi(u)) + (1 - u)\,\ell(-1, \psi(u))$ is twice differentiable,*

- *the minimizer $f_{\mathcal{H}}$ satisfies $f_{\mathcal{H}}(x) = \psi(\rho(y = 1|x))$.*

**Assumption 2** (source condition). *There exist some $r \in (0, \frac{1}{2}]$, $v \in \mathcal{H}$ satisfying $f_{\mathcal{H}} = \mathbf{H}(f_{\mathcal{H}})^r v$ for the expected Hessian $\mathbf{H}(f) := \mathbb{E}_{(x,y)\sim\rho}[\nabla^2 \ell(y, f(x))]$.*

**Assumption 3** (capacity condition). *There exist $\alpha \geq 1$ and $S > 0$ such that $\mathrm{df}_\lambda \leq S\lambda^{-\frac{1}{\alpha}}$ with the degrees of freedom*

$$\mathrm{df}_\lambda := \mathbb{E}_{(x,y)\sim\rho} \left[ \left\| \mathbf{H}_\lambda(f_{\mathcal{H}})^{-\frac{1}{2}} \nabla\ell(y, f_{\mathcal{H}}(x)) \right\|_{\mathcal{H}}^2 \right]$$

*and $\mathbf{H}_\lambda(f) := \mathbf{H}(f) + \lambda I$.*

Both, source and capacity condition, are typically used in learning theory (Caponnetto & De Vito, 2007; Bauer et al., 2007; Marteau-Ferey et al., 2019; Pereverzyev, 2022) to encode the regularity of the underlying problem. We also need the following self-concordance assumption (Bach, 2010; Ostrovskii & Bach, 2021) on our loss function, which is known to be satisfied for the examples we discussed (cf. Zellinger et al. (2023)).

**Assumption 4** (Pseudo self-concordance). *For any $y \in \mathcal{Y}$, the function $\ell_y : \mathbb{R} \to \mathbb{R}$ defined by $\ell_y(\eta) := \ell(y, \eta)$ is convex, three times differentiable and satisfies*

$$\left| \ell_y'''(\eta) \right| \leq \ell_y''(\eta). \tag{9}$$

Note that by Lemma 3 in (Gruber et al., 2024), Assumption 4 implies generalized self concordance (Bach, 2010), which is also employed in Marteau-Ferey et al. (2019); Zellinger et al. (2023). As a next step we discuss the specific norms that occur in (6) and (7) in more detail. From Menon & Ong (2016); Marteau-Ferey et al. (2019) we know that

$$B_F\left(\beta, \widehat{\beta}\right) - B_F(\beta, g(f_{\mathcal{H}})) \leq 2 \left\| g^{-1}(\widehat{\beta}) - f_{\mathcal{H}} \right\|_{\mathbf{H}_\lambda(f_{\mathcal{H}})}^2$$

for $f_{\mathcal{H}}$, $\mathbf{H}_\lambda(f_{\mathcal{H}})$ as in Assumption 3, and

$$\|f\|_{\mathbf{H}_\lambda(f_{\mathcal{H}})} := \left\| \mathbf{H}_\lambda(f_{\mathcal{H}})^{1/2} f \right\|_{\mathcal{H}}. \tag{10}$$

However, the norm values $\|\cdot\|_{\mathbf{H}_\lambda(f_{\mathcal{H}})}$ are not directly accessible, as we have only observation from the measure $\rho$. That is, we have to estimate the norm to find an approximate solution to the optimization problem (6). In the following, consider some small regularization parameter value $\lambda > 0$ (as precisely specified in Section A) and the empirical Hessian and empirical risk minimizer

$$\widehat{\mathbf{H}}_\lambda(f) := \frac{1}{M+N} \sum_{(x,y)\in\mathbf{z}} \nabla^2\ell(y, f(x)) + \lambda I, \quad \widehat{f}^\lambda := \underset{f\in\mathcal{H}}{\arg\min} \frac{1}{M+N} \sum_{(x,y)\in\mathbf{z}} \ell(y, f(x)) + \frac{\lambda}{2} \|f\|_{\mathcal{H}}^2 \tag{11}$$

with combined sample $\mathbf{z}$ as defined in equation 2. It is worth noting that, for practical applications, the Hessian terms can be obtained analytically and do not incur any additional computational cost. We are now in the position of proving our main convergence rates result, the proof and constants can be found in Appendix Section A.

**Theorem 1.** *Let assumptions 1–4 and technical assumptions from Appendix A be satisfied. Consider $K > 1$, $\delta > 0$, $\{\lambda_k\}_{k=1}^K$ as defined in Appendix A and associated $\widehat{f}^{\lambda_k}$ as in equation 11. Then we have that for $\widehat{\beta}$ of Algorithm 1 applied with $\beta_k := g(\widehat{f}^{\lambda_k})$:*

$$B_F\left(\beta, \widehat{\beta}\right) - B_F(\beta, g(f_{\mathcal{H}})) \leq C(M+N)^{-\frac{2r\alpha+\alpha}{2r\alpha+\alpha+1}}, \tag{12}$$

*with probability at least $1 - (9 + 2K)\delta$, for sufficiently large sample sizes $M, N$ and some $C > 0$ that is independent of $M, N$.*

**Remark 1.** *To the best of our knowledge, Theorem 1 provides the first provably minimax-optimal convergence rates for a parameter choice procedure in DRE. Zellinger et al. (2023) provide similar rates but they rely on a heuristically fixed constant for their implementations of the balancing principle. For proofs of the minimax optimality see Caponnetto & De Vito (2007); Blanchard & Mücke (2018); Steinwart et al. (2009); Marteau-Ferey et al. (2019); Beugnot et al. (2021); Zellinger et al. (2023). While we focus on the regularization parameter, our algorithm can be applied to any hyperparameter setting of a given DRE method, see Ablations 1 and 2 in Table 1. For a detailed discussion about how noise affects this bound, we refer to Appendix Section C.*

In the following, we go beyond theoretical convergence results and conduct a comprehensive empirical evaluation of our algorithm across multiple benchmark datasets.

## 6.2 EXPERIMENTS

We evaluate on benchmarks with *known density ratios* (Kanamori et al., 2012b) where we generate ten distinct geometric datasets using Gaussian Mixture distributions in a 50-dimensional space, see Appendix D for more details.

To evaluate the effect of our algorithm in large-scale real-world scenarios we conduct extensive experiments involving the training of over 9000 deep neural networks on the task of *domain adaptation* (DA). Given a source dataset $\{x'_n\}_{n=1}^N \overset{\text{iid}}{\sim} q$ with labels $\{y_n\}_{n=1}^N$ and a target dataset $\{x_m\}_{m=1}^M \overset{\text{iid}}{\sim} p$ *without labels*, the goal is to learn a model $\phi'$ with low expected risk $\mathcal{R}(\phi') := \mathbb{E}_{(x,y)\sim p}[\ell(y, \phi'(x))]$ on the target domain without having sampled labels from the target domain. Motivated by Sugiyama et al. (2007) this task can be approached by approximating the density ratio $\frac{p}{q}$ with estimator $\widehat{\beta}$ and empirical risk minimization such that

$$\phi' = \arg\min_{\phi \in \Phi} \frac{1}{N} \sum_{n=1}^N \widehat{\beta}(x'_n)\ell(y_n, \phi(x_n)). \tag{13}$$

In our case, $\phi'$ is a downstream predictor constrained by a DA method as in Table 3 (e.g., MMDA) and optimized separately for each DA method. Our aggregation approach provides an improved density ratio estimator $\widehat{\beta}(\cdot)$, see App. D for further details. To comprehensively test our aggregation approach, we use the four popular DRE methods presented in Example 1 in all our main experiments.

**Results for known density ratios** In Table 1 we compare cross-validation-based model selection (left part of the table) to our aggregation approach (right part). Across all 10 geometrically constructed datasets, the performance of all evaluated DRE methods is clearly better for aggregation than for cross-validation. All aggregated methods consistently outperform their non-aggregated counterparts where models are selected by cross-validation. In Fig. 2 (Left) we demonstrate that our method scales computationally more favorable than 10-fold cross validation when increasing the number of hyperparameter settings.

We further conduct two ablation studies on geometric datasets to demonstrate that our approach remains broadly applicable under settings with relaxed theoretical assumptions. All other experiments conducted fall fully within the scope of our theoretical analysis. For **Ablation 1** we train several deep neural network-based logistic regression models (DeepLR) with different hyperparameter settings beyond regularization parameters (see Appendix D) and optimize objective (6) as done for the other DRE methods. In Table 1, Ablation 1 we can see that our aggregation approach improves experimental results on all datasets compared to its baseline model that was selected by cross-validation (CV). Similarly, in **Ablation 2** we use our approach to tune the kernel bandwidth parameter in Kernel Mean Matching where CV cannot be used. As baseline we set the bandwidth parameter according to the popular median heuristic (Schölkopf & Smola, 2002). In Table 1, Ablation 2 it can be seen that our aggregation method improves upon this on all datasets.

In **Ablations 3 and 4**, we evaluate our method on geometric datasets against aggregation baselines employing model averaging and ensembling strategies for KuLSIF. As shown in Tab. 1 (Ablation 3), our aggregation method consistently outperforms model averaging as proposed by Raza & Samothrakis (2019). Ablation 4 investigates how the performance changes as we increase the number of trained models. Fig. 2 (Right) demonstrates that our approach remains robust as the number of (potentially suboptimal) models increases. In contrast, advanced and recent model averaging and ensembling techniques like Raza & Samothrakis (2019), Bayesian Model Averaging (BMA) (Fragoso et al., 2018; Yeung et al., 2005), and Super Learner (SL) (Wu & Benkeser, 2024) exhibit instabilities and suffer from severe performance degradation. These findings highlight the theoretical guarantees of our method, which ensure fast error convergence when optimizing the aggregation weights. This is further illustrated in Fig. 3 (Left) in the Appendix, where we examine the weight assignments to KuLSIF estimators with different hyperparameters on dataset "c3,d1.70." More accurate models are assigned larger weights.

In **Ablation 5** we test the scalability of our method using the KuLSIF and LR DRE methods by gradually increasing the dimensionality of the Gaussian mixture datasets. As shown in Fig. 3 (Right) of the Appendix, our aggregation method (dashed lines) maintains robust performance across di-

| | Geometric Figures | | | | | | | |
|---|---|---|---|---|---|---|---|---|
| | **Cross-Validation for Binary Classifier** | | | | **Aggregation** | | | |
| **Dataset** | KuLSIF | Exp | LR | SQ | KuLSIF | Exp | LR | SQ |
| c3,d1.70 | 8.616($\pm$0.011) | 8.322($\pm$0.009) | 8.840($\pm$0.021) | 9.170($\pm$0.011) | **8.320($\pm$0.004)** | **8.151($\pm$0.011)** | **8.572($\pm$0.016)** | **8.831($\pm$0.011)** |
| c2,d1.72 | 13.031($\pm$0.005) | 12.994($\pm$0.015) | 13.255($\pm$0.013) | 13.537($\pm$0.027) | **12.854($\pm$0.007)** | **12.365($\pm$0.011)** | **13.250($\pm$0.009)** | **13.102($\pm$0.019)** |
| c2,d1.59 | 12.625($\pm$0.005) | 19.748($\pm$0.037) | 12.829($\pm$0.014) | 13.056($\pm$0.015) | **12.422($\pm$0.010)** | **12.441($\pm$0.042)** | **12.719($\pm$0.012)** | **12.615($\pm$0.011)** |
| c1,d1.55 | 11.813($\pm$0.007) | 14.477($\pm$0.103) | 12.001($\pm$0.023) | 12.179($\pm$0.013) | **11.625($\pm$0.004)** | **11.324($\pm$0.018)** | **12.001($\pm$0.015)** | **11.458($\pm$0.010)** |
| c2,d1.78 | 9.632($\pm$0.003) | 18.008($\pm$0.069) | 9.802($\pm$0.035) | 9.990($\pm$0.006) | **9.425($\pm$0.013)** | **17.043($\pm$0.015)** | **9.702($\pm$0.023)** | **9.625($\pm$0.007)** |
| c2,d1.55 | 10.371($\pm$0.007) | 9.774($\pm$0.019) | 10.555($\pm$0.059) | 10.757($\pm$0.023) | **10.001($\pm$0.002)** | **9.523($\pm$0.010)** | **10.415($\pm$0.039)** | **10.317($\pm$0.019)** |
| c3,d1.57 | 12.014($\pm$0.003) | 18.995($\pm$0.126) | 12.214($\pm$0.037) | 14.048($\pm$0.029) | **12.003($\pm$0.007)** | **12.021($\pm$0.008)** | **11.238($\pm$0.013)** | **12.940($\pm$0.018)** |
| c2,d1.61 | 11.614($\pm$0.004) | 11.282($\pm$0.034) | 11.800($\pm$0.008) | 12.242($\pm$0.007) | **11.365($\pm$0.003)** | **10.787($\pm$0.013)** | **10.920($\pm$0.008)** | **11.891($\pm$0.007)** |
| c3,d1.46 | 12.803($\pm$0.009) | 12.616($\pm$0.008) | 12.971($\pm$0.007) | 13.159($\pm$0.006) | **9.421($\pm$0.003)** | **12.025($\pm$0.010)** | **12.132($\pm$0.004)** | **12.970($\pm$0.004)** |
| c1,d1.63 | 9.527($\pm$0.006) | 9.704($\pm$0.009) | 9.732($\pm$0.014) | 9.965($\pm$0.015) | **9.397($\pm$0.008)** | **9.611($\pm$0.011)** | **9.071($\pm$0.007)** | **9.729($\pm$0.009)** |
| Avg. | 11.205($\pm$0.006) | 13.392($\pm$0.097) | 11.400($\pm$0.023) | 11.810($\pm$0.012) | **10.683($\pm$0.006)** | **11.529($\pm$0.015)** | **11.002($\pm$0.014)** | **11.348($\pm$0.012)** |

| | Ablation 1: Deep Learning | | | | | | | |
|---|---|---|---|---|---|---|---|---|
| **Methods** | c3,d1.70 | c2,d1.72 | c2d1.59 | c1d1.55 | c2d1.78 | c2d1.55 | c3d1.57 | c2d1.61 | c3d1.46 |
| DeepLR | 7.280($\pm$0.011) | 14.539($\pm$0.010) | 13.001($\pm$0.008) | 9.963($\pm$0.012) | 9.857($\pm$0.013) | 10.992($\pm$0.009) | 13.421($\pm$0.010) | 11.892($\pm$0.011) | 9.573($\pm$0.006) |
| Agg. DeepLR | **6.504($\pm$0.012)** | **14.249($\pm$0.013)** | **12.733($\pm$0.009)** | **9.631($\pm$0.014)** | **10.428($\pm$0.008)** | **12.733($\pm$0.011)** | **11.229($\pm$0.010)** | **8.954($\pm$0.005)** |

| | Ablation 2: KMM | | | | | | | |
|---|---|---|---|---|---|---|---|---|
| **Methods** | c3,d1.70 | c2,d1.72 | c2d1.59 | c1d1.55 | c2d1.78 | c2d1.55 | c3d1.57 | c2d1.61 | c3d1.46 |
| KMM Med. Heur. | 16.783($\pm$0.091) | 21.051($\pm$0.104) | 20.601($\pm$0.097) | 19.833($\pm$0.081) | 17.231($\pm$0.089) | 19.364($\pm$0.097) | 20.911($\pm$0.153) | 19.981($\pm$0.099) | 18.59($\pm$0.077) |
| KMM Agg | **13.223($\pm$0.071)** | **18.155($\pm$0.097)** | **16.532($\pm$0.083)** | **15.319($\pm$0.067)** | **13.968($\pm$0.066)** | **15.473($\pm$0.081)** | **17.003($\pm$0.114)** | **16.712($\pm$0.073)** | **14.538($\pm$0.035)** |

| | Ablation 3: Averaging | | | | | | | |
|---|---|---|---|---|---|---|---|---|
| **Methods** | c3,d1.70 | c2,d1.72 | c2d1.59 | c1d1.55 | c2d1.78 | c2d1.55 | c3d1.57 | c2d1.61 | c3d1.46 |
| Avg. KuLSIF | 8.553($\pm$0.009) | 13.151($\pm$0.008) | 12.597($\pm$0.007) | 11.803($\pm$0.011) | 9.533($\pm$0.010) | 10.292($\pm$0.009) | 12.011($\pm$0.005) | 11.524($\pm$0.003) | 10.357($\pm$0.006) |
| Agg. KuLSIF | **8.320($\pm$0.004)** | **12.854($\pm$0.007)** | **12.422($\pm$0.010)** | **11.625($\pm$0.004)** | **9.425($\pm$0.013)** | **10.001($\pm$0.002)** | **12.003($\pm$0.007)** | **11.365($\pm$0.003)** | **9.421($\pm$0.003)** |

Table 1: Mean and standard deviation (after $\pm$) of twice the Bregman divergence error on the geometrically constructed datasets following Kanamori et al. (2012b) over ten different sample draws from $P$ and $Q$.

mensions, whereas the cross-validation approach (solid lines) degrades due to growing difficulty in hyperparameter selection.

These results confirm that, compared to other averaging and ensembling methods for DRE, our approach exhibits fast error convergence and remains stable when increasing the number of models. For completeness, we also include a BMA method discussed in (Fragoso et al., 2018) and based on (Yeung et al., 2005), and Super Learner (Wu & Benkeser, 2024) as model averaging and ensembling baselines in our experimental evaluation. In Table 2 we show the performance of all compared averaging methods on the geometric datasets, results are averaged over datasets, for individual results see Table 4 in the Appendix. Consistent with prior findings, our approach consistently outperforms the baseline averaging and ensembling methods across all geometric datasets.

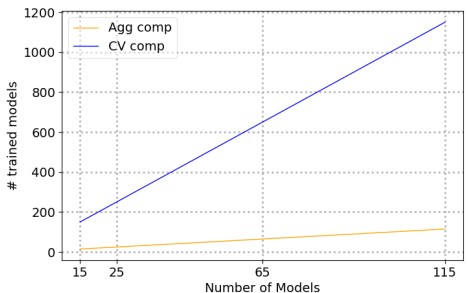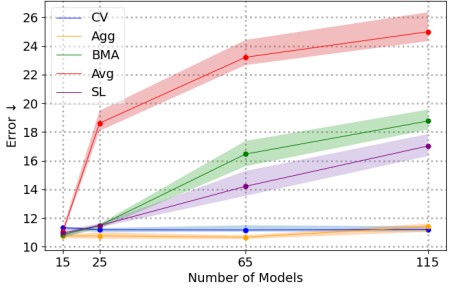

Figure 2: Left: Computational complexity, x-axis with the number of models with distinct hyperparameter settings, on y-axis the number of models that have to be trained. Our method (orange) needs to train by a factor of $n$-folds (here 10) less models compared to CV (blue). Right: Our aggregation method (orange) remains robust to models with suboptimal hyperparameters. Other averaging and ensembling methods (red, green, purple) severely degrade when increasing the number of models.

**Results for domain adaptation** We evaluate the performance of our method on three DA modalities (image, text, time-series) as conceptually introduced in Eq. 13, for more details see App. D. The results in Tab. 3 (due to space limitation disaggregated by DA methods only for MiniDomainNet) demonstrate that combining established DRE methods with our aggregation approach consistently outperforms their non-aggregated counterparts for which model selection is done by cross-

| Geometric Figures (Avg.) | | | | |
|---|---|---|---|---|
| **Method** | KuLSIF | Exp | LR | SQ |
| Cross Validation | 11.205($\pm$0.006) | 13.392($\pm$0.097) | 11.400($\pm$0.023) | 11.810($\pm$0.015) |
| Bayesian Model Averaging | 11.119($\pm$0.006) | 12.972($\pm$0.017) | 11.397($\pm$0.019) | 11.785($\pm$0.013) |
| Super Learner | 11.009($\pm$0.007) | 12.386($\pm$0.015) | 11.320($\pm$0.016) | 11.658($\pm$0.012) |
| Aggregation (Ours) | **10.683($\pm$0.006)** | **11.529($\pm$0.015)** | **11.002($\pm$0.014)** | **11.348($\pm$0.012)** |

Table 2: Mean and standard deviation (after $\pm$) of twice the Bregman divergence error on the geometrically constructed datasets following Kanamori et al. (2012b) over ten different sample draws from $P$ and $Q$ and averaged over datasets.

validation. The observed improvement is consistent across the three domain adaptation modalities and all evaluated domain adaptation methods. Notably, our approach also achieves state-of-the-art performance, surpassing the results of Dinu et al. (2023) on MiniDomainNet and Amazon Reviews. Analogous to the experiments on the geometric datasets, we include model averaging and ensembling baselines for completeness in Tab. 3. As previously, our approach consistently outperforms the competing methods. More detailed results, including the performance of individual model averaging approaches and DRE methods across specific DA scenarios, are presented in Tab. 5–74.

| Domain Adaptation: MiniDomainNet | | | | | | | | |
|---|---|---|---|---|---|---|---|---|
| | Cross-Validation for Binary Classifier | | | | Aggregation | | | |
| **DA-Method** | KuLSIF | Exp | LR | SQ | KuLSIF | Exp | LR | SQ |
| MMDA | 0.527($\pm$0.009) | 0.528($\pm$0.011) | 0.528($\pm$0.012) | 0.518($\pm$0.010) | **0.536($\pm$0.007)** | **0.539($\pm$0.011)** | **0.536($\pm$0.006)** | **0.535($\pm$0.009)** |
| CoDATS | 0.536($\pm$0.012) | 0.532($\pm$0.015) | 0.530($\pm$0.020) | 0.517($\pm$0.018) | **0.542($\pm$0.010)** | **0.543($\pm$0.014)** | **0.540($\pm$0.014)** | **0.533($\pm$0.017)** |
| DANN | 0.531($\pm$0.009) | 0.522($\pm$0.012) | 0.520($\pm$0.019) | 0.506($\pm$0.016) | **0.536($\pm$0.007)** | **0.535($\pm$0.011)** | **0.526($\pm$0.013)** | **0.519($\pm$0.015)** |
| CDAN | 0.531($\pm$0.012) | 0.531($\pm$0.017) | 0.524($\pm$0.023) | 0.512($\pm$0.021) | **0.537($\pm$0.009)** | **0.544($\pm$0.017)** | **0.535($\pm$0.017)** | **0.526($\pm$0.020)** |
| DSAN | 0.539($\pm$0.011) | 0.532($\pm$0.015) | 0.527($\pm$0.015) | 0.513($\pm$0.013) | **0.544($\pm$0.009)** | **0.546($\pm$0.014)** | **0.535($\pm$0.009)** | **0.525($\pm$0.012)** |
| DIRT | 0.517($\pm$0.026) | 0.386($\pm$0.177) | 0.520($\pm$0.023) | 0.509($\pm$0.021) | **0.523($\pm$0.025)** | **0.395($\pm$0.175)** | **0.526($\pm$0.017)** | **0.525($\pm$0.020)** |
| AdvSKM | 0.516($\pm$0.006) | 0.515($\pm$0.009) | 0.512($\pm$0.012) | 0.500($\pm$0.010) | **0.522($\pm$0.005)** | **0.529($\pm$0.008)** | **0.521($\pm$0.006)** | **0.517($\pm$0.008)** |
| HoMM | 0.531($\pm$0.008) | 0.529($\pm$0.012) | 0.521($\pm$0.015) | 0.505($\pm$0.013) | **0.539($\pm$0.007)** | **0.544($\pm$0.011)** | **0.529($\pm$0.009)** | **0.528($\pm$0.012)** |
| DDC | 0.517($\pm$0.010) | 0.517($\pm$0.013) | 0.514($\pm$0.012) | 0.500($\pm$0.010) | **0.527($\pm$0.008)** | **0.529($\pm$0.012)** | **0.524($\pm$0.008)** | **0.515($\pm$0.009)** |
| DeepCoral | 0.535($\pm$0.012) | 0.528($\pm$0.013) | 0.530($\pm$0.012) | 0.514($\pm$0.011) | **0.543($\pm$0.010)** | **0.540($\pm$0.012)** | **0.536($\pm$0.007)** | **0.527($\pm$0.009)** |
| CMD | 0.529($\pm$0.010) | 0.524($\pm$0.021) | 0.521($\pm$0.021) | 0.510($\pm$0.019) | **0.536($\pm$0.008)** | **0.538($\pm$0.020)** | **0.527($\pm$0.015)** | **0.521($\pm$0.018)** |
| Avg. | 0.528($\pm$0.011) | 0.513($\pm$0.029) | 0.523($\pm$0.017) | 0.510($\pm$0.015) | **0.535($\pm$0.010)** | **0.526($\pm$0.028)** | **0.530($\pm$0.011)** | **0.525($\pm$0.014)** |

| Domain Adaptation: MiniDomainNet (Avg.) | | | | |
|---|---|---|---|---|
| **Method** | KuLSIF | Exp | LR | SQ |
| Cross Validation | 0.528($\pm$0.011) | 0.513($\pm$0.029) | 0.523($\pm$0.017) | 0.510($\pm$0.015) |
| Bayesian Model Averaging | 0.528($\pm$0.008) | 0.514($\pm$0.027) | 0.523($\pm$0.015) | 0.507($\pm$0.015) |
| Super Learner | 0.530($\pm$0.010) | 0.514($\pm$0.027) | 0.523($\pm$0.011) | 0.511($\pm$0.013) |
| Aggregation (Ours) | **0.535($\pm$0.010)** | **0.526($\pm$0.028)** | **0.530($\pm$0.011)** | **0.525($\pm$0.014)** |

| Domain Adaptation: Amazon Reviews (Avg.) | | | | |
|---|---|---|---|---|
| **Method** | KuLSIF | Exp | LR | SQ |
| Cross Validation | 0.787($\pm$0.011) | 0.771($\pm$0.010) | 0.785($\pm$0.011) | 0.781($\pm$0.009) |
| Bayesian Model Averaging | 0.787($\pm$0.012) | 0.772($\pm$0.010) | 0.785($\pm$0.011) | 0.781($\pm$0.009) |
| Super Learner | 0.788($\pm$0.010) | 0.773($\pm$0.009) | 0.785($\pm$0.010) | 0.782($\pm$0.008) |
| Aggregation (Ours) | **0.795($\pm$0.009)** | **0.783($\pm$0.009)** | **0.789($\pm$0.011)** | **0.789($\pm$0.007)** |

| Domain Adaptation: HHAR (Avg.) | | | | |
|---|---|---|---|---|
| **Method** | KuLSIF | Exp | LR | SQ |
| Cross Validation | 0.737($\pm$0.080) | 0.694($\pm$0.085) | 0.772($\pm$0.019) | 0.734($\pm$0.011) |
| Bayesian Model Averaging | 0.738($\pm$0.077) | 0.711($\pm$0.084) | 0.772($\pm$0.018) | 0.738($\pm$0.013) |
| Super Learner | 0.740($\pm$0.076) | 0.721($\pm$0.077) | 0.773($\pm$0.015) | 0.743($\pm$0.010) |
| Aggregation (Ours) | **0.780($\pm$0.082)** | **0.737($\pm$0.075)** | **0.785($\pm$0.014)** | **0.759($\pm$0.010)** |

Table 3: Mean and standard deviation (after $\pm$) of target classification accuracy on MiniDomainNet, Amazon Reviews and HHAR datasets over three different random initialization of model weights and several domain adaptation scenarios.

## 7 CONCLUSION

To approach hyperparameter choice issues in DRE, we proposed a novel model aggregation algorithm that trains models with distinct hyperparameter settings and subsequently aggregates them. The aggregation coefficients are optimized to achieve minimax-optimal convergence rates without requiring prior knowledge of density ratio smoothness. Empirical evaluations on DRE benchmarks and large-scale domain adaptation tasks demonstrate that our method consistently outperforms established DRE methods, and model averaging and ensembling baselines, achieving new state-of-the-art results on MiniDomainNet and Amazon Reviews.

## REPRODUCIBILITY STATEMENT

We have taken several steps to ensure the reproducibility of our results. All theoretical assumptions and complete proofs of our main results are provided in Appendix A . Details on the derivation of loss functions and Hessians used in our algorithms are included in Appendix B . For empirical reproducibility, Appendix D provides comprehensive information on experimental protocols, including dataset generation for benchmarks with known density ratios, preprocessing steps for MiniDomain-Net, HHAR, and Amazon Reviews datasets, as well as details on compared methods and hyperparameter ranges . We also report full ablation studies, experimental setups, and results tables in the appendix to facilitate verification. A reference implementation of the methods presented in this paper is available at: https://github.com/lugruber/dre_agg.

## LLM USAGE

An LLM was used solely to aid with polishing the writing. It did not contribute to research ideation, analysis, or content creation.

## ACKNOWLEDGMENTS

The ELLIS Unit Linz, the LIT AI Lab, the Institute for Machine Learning, are supported by the Federal State Upper Austria. We thank the projects FWF AIRI FG 9-N (10.55776/FG9), AI4GreenHeatingGrids (FFG- 899943), Stars4Waters (HORIZON-CL6-2021-CLIMATE-01-01), FWF Bilateral Artificial Intelligence (10.55776/COE12). We thank NXAI GmbH, Audi AG, Silicon Austria Labs (SAL), Merck Healthcare KGaA, GLS (Univ. Waterloo), TÜV Holding GmbH, Software Competence Center Hagenberg GmbH, dSPACE GmbH, TRUMPF SE + Co. KG. We acknowledge EuroHPC Joint Undertaking for awarding us access to o Karolina at IT4Innovations, Czech Republic, Leonardo at CINECA, Italy, MeluXina at LuxProvide, Luxembourg, LUMI at CSC, Finland and MareNostrum5 as BSC, Spain.

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

# A PROOF OF THEOREM 1

**Sketch of the proof:**

- First we bound the excess risk $\mathcal{R}(\widehat{f}^\lambda) - \mathcal{R}(f_\mathcal{H})$ of the empirical estimator by $\left\| \widehat{f}^\lambda - f_\mathcal{H} \right\|^2_{\mathbf{H}_\lambda(f_\mathcal{H})}$ using results from Marteau-Ferey et al. (2019).

- To obtain bounds for $\min_{1 \leq k \leq K} \left\| \widehat{f}^{\lambda_k} - f_\mathcal{H} \right\|^2_{\mathbf{H}_{\lambda_k}(f_\mathcal{H})}$ we construct an admissible sequence of parameters $\{\lambda_k\}_{k=1}^K$ and use some techniques developed for analyzing Lepskii's balancing principle (Lepskii, 1991; Goldenshluger & Pereverzev, 2000; Birgé, 2001; Mathé, 2006; De Vito et al., 2010; Mücke, 2018; Blanchard et al., 2019; Zellinger et al., 2021; Pereverzyev, 2022).

- Finally we show (12) by using several concentration bounds on the Hessian-weighted norms to conclude the proof.

To prove our main theorem, we will need to recall several concentration bounds on weighted norms, which were to a large extend established by Marteau-Ferey et al. (2019). We will need the following technical assumption:

**Assumption 5** (technical assumption)**.**
- *The kernel $k$ is continuous and bounded $\sup_{x \in \mathcal{X}} \|k(x, \cdot)\|_\mathcal{H} \leq R$.*

- *Let $z = (x, y) \in \mathcal{X} \times \mathcal{Y}$ and denote $\ell_z(f) = \ell(y, f(x))$. Then the quantities $|\ell_z(0)|, \|\nabla \ell_z(0)\|_\mathcal{H}, \mathrm{Tr}(\nabla^2 \ell_z(0))$ are almost surely (wrt. $\rho$) bounded.*

The first estimate deals with estimates for empirical risk minimizers $\widehat{f}^\lambda$ in weighted norms.

**Lemma 1** (Marteau-Ferey et al. (2019), Theorem 38)**.** *Let Assumptions 1–5 be satisfied, $\delta \in (0, \frac{1}{2}]$, and define $B_1^*$, $B_2^*$ and $L$ by*

$$B_1^* := \sup_{z \in \mathrm{supp}(\rho)} \mathrm{Tr}(\nabla \ell_z(f_\mathcal{H})), \quad B_2^* := \sup_{z \in \mathrm{supp}(\rho)} \mathrm{Tr}(\nabla^2 \ell_z(f_\mathcal{H})), \quad L := \|f_\mathcal{H}\|_{\mathbf{H}^{-2r}(f_\mathcal{H})}.$$

*Whenever $0 < \lambda \leq \min\{B_2^*, (2LR\log\frac{2}{\delta})^{-1/r}, S^{2\alpha}(B_2^*)^\alpha(B_1^*)^{-2\alpha}\}$ with $S$ as in Assumption 3 and $M + N$ is larger than*

$$\max\left\{ 5184 \frac{B_2^*}{\lambda} \log\left( \frac{8 \cdot 414^2 B_2^*}{\lambda\delta} \right), \frac{1296 S^2}{L^2 \lambda^{1+2r+1/\alpha}} \right\},$$

*then a minimizer $\widehat{f}^\lambda$ of equation 11 satisfies, with probability at least $1 - 2\delta$,*

$$\mathcal{R}(\widehat{f}^\lambda) - \mathcal{R}(f_\mathcal{H}) \leq \left\| \widehat{f}^\lambda - f_\mathcal{H} \right\|^2_{\mathbf{H}(f_\mathcal{H})} \leq \left\| \widehat{f}^\lambda - f_\mathcal{H} \right\|^2_{\mathbf{H}_\lambda(f_\mathcal{H})} \tag{14}$$

$$\leq 414 \frac{S^2}{(M+N)\lambda^{1/\alpha}} \log\left( \frac{2}{\delta} \right) + 414 L^2 \lambda^{1+2r} =: S(M+N, \delta, \lambda) + A(\lambda).$$

Next we use similar arguments as in Zellinger et al. (2023) to construct an admissible sequence of regularization parameters and associated regularized estimators, which we want to aggregate in the end. The same reasoning as used in the well known Lepskii balancing principle (Lepskii, 1991; Goldenshluger & Pereverzev, 2000; Birgé, 2001; Mathé, 2006; De Vito et al., 2010; Mücke, 2018; Blanchard et al., 2019; Lu et al., 2020; Zellinger et al., 2021; Pereverzyev, 2022) will give us, that the minimal (over all admissible values for $\lambda$) risk difference to the target even achieves the optimal rate:

**Lemma 2.** *Let Assumptions 1– 5 be satisfied and let*

$$\lambda_k := \lambda_0 \cdot \xi^k, k \in \{1, \ldots, K\}$$

*with $\xi > 1$, $\lambda_0 \leq \xi^{-K} \min\left\{ B_2^*, \left(2LR\log\frac{2}{\delta}\right)^{-\frac{1}{r}}, S^{2\alpha}(B_2^*)^\alpha(B_1^*)^{-2\alpha} \right\}$, $K < \frac{e^{1296}}{4} - 2$ and $\delta \in [\frac{2}{e^{1296}}, \frac{1}{4+2K}]$. Moreover, let $\eta = 1296 \log^{-1}\left(\frac{2}{\delta}\right)$ and $\lambda^*$ be the solution of $\eta S(M+N, \delta, \lambda^*) = A(\lambda^*)$. Then $\lambda_1, ..., \lambda_K$ and $\lambda^*$ satisfy the assumptions of the previous Lemma 1 and we have for $k = 1, ..., K$:*

$$\left\| \widehat{f}^{\lambda_k} - f_{\mathcal{H}} \right\|^2_{\mathbf{H}_{\lambda_k}(f_{\mathcal{H}})} \leq S(M + N, \delta, \lambda_k) + A(\lambda_k). \tag{15}$$

*and even more:*

$$\min_{1 \leq k \leq K} \left\| \widehat{f}^{\lambda_k} - f_{\mathcal{H}} \right\|^2_{\mathbf{H}_{\lambda_k}(f_{\mathcal{H}})} \leq C^*(S(M + N, \delta, \lambda^*) + A(\lambda^*)) = C^*(M + N)^{-\frac{2r\alpha+\alpha}{2r\alpha+\alpha+1}}. \tag{16}$$

*with probability at least $1 - (4 + 2K)\delta$, for large enough sample size $M + N$ greater than*

$$\max \left\{ 5184 \frac{B_2^*}{\lambda_0} \log \left( \frac{8 \cdot 414^2 B_2^*}{\lambda_0 \delta} \right), \frac{1296 S^2}{L^2 \lambda_0^{1+2r+1/\alpha}} \right\} \tag{17}$$

*and $C^* := 16560 \xi^{\max(1+2r,1/\alpha)} L^2 \left( \frac{1296 S^2}{L^2} \right)^{\frac{\alpha+2r\alpha}{1+2\alpha r+\alpha}}$.*

*Proof.* It has been shown in the proof of Zellinger et al. (2023, Theorem 1) that under the given conditions, $\lambda_k$ for $i = 1, ..., K$ as well as $\lambda^*$ are admissible according to Lemma 1, which establishes (14). Let us now show (16). First note that $S$ is a decreasing and $A$ an increasing function with respect to $\lambda$, so that $\lambda^*$ is a minimizer of $S + A$. Due to the structure of $S + A$, it is also clear that there is an index $1 \leq j \leq K$ with $\lambda_j \leq \lambda^* \leq \lambda_{j+1}$ so that

$$\lambda_\dagger := \underset{\lambda \in \{\lambda_1, ..., \lambda_K\}}{\arg\min} S(M + N, \delta, \lambda) + A(\lambda) \in \{\lambda_j, \lambda_{j+1}\}.$$

We now distinguish two cases: Let us first consider $\lambda_\dagger = \lambda_j$. Then $\lambda_\dagger \geq \lambda^* \frac{1}{\xi}$ which on the other hand implies $S(\lambda_\dagger) = \frac{B(\delta)}{\lambda_\dagger^{1/\alpha}(M+N)} \leq B(\delta)(\lambda^*)^{-1/\alpha}\xi^{1/\alpha}(M + N)^{-1} = \xi^{1/\alpha}S(\lambda^*)$ (for $B(\delta) = 414 S^2 \log \left( \frac{2}{\delta} \right)$) and $A(\lambda_\dagger) \leq A(\lambda^*) \leq \xi^{1/\alpha}A(\lambda^*)$.

In the second case we assume $\lambda_\dagger = \lambda_{j+1}$, which yields $\lambda_\dagger \leq \lambda^*\xi$ and therefore $A(\lambda_\dagger) \leq A(\xi\lambda^*) = A(\lambda^*)\xi^{1+2r}$ and also $S(\lambda_\dagger) \leq S(\lambda^*) \leq \xi^{1+2r}S(\lambda^*)$. Thus in both cases: $S(\lambda_\dagger) + A(\lambda_\dagger) \leq \xi^{\max(1+2r,1/\alpha)}(S(\lambda^*) + A(\lambda^*))$, implying (16). $\square$

As a next step, let us state some concentration bounds on weighted norms, that relate on the one hand the Hessian norms $\mathbf{H}_\lambda(f)$ evaluated for different $f \in \mathcal{H}$, where on the other hand, the connections between empirical and non-empirical Hessian norms are explored.

To do so, we will also need to introduce further notation: we write $\mathbf{B} \preccurlyeq \mathbf{A}$ iff $\mathbf{A} - \mathbf{B}$ is positive semi-definite. Moreover, we define $\mathbf{t}(f) := \sup_{(x,y)\in\text{supp}(\rho)} \sup_{g\in\{yk(x,\cdot)\}} |f \cdot g|$.

**Lemma 3** (Marteau-Ferey et al. (2019), Proposition 15)**.** *Let Assumptions 1–5 be satisfied, $\lambda \geq 0$ and $f_1, f_2 \in \mathcal{H}$. Then, we have*

$$\mathbf{H}_\lambda(f_1) \preccurlyeq e^{\mathbf{t}(f_1-f_2)}\mathbf{H}_\lambda(f_2). \tag{18}$$

**Lemma 4** (Marteau-Ferey et al. (2019); Zellinger et al. (2023))**.** *Under the conditions of Lemma 1, we have, with probability at least $1 - 2\delta$,*

$$\mathbf{t}(f_{\mathcal{H}} - f^\lambda) \leq \log(2), \quad \mathbf{t}(f^\lambda - \widehat{f}^\lambda) \leq \log(2) \quad \mathbf{t}(f_{\mathcal{H}} - \widehat{f}^\lambda) \leq 2\log(2). \tag{19}$$

**Lemma 5** (Marteau-Ferey et al. (2019), Rudi & Rosasco (2017))**.** *Let the conditions of Lemma 1 be satisfied. Then, it holds, with probability at least $1 - \delta$,*

$$\mathbf{H}_\lambda(f) \preccurlyeq 2\widehat{\mathbf{H}}_\lambda(f). \tag{20}$$

*If, in addition, $0 < \|\mathbf{H}(f)\|_{\mathcal{H}}$, then it holds, for all*

$$M + N \geq \frac{16 B_2^*}{\|\mathbf{H}(f)\|_{\mathcal{H}}} \log\left(\frac{2}{\delta}\right), \tag{21}$$

*with probability at least $1 - \delta$,*

$$\widehat{\mathbf{H}}_\lambda(f) \preccurlyeq \frac{3}{2}\mathbf{H}_\lambda(f). \tag{22}$$

We will need another technical assumption in order to apply (22) to a sequence of models:

**Assumption 6.** *For the sequence $(\lambda_k)_{k=1}^K$ there exists $b^* > 0$ with $b^* \leq \min_{k \in \{1,...,K\}} \left\| \mathbf{H}(\widehat{f}^{\lambda_k}) \right\|_{\mathcal{H}}$.*

We are now in the position to formulate and prove a detailed version of our finite sample bounds for aggregation of empirical DR-estimators:

**Theorem 2.** *Let assumptions 1–6 be fulfilled. Consider $K > 1$ and a sequence $\lambda_k$ of regularization parameters and associated empirical risk minimizers $\widehat{f}^{\lambda_k}$ for $0 \leq k \leq K$, as defined in Lemma 2. Let moreover $\delta \in [\frac{2}{e^{1296}}, \frac{1}{9+2K}]$. Then we have that for $\widehat{\beta}$ of Algorithm 1 applied with $\beta_k := g(\widehat{f}^{\lambda_k})$:*

$$B_F\left(\beta, \widehat{\beta}\right) - B_F(\beta, g(f_{\mathcal{H}})) \leq C(M+N)^{-\frac{2r\alpha+\alpha}{2r\alpha+\alpha+1}}, \tag{23}$$

*with probability at least $1 - (9+2K)\delta$ for $M + N$ larger than (26) and $C$ given by equation 25.*

*Proof.* If the inequalities that we are going to state hold with high probability (under conditions mentioned in the theorem), we will write $\leq_\rho$. Let us start by following Menon & Ong (2016):

$$B_F\left(\beta, g\left(\sum_{k=1}^K \widehat{\alpha_k}\widehat{f}^{\lambda_k}\right)\right) - B_F(\beta, g(f_{\mathcal{H}})) = 2\left(\mathcal{R}\left(\sum_{k=1}^K \widehat{\alpha_k}\widehat{f}^{\lambda_k}\right) - \mathcal{R}(f_{\mathcal{H}})\right).$$

Next we apply (14), combine it with (18) and (19) to obtain:

$$B_F\left(\beta, g\left(\sum_{k=1}^K \widehat{\alpha_k}\widehat{f}^{\lambda_k}\right)\right) - B_F(\beta, g(f_{\mathcal{H}})) \underbrace{\leq_\rho}_{(A)} 2\left\|\sum_{k=1}^K \widehat{\alpha_k}\widehat{f}^{\lambda_k} - f_{\mathcal{H}}\right\|_{\mathbf{H}_{\lambda_0}(f_{\mathcal{H}})}^2$$

$$\underbrace{\leq_\rho}_{(B)} 8\left\|\sum_{k=1}^K \widehat{\alpha_k}\widehat{f}^{\lambda_k} - f_{\mathcal{H}}\right\|_{\mathbf{H}_{\lambda_0}(\widehat{f}^{\lambda_0})}^2.$$

An application of (20) then yields

$$8\left\|\sum_{k=1}^K \widehat{\alpha_k}\widehat{f}^{\lambda_k} - f_{\mathcal{H}}\right\|_{\mathbf{H}_{\lambda_0}(\widehat{f}^{\lambda_0})}^2 \underbrace{\leq_\rho}_{(C)} 16\left\|\sum_{k=1}^K \widehat{\alpha_k}\widehat{f}^{\lambda_k} - f_{\mathcal{H}}\right\|_{\widehat{\mathbf{H}}_{\lambda_0}(\widehat{f}^{\lambda_0})}^2$$

$$= 16 \min_{\alpha_1,...,\alpha_K \in \mathbb{R}}\left\|\sum_{k=1}^K \alpha_k\widehat{f}^{\lambda_k} - f_{\mathcal{H}}\right\|_{\widehat{\mathbf{H}}_{\lambda_0}(\widehat{f}^{\lambda_0})}^2$$

$$\leq 16 \min_{k \in \{1,...,K\}}\left\|\widehat{f}^{\lambda_k} - f_{\mathcal{H}}\right\|_{\widehat{\mathbf{H}}_{\lambda_0}(\widehat{f}^{\lambda_0})}^2, \tag{24}$$

where the last inequality follows from (7). Next we use that $\lambda_0 < \lambda_k$ to upper-bound (24) by:

$$\min_{k \in \{1,...,K\}}\left\|\widehat{f}^{\lambda_k} - f_{\mathcal{H}}\right\|_{\widehat{\mathbf{H}}_{\lambda_0}(\widehat{f}^{\lambda_0})}^2 \leq \min_{k \in \{1,...,K\}}\left\|\widehat{f}^{\lambda_k} - f_{\mathcal{H}}\right\|_{\widehat{\mathbf{H}}_{\lambda_k}(\widehat{f}^{\lambda_0})}^2.$$

We can further apply (22), which is valid due to Assumption 6, and moreover, we use (18) and (19), so that we end up with:

$$16 \min_{k \in \{1,...,K\}}\left\|\widehat{f}^{\lambda_k} - f_{\mathcal{H}}\right\|_{\widehat{\mathbf{H}}_{\lambda_k}(\widehat{f}^{\lambda_0})}^2 \underbrace{\leq_\rho}_{(D)} 24 \min_{k \in \{1,...,K\}}\left\|\widehat{f}^{\lambda_k} - f_{\mathcal{H}}\right\|_{\mathbf{H}_{\lambda_k}(\widehat{f}^{\lambda_0})}^2$$

$$\underbrace{\leq_\rho}_{(E)} 96 \min_{k \in \{1,...,K\}}\left\|\widehat{f}^{\lambda_k} - f_{\mathcal{H}}\right\|_{\mathbf{H}_{\lambda_k}(f_{\mathcal{H}})}^2$$

$$\underbrace{\leq_\rho}_{(F)} C(M+N)^{-\frac{2r\alpha+\alpha}{2r\alpha+\alpha+1}},$$

where (F) follows from invoking equation 16 and the constant is given by

$$C := 96C^* = 96 \cdot 16560 \xi^{\max(1+2r,1/\alpha)} L^2 \left( \frac{1296S^2}{L^2} \right)^{\frac{\alpha+2r\alpha}{1+2\alpha r+\alpha}}, \tag{25}$$

so that equation 23 is established. Note that by Lemma 5, (F) holds with probability $1 - (4 + 2K)\delta$ and (A)-(E) all hold with probability $1 - \delta$, so that (23) is valid with probability $1 - (9 + 2K)\delta$, yielding the mentioned admissible range for $\delta$. Moreover, taking into account the requirements (17) and (21), we obtain that $M + N$ needs to be at least as large as

$$\max \left\{ 5184 \frac{B_2^*}{\lambda_0} \log \left( \frac{8 \cdot 414^2 B_2^*}{\lambda_0 \delta} \right), \frac{1296S^2}{L^2 \lambda_0^{1+2r+1/\alpha}}, \frac{16B_2^*}{b^*} \log \left( \frac{2}{\delta} \right) \right\}. \tag{26}$$

$\square$

## B   DERIVATION OF LOSS AND HESSIAN FOR LR

Starting with $F, g$ of LR from Example 1 we have

$$F(x) := x \log(x) - (1 + x) \log(1 + x)$$
$$g(x) := e^x$$

From Theorem 1 from Zellinger (2025) we have

$$\psi^{-1}(x) := \frac{g(x)}{1 + g(x)}$$

$$\frac{\psi^{-1}(x)}{1 - \psi^{-1}(x)} = e^x$$

$$\gamma(\eta) := -F \left( \frac{\eta}{1 - \eta} \right) (1 - \eta)$$

$$= - \left( \frac{\eta}{1 - \eta} \log \left( \frac{\eta}{1 - \eta} \right) - \left( 1 + \frac{\eta}{1 - \eta} \right) \log \left( 1 + \frac{\eta}{1 - \eta} \right) \right) (1 - \eta).$$

Using these terms leads after simplification to

$$\gamma \left( \psi^{-1}(x) \right) = -F \left( e^x \right) \left( 1 - \psi^{-1}(x) \right) = \frac{x}{1 + e^x} + \log \left( 1 + e^x \right) - x$$

$$\gamma'(\eta) = \frac{\partial}{\partial \eta} \left( - \left( \frac{\eta}{1 - \eta} \log \left( \frac{\eta}{1 - \eta} \right) - \left( 1 + \frac{\eta}{1 - \eta} \right) \log \left( 1 + \frac{\eta}{1 - \eta} \right) \right) (1 - \eta) \right)$$

$$= -\log \left( \frac{\eta}{1 - \eta} \right)$$

Taking the result for loss function $\ell(y, x)$ from Zellinger (2025) gives us further after simplifying

$$\ell(-1, x) = \gamma(\psi^{-1}(x)) - \psi^{-1}(x)\gamma'(\psi^{-1}(x))$$

$$= \frac{x}{1 + e^x} + \log(1 + e^x) - x - \frac{e^x}{1 + e^x}(-x) = \log(1 + e^x)$$

$$\ell(1, x) = \gamma(\psi^{-1}(x)) + (1 - \psi^{-1}(x))\gamma'(\psi^{-1}(x))$$

$$= \frac{x}{1 + e^x} + \log(1 + e^x) - x + \left( 1 - \frac{e^x}{1 + e^x} \right)(-x) = \log(1 + e^{-x})$$

which is the loss function for LR in Example 2. In the next step we derive the second derivative $\nabla^2 \ell(y, f(x))$ w.r.t. $f$ as used in Algorithm 1. Taking LR loss we get after simplification

$$\nabla \ell(-1, f(x)) = \frac{e^{f(x)}}{1 + e^{f(x)}},$$

$$\nabla^2 \ell(-1, f(x)) = \frac{e^{f(x)}}{(1 + e^{f(x)})^2},$$

$$\nabla \ell(1, f(x)) = -\frac{1}{1 + e^{f(x)}},$$

$$\nabla^2 \ell(1, f(x)) = \frac{e^{f(x)}}{(1 + e^{f(x)})^2}.$$

Other loss functions with respective Hessians can be derived analogously.

## C    DISCUSSION ABOUT MODEL AND DATA NOISE

Model output variance / noise is handled by the least squares construction of the weights $\widehat{\boldsymbol{\alpha}} = \widehat{\mathbf{G}}^{-1}\widehat{\mathbf{r}}$: Consider a noisy model $f^{\lambda_k}$. Its self-correlation $\widehat{G}_{kk}$, i.e., the $k$-th diagonal element of the empirical Gram matrix, becomes large, while its alignment with the target function, $\widehat{r}_k = \langle f_k, f_{\mathcal{H}} \rangle$, remains small because noise does not correlate with the true signal. This is desirable because the optimal weights satisfy $\widehat{\boldsymbol{\alpha}} = \widehat{\mathbf{G}}^{-1}\widehat{\mathbf{r}}$, and, in particular, the contribution of model $k$ is governed by the ratio $\widehat{r}_k/\widehat{G}_{kk}$ (in the diagonal approximation, intuitively $\widehat{\alpha}_k \approx \widehat{r}_k/\widehat{G}_{kk}$). Hence, a large denominator and small numerator force $\widehat{\alpha}_k$ to be small, ensuring that noisy or unstable models are down-weighted in the aggregation.

Data noise appears in two places: First, in Lemma 5, the deviation between the population Hessian $\mathbf{H}$ and the empirical Hessian $\widehat{\mathbf{H}}$ is controlled exactly as in classical uniform per-sample operator–trace bounds (e.g., matrix Bernstein), through the constant $B_2^*$, see Eq. (20)–(22). Second, in Lemma 4, the deviation measure $\mathbf{t}(f_{\mathcal{H}} - \widehat{f}^{\lambda_0})$ is controlled in the standard way used in self-concordant and kernel-learning analyses, via the bounded kernel norm $R$, the bounded derivative $|\nabla \ell_z(0)|$, and the Hessian trace bounds $B_1^*$ and $B_2^*$. Summarizing, these quantities affect the concentration bounds and therefore the multiplicative constant $C$ in the final bound (12). Larger values increase the deviation terms controlling the empirical–population differences and enlarge the sample size required for the guarantees to hold. Importantly, however, they do not influence the non-asymptotic rate in $M + N$, which remains unchanged.

## D    DETAILS ON EXPERIMENTS

### D.1    DETAILS ON DATASETS WITH KNOWN DENSITY RATIOS

To evaluate the accuracy of our aggregation algorithm, we build upon the methodologies proposed by Kanamori et al. (2012b); Nguyen et al. (2010). Specifically, we construct high-dimensional data with precisely known density ratios, enabling a systematic analysis of the estimation accuracy. Our approach extends and generalizes the settings used in prior studies. For instance, widely used datasets such as Ringnorm and Twonorm (Breiman, 1996) represent specific cases of Gaussian mixture models, where the mean and covariance parameters are set to predefined values for each mixture component. Similarly, the experiments described in Nguyen et al. (2010) can also be framed within a Gaussian mixture model structure. To introduce greater complexity, we generate distributions by randomly sampling the number, weights, and covariances of the mixture components from 50-dimensional space, which exceeds the dimensionality considered in existing studies. More concretely, the means are sampled uniformly from $[0, 0.5]^{50}$, while the weights of the mixture components are sampled from $[0, 1]$ and subsequently normalized to sum to 1. Each distribution (source and target) is assigned a distinct Gaussian Mixture distribution. We use the following naming convention for the Gaussian-mixture-based datasets. The notation "c3, d1.70" serves as an identifier for one of the 10 Gaussian–mixture–based datasets used in our evaluation. Here, "c3" indicates that

the denominator distribution of the density ratio has 3 mixture components, and "d1.70" denotes the minimum separation (1.70) between any pair of modes across both the numerator and denominator distributions. The remaining datasets follow the same naming convention, and each contains 2,500 samples.

## D.2    DETAILS ON DATASETS FOR DOMAIN ADAPTATION

We use the MiniDomainNet (Peng et al., 2019; Zellinger et al., 2017), Heterogeneity Activity Recognition (Stisen et al., 2015), and Amazon Reviews (Blitzer et al., 2006) dataset to cover image, time series (sensory), and language modalities respectively for evaluating the performance of our algorithm. Each dataset consists of a number of domains which induce several domain adaptation scenarios with the goal of adapting prediction models from a source to a target domain as described in equation 13.

**Image data for domain adaptation.**    The DomainNet-2019 dataset (Peng et al., 2019) includes the six distinct image domains "Real", "Clipart", "Quickdraw", "Sketch", "Painting", and "Infograph". Following the approach of Zellinger et al. (2017) we utilize a simplified version of this dataset known as MiniDomainNet. This reduced version narrows the focus to the five largest classes in the training set across all six domains. To optimize computational efficiency we use a ResNet-18 (He et al., 2016) trained on ImageNet (Krizhevsky et al., 2012). This pre-trained backbone is assumed to have already learned low-level filters effective for the "Real" image domain requiring adaptation only for the remaining five domains resulting in five domain adaptation tasks.

**Time series (sensory) data for domain adaptation.**    The Heterogeneity Activity Recognition dataset (Stisen et al., 2015) explores the variations specific to sensors, devices, and workloads for human activity recording. It utilizes data collected from 36 different smartphones and smartwatches comprising 13 different device models from four manufacturers. Our experimental setup incorporates all five domain adaptation scenarios analyzed in Ragab et al. (2023).

**Text data for domain adaptation.**    The Amazon Reviews dataset (Blitzer et al., 2006) contains bag-of-words representations of textual reviews across four categories: books, DVDs, kitchen, and electronics with binary labels indicating the class of review. Each category represents a different domain from which a total of twelve domain adaptation tasks are constructed by pairing each domain as a source domain with every other domain as a target domain.

## D.3    DETAILS ON COMPARED METHODS

To test our aggregation approach we pick four popular representatives of the large class of widely used DRE methods that can be modeled as Bregman divergences as in equation 1. For this we use the methods presented in Example 1: KuLSIF (Kanamori et al., 2009a), Exp (Menon & Ong, 2016), SQ (Menon & Ong, 2016), and LR (Bickel et al., 2009) with cross-validation as hyperparameter selection procedure and Bayesian Model Averaging (Fragoso et al., 2018) and Super Learner (Wu & Benkeser, 2024) as baselines and compare these for each of the four DRE methods against our aggregation approach. For KuLSIF, which offers a closed-form solution, no numerical optimization is required. For the other methods, we utilize the CG algorithm from Python's SciPY library (Virtanen et al., 2020), employing the Polak-Ribière line search strategy (Hager & Zhang, 2006). In our domain adaptation experiments, we adhere to the evaluation protocol established by Dinu et al. (2023); Gruber et al. (2024). Specifically, we calculate an ensemble of various deep neural networks using the importance-weighted functional regression approach. We use the density ratio estimates from the aggregated and non-aggregated versions of KuLSIF, Exp, SQ, and LR as importance weights. To comprehensively assess the impact on different domain adaptation techniques, we generate ensemble model candidates for 11 domain adaptation methods. These include Minimum Discrepancy Estimation for Deep Domain Adaptation (MMDA) (Rahman et al., 2020), the Convolutional deep Domanin Adaptation model for Time-Series data (CoDATS) (Wilson et al., 2020), Domain-Adversarial Neural Networks (DANN) (Ganin et al., 2016), Conditional Adversarial Domain Adaptation (CDAN) (Long et al., 2018), Deep Subdomain Adaptation (DSAN) (Zhou et al., 2021), the DIRT-T approach to Unsupervised Domain Adaptation (DIRT) (Shu et al., 2018), Adversarial Spectral Kernel Matching (AdvSKM) (Liu & Xue, 2021), Higher-order Moment Match-

ing (HoMM) (Chen et al., 2020), Deep Domain Confusion (DDC) (Tzeng et al., 2014), Correlation Alignment via Deep Neural Networks (Deep Coral) (Sun et al., 2017), and Central Moment Discrepancy (CMD) (Zellinger et al., 2017). Altogether, this benchmark involved training over 9,000 deep neural network models and evaluating the complete ensembling benchmark proposed by Dinu et al. (2023) for Amazon Reviews, MiniDomainNet, and HHAR. The experiments on these datasets were conducted to compare 16 DRE methods (with and without aggregation).

### D.3.1 KNOWN DENSITY RATIOS

For each Gaussian mixture dataset 5,000 samples are drawn from the underlying distributions. The selection and evaluation of the baseline DRE methods are conducted using a standard train/validation/test split approach with split ratios 64/16/20 respectively. The corresponding aggregated DRE methods are trained and evaluated on the same splits while the aggregation weights are computed on the respective validation sets. For model selection, the regularization parameter $\lambda$ is selected by cross-validation using grid-search from the range $\{10^{-6}, 10^{-5}, \ldots, 10^4\}$, whereas for model aggregation the models corresponding to distinct $\lambda$ values are aggregated. Each experiment is repeated 10 times to ensure statistical robustness. Consistent with Kanamori et al. (2012a), a Gaussian kernel is employed for all density ratio estimation methods, with the kernel width determined using the median heuristic (Schölkopf & Smola, 2002).

### D.3.2 ABLATION STUDIES

Ablation 1: We use a deep neural network based logistic regression model trained with binary cross-entropy loss, different regularization parameters in $\{10^{-6}, 10^{-5}, \ldots, 10^4\}$, number of hidden layers in $\{1, 2, 4, 6, 8\}$, different layer width settings, and learning rates in $\{10^{-2}, 10^{-3}, 10^{-4}, 10^{-5}\}$. The model is optimized by gradient descent for 125 epochs. We evaluate this model on geometrically-constructed datasets by using the exact same data splits as for the other DRE methods.

In the remaining ablation studies we follow the experimental protocol from our main experiments on the geometric Gaussian mixture datasets. For Ablations 4 and 5 we illustrate median error and 50% confidence intervals.

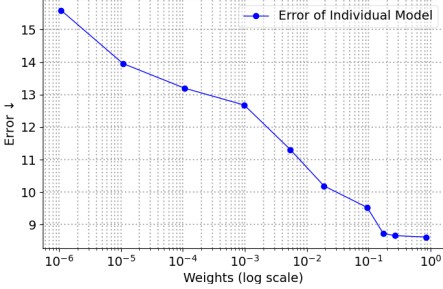
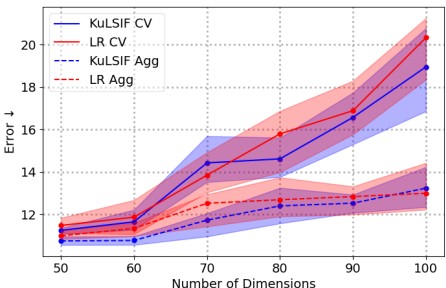

Figure 3: Left: Value of the aggregation weights against twice the Bregman diveregence error on the geometrically constructed dataset "c3,d1.70". More accurate models get assigned higher weights. Right: Our method (dotted lines) scales favorably when increasing the dimensionality of the data. The error is computed by averaging over all geometric datasets, shaded confidence intervals.

### D.3.3 DOMAIN ADAPTATION

The results of the domain adaptation benchmark experiment were computed using gradient-based training across a total of 9,174 models. The implementation of certain components is based on the codebase by Dinu et al. (2023); Gruber et al. (2024). The specifics of the experimental setup are as follows:

- MiniDomainNet (image data): 11 methods × 8 parameters × 5 domain adaptation tasks × 3 seeds = 1320 trained models

- AmazonReviews (text data): 11 methods $\times$ 14 parameters $\times$ 12 domain adaptation tasks $\times$ 3 seeds = 5544 trained models

- HHAR (sensory data): 11 methods $\times$ 14 parameters $\times$ 5 domain adaptation tasks $\times$ 3 seeds = 2310 trained models

Consistent with Dinu et al. (2023), we utilized 11 domain adaptation methods from the AdaTime benchmark (Ragab et al., 2023). For each method and domain adaptation modality (text, image, sensory), we evaluated all 16 density ratio estimation approaches across 22 domain adaptation scenarios. The model implementation for domain adaptation methods and experimental configurations also followed Dinu et al. (2023). Specifically, fully connected networks were used for Amazon Reviews, while a pretrained ResNet-18 backbone was employed for MiniDomainNet. For training and selecting/aggregating the density ratio estimation methods within this pipeline we perform an additional train/val split of $80/20$ on the datasets that are used for training the domain adaption methods. For the regularization parameter $\lambda$ we use $\{10^{-6}, 10^{-5}, \ldots, 10^4\}$ as hyperparameter space and use the same selection/aggregation procedure as for known density ratios. Following Kanamori et al. (2012a), we use a Gaussian kernel with kernel width set according to the median heuristic (Schölkopf & Smola, 2002) for all DRE methods. In the results tables, we report the classification accuracy on the respective test sets of the target distribution for all compared DRE methods.

## D.4 DETAILED EXPERIMENTAL RESULTS

| | Geometric Figures | | | | | | | |
|---|---|---|---|---|---|---|---|---|
| | Bayesian Model Averaging | | | | Super Learner | | | |
| Dataset | KuLSIF | Exp | LR | SQ | KuLSIF | Exp | LR | SQ |
| c3,d1.70 | 8.616($\pm$0.010) | 8.315($\pm$0.008) | 8.840($\pm$0.020) | 9.160($\pm$0.011) | 8.532($\pm$0.006) | 8.257($\pm$0.010) | 8.839($\pm$0.016) | 9.008($\pm$0.011) |
| c2,d1.72 | 13.001($\pm$0.006) | 12.894($\pm$0.013) | 13.255($\pm$0.013) | 13.536($\pm$0.021) | 12.957($\pm$0.008) | 12.633($\pm$0.010) | 13.255($\pm$0.010) | 13.402($\pm$0.020) |
| c2,d1.59 | 12.621($\pm$0.005) | 16.610($\pm$0.032) | 12.828($\pm$0.013) | 13.050($\pm$0.014) | 12.532($\pm$0.009) | 14.597($\pm$0.040) | 12.799($\pm$0.012) | 12.961($\pm$0.013) |
| c1,d1.55 | 11.811($\pm$0.006) | 13.562($\pm$0.023) | 12.001($\pm$0.021) | 12.009($\pm$0.011) | 11.801($\pm$0.005) | 12.915($\pm$0.019) | 12.001($\pm$0.019) | 11.843($\pm$0.010) |
| c2,d1.78 | 9.622($\pm$0.004) | 17.994($\pm$0.029) | 9.801($\pm$0.025) | 9.990($\pm$0.006) | 9.599($\pm$0.012) | 17.579($\pm$0.017) | 9.792($\pm$0.024) | 9.830($\pm$0.007) |
| c2,d1.55 | 10.369($\pm$0.007) | 9.773($\pm$0.015) | 10.555($\pm$0.049) | 10.751($\pm$0.021) | 10.291($\pm$0.003) | 9.632($\pm$0.011) | 10.495($\pm$0.040) | 10.637($\pm$0.019) |
| c3,d1.57 | 12.014($\pm$0.004) | 16.973($\pm$0.012) | 12.204($\pm$0.023) | 13.987($\pm$0.022) | 12.013($\pm$0.006) | 15.021($\pm$0.009) | 11.938($\pm$0.015) | 13.891($\pm$0.021) |
| c2,d1.61 | 11.604($\pm$0.005) | 11.282($\pm$0.029) | 11.795($\pm$0.007) | 12.241($\pm$0.007) | 11.523($\pm$0.004) | 11.008($\pm$0.014) | 11.592($\pm$0.008) | 12.157($\pm$0.007) |
| c3,d1.46 | 12.008($\pm$0.009) | 12.616($\pm$0.007) | 12.961($\pm$0.006) | 13.159($\pm$0.006) | 11.352($\pm$0.005) | 12.521($\pm$0.010) | 12.832($\pm$0.004) | 13.025($\pm$0.005) |
| c1,d1.63 | 9.526($\pm$0.005) | 9.704($\pm$0.009) | 9.731($\pm$0.009) | 9.964($\pm$0.013) | 9.498($\pm$0.007) | 9.699($\pm$0.011) | 9.661 $\pm$ 0.008) | 9.829($\pm$0.010) |
| Avg. | 11.119($\pm$0.006) | 12.972($\pm$0.017) | 11.397($\pm$0.019) | 11.785($\pm$0.013) | 11.009($\pm$0.007) | 12.386($\pm$0.015) | 11.320($\pm$0.016) | 11.658($\pm$0.012) |

Table 4: Mean and standard deviation (after $\pm$) of twice the Bregman divergence error on the geometrically constructed datasets following Kanamori et al. (2012b) over ten different sample draws from $P$ and $Q$.

| | Domain Adaptation: Amazon Reviews | | | | | | | |
|---|---|---|---|---|---|---|---|---|
| | Cross-Validation for Binary Classifier | | | | Aggregation | | | |
| DA-Method | KuLSIF | Exp | LR | SQ | KuLSIF | Exp | LR | SQ |
| MMDA | 0.786(±0.010) | 0.771(±0.009) | 0.784(±0.010) | 0.780(±0.008) | **0.794(±0.008)** | **0.782(±0.008)** | **0.790(±0.010)** | **0.789(±0.006)** |
| CoDATS | 0.795(±0.012) | 0.777(±0.011) | 0.794(±0.012) | 0.788(±0.009) | **0.803(±0.010)** | **0.790(±0.010)** | **0.797(±0.012)** | **0.793(±0.008)** |
| DANN | 0.794(±0.011) | 0.779(±0.011) | 0.793(±0.012) | 0.790(±0.010) | **0.804(±0.009)** | **0.792(±0.010)** | **0.800(±0.012)** | **0.799(±0.008)** |
| CDAN | 0.787(±0.012) | 0.772(±0.010) | 0.787(±0.010) | 0.785(±0.009) | **0.792(±0.010)** | **0.783(±0.009)** | **0.790(±0.010)** | **0.792(±0.007)** |
| DSAN | 0.794(±0.011) | 0.778(±0.012) | 0.793(±0.013) | 0.790(±0.011) | **0.803(±0.009)** | **0.790(±0.011)** | **0.798(±0.013)** | **0.798(±0.009)** |
| DIRT | 0.787(±0.011) | 0.776(±0.011) | 0.788(±0.012) | 0.784(±0.010) | **0.797(±0.009)** | **0.787(±0.010)** | **0.790(±0.012)** | **0.794(±0.008)** |
| AdvSKM | 0.779(±0.011) | 0.761(±0.009) | 0.777(±0.010) | 0.772(±0.008) | **0.784(±0.009)** | **0.774(±0.008)** | **0.781(±0.010)** | **0.782(±0.006)** |
| HoMM | 0.777(±0.009) | 0.760(±0.010) | 0.774(±0.011) | 0.770(±0.009) | **0.786(±0.007)** | **0.772(±0.009)** | **0.779(±0.011)** | **0.779(±0.008)** |
| DDC | 0.780(±0.011) | 0.764(±0.011) | 0.778(±0.012) | 0.774(±0.010) | **0.787(±0.009)** | **0.777(±0.010)** | **0.782(±0.012)** | **0.783(±0.008)** |
| DeepCoral | 0.784(±0.010) | 0.767(±0.010) | 0.781(±0.011) | 0.776(±0.009) | **0.794(±0.008)** | **0.779(±0.009)** | **0.785(±0.011)** | **0.784(±0.007)** |
| CMD | 0.790(±0.013) | 0.773(±0.011) | 0.786(±0.012) | 0.781(±0.011) | **0.797(±0.011)** | **0.783(±0.011)** | **0.790(±0.012)** | **0.787(±0.008)** |
| Avg. | 0.787(±0.011) | 0.771(±0.010) | 0.785(±0.011) | 0.781(±0.009) | **0.795(±0.009)** | **0.783(±0.009)** | **0.789(±0.011)** | **0.789(±0.007)** |

| | Domain Adaptation: HHAR | | | | | | | |
|---|---|---|---|---|---|---|---|---|
| | Cross-Validation for Binary Classifier | | | | Aggregation | | | |
| DA-Method | KuLSIF | Exp | LR | SQ | KuLSIF | Exp | LR | SQ |
| MMDA | 0.780(±0.008) | 0.670(±0.098) | 0.773(±0.013) | 0.736(±0.006) | **0.826(±0.023)** | **0.711(±0.085)** | **0.786(±0.007)** | **0.763(±0.005)** |
| CoDATS | 0.723(±0.130) | 0.776(±0.021) | 0.779(±0.021) | 0.741(±0.011) | **0.765(±0.120)** | **0.818(±0.016)** | **0.793(±0.015)** | **0.767(±0.010)** |
| DANN | 0.697(±0.179) | 0.785(±0.024) | 0.795(±0.021) | 0.757(±0.011) | **0.738(±0.170)** | **0.828(±0.015)** | **0.807(±0.015)** | **0.780(±0.010)** |
| CDAN | 0.792(±0.029) | 0.706(±0.166) | 0.788(±0.029) | 0.751(±0.022) | **0.834(±0.031)** | **0.750(±0.144)** | **0.804(±0.024)** | **0.775(±0.021)** |
| DSAN | 0.754(±0.128) | 0.528(±0.228) | 0.792(±0.015) | 0.754(±0.014) | **0.794(±0.133)** | **0.572(±0.219)** | **0.805(±0.013)** | **0.775(±0.013)** |
| DIRT | 0.731(±0.084) | 0.790(±0.040) | 0.790(±0.023) | 0.753(±0.014) | **0.774(±0.086)** | **0.828(±0.023)** | **0.801(±0.017)** | **0.779(±0.013)** |
| AdvSKM | 0.752(±0.007) | 0.746(±0.013) | 0.746(±0.018) | 0.708(±0.008) | **0.793(±0.024)** | **0.790(±0.019)** | **0.759(±0.012)** | **0.735(±0.006)** |
| HoMM | 0.759(±0.009) | 0.662(±0.096) | 0.754(±0.021) | 0.716(±0.011) | **0.803(±0.022)** | **0.707(±0.083)** | **0.768(±0.016)** | **0.749(±0.010)** |
| DDC | 0.748(±0.017) | 0.454(±0.215) | 0.744(±0.019) | 0.706(±0.009) | **0.794(±0.018)** | **0.496(±0.196)** | **0.759(±0.013)** | **0.732(±0.009)** |
| DeepCoral | 0.701(±0.115) | 0.749(±0.021) | 0.758(±0.015) | 0.720(±0.006) | **0.745(±0.118)** | **0.791(±0.013)** | **0.769(±0.011)** | **0.743(±0.005)** |
| CMD | 0.671(±0.170) | 0.770(±0.015) | 0.770(±0.016) | 0.732(±0.008) | **0.714(±0.164)** | **0.814(±0.017)** | **0.780(±0.010)** | **0.754(±0.007)** |
| Avg. | 0.737(±0.080) | 0.694(±0.085) | 0.772(±0.019) | 0.734(±0.011) | **0.780(±0.082)** | **0.737(±0.075)** | **0.785(±0.014)** | **0.759(±0.010)** |

Table 5: Mean and standard deviation (after ±) of target classification accuracy on Amazon Reviews and HHAR datasets over three different random initialization of model weights and several domain adaptation scenarios.

| | Domain Adaptation: MiniDomainNet | | | | | | | |
|---|---|---|---|---|---|---|---|---|
| | Bayesian Model Averaging | | | | Super Learner | | | |
| DA-Method | Kul | Exp | Log | SQ | Kul | Exp | Log | SQ |
| MMDA | 0.525(±0.005) | 0.529(±0.009) | 0.528(±0.010) | 0.512(±0.010) | 0.527(±0.007) | 0.530(±0.009) | 0.529(±0.006) | 0.516(±0.008) |
| CoDATS | 0.537(±0.009) | 0.533(±0.013) | 0.530(±0.018) | 0.514(±0.018) | 0.539(±0.011) | 0.534(±0.013) | 0.531(±0.014) | 0.518(±0.016) |
| DANN | 0.537(±0.003) | 0.522(±0.010) | 0.520(±0.016) | 0.503(±0.017) | 0.539(±0.005) | 0.522(±0.010) | 0.520(±0.012) | 0.507(±0.014) |
| CDAN | 0.531(±0.010) | 0.531(±0.015) | 0.524(±0.021) | 0.505(±0.021) | 0.533(±0.012) | 0.533(±0.016) | 0.524(±0.017) | 0.509(±0.019) |
| DSAN | 0.544(±0.008) | 0.532(±0.013) | 0.527(±0.013) | 0.513(±0.013) | 0.546(±0.010) | 0.533(±0.013) | 0.529(±0.009) | 0.517(±0.011) |
| DIRT | 0.513(±0.026) | 0.387(±0.175) | 0.520(±0.021) | 0.503(±0.021) | 0.515(±0.028) | 0.389(±0.175) | 0.522(±0.017) | 0.507(±0.019) |
| AdvSKM | 0.514(±0.005) | 0.516(±0.007) | 0.512(±0.009) | 0.500(±0.010) | 0.516(±0.007) | 0.514(±0.007) | 0.513(±0.006) | 0.504(±0.008) |
| HoMM | 0.529(±0.004) | 0.531(±0.010) | 0.521(±0.013) | 0.506(±0.013) | 0.531(±0.006) | 0.530(±0.010) | 0.522(±0.009) | 0.510(±0.011) |
| DDC | 0.514(±0.007) | 0.517(±0.011) | 0.514(±0.010) | 0.501(±0.010) | 0.516(±0.009) | 0.519(±0.011) | 0.516(±0.008) | 0.505(±0.009) |
| DeepCoral | 0.537(±0.008) | 0.530(±0.011) | 0.530(±0.010) | 0.513(±0.010) | 0.539(±0.010) | 0.531(±0.011) | 0.531(±0.007) | 0.517(±0.008) |
| CMD | 0.531(±0.004) | 0.525(±0.019) | 0.521(±0.019) | 0.504(±0.018) | 0.533(±0.006) | 0.525(±0.019) | 0.522(±0.015) | 0.508(±0.016) |
| Avg. | 0.528(±0.008) | 0.514(±0.027) | 0.523(±0.015) | 0.507(±0.015) | 0.530(±0.010) | 0.514(±0.027) | 0.523(±0.011) | 0.511(±0.013) |

Table 6: Mean and standard deviation (after ±) of target classification accuracy on MiniDomainNet over three different random initialization of model weights and several domain adaptation scenarios.

| | Domain Adaptation: Amazon Reviews | | | | | | | |
|---|---|---|---|---|---|---|---|---|
| | Bayesian Model Averaging | | | | Super Learner | | | |
| DA-Method | Kul | Exp | Log | SQ | Kul | Exp | Log | SQ |
| MMDA | 0.786(±0.009) | 0.772(±0.009) | 0.784(±0.010) | 0.781(±0.007) | 0.789(±0.008) | 0.773(±0.008) | 0.784(±0.009) | 0.782(±0.006) |
| CoDATS | 0.796(±0.012) | 0.781(±0.011) | 0.794(±0.012) | 0.793(±0.009) | 0.800(±0.011) | 0.782(±0.010) | 0.794(±0.011) | 0.792(±0.008) |
| DANN | 0.789(±0.014) | 0.781(±0.011) | 0.793(±0.012) | 0.790(±0.009) | 0.792(±0.012) | 0.782(±0.010) | 0.793(±0.011) | 0.791(±0.009) |
| CDAN | 0.787(±0.016) | 0.775(±0.010) | 0.787(±0.010) | 0.784(±0.008) | 0.787(±0.014) | 0.776(±0.009) | 0.787(±0.009) | 0.784(±0.007) |
| DSAN | 0.794(±0.012) | 0.780(±0.011) | 0.793(±0.013) | 0.788(±0.010) | 0.794(±0.012) | 0.781(±0.010) | 0.793(±0.012) | 0.789(±0.009) |
| DIRT | 0.789(±0.011) | 0.775(±0.011) | 0.788(±0.012) | 0.782(±0.009) | 0.791(±0.010) | 0.776(±0.010) | 0.788(±0.011) | 0.784(±0.008) |
| AdvSKM | 0.780(±0.009) | 0.763(±0.010) | 0.777(±0.010) | 0.772(±0.007) | 0.781(±0.007) | 0.764(±0.008) | 0.777(±0.009) | 0.775(±0.006) |
| HoMM | 0.780(±0.007) | 0.761(±0.010) | 0.774(±0.011) | 0.769(±0.009) | 0.779(±0.005) | 0.762(±0.009) | 0.775(±0.010) | 0.771(±0.008) |
| DDC | 0.781(±0.008) | 0.767(±0.010) | 0.778(±0.012) | 0.771(±0.009) | 0.782(±0.007) | 0.768(±0.009) | 0.778(±0.011) | 0.773(±0.008) |
| DeepCoral | 0.787(±0.008) | 0.766(±0.010) | 0.781(±0.011) | 0.776(±0.008) | 0.788(±0.006) | 0.767(±0.010) | 0.782(±0.010) | 0.778(±0.008) |
| CMD | 0.784(±0.020) | 0.772(±0.012) | 0.786(±0.012) | 0.782(±0.009) | 0.786(±0.018) | 0.773(±0.011) | 0.787(±0.011) | 0.784(±0.008) |
| Avg. | 0.787(±0.012) | 0.772(±0.010) | 0.785(±0.011) | 0.781(±0.009) | 0.788(±0.010) | 0.773(±0.009) | 0.785(±0.010) | 0.782(±0.008) |

Table 7: Mean and standard deviation (after ±) of target classification accuracy on Amazon Reviews over three different random initialization of model weights and several domain adaptation scenarios.

| | Domain Adaptation: HHAR | | | | | | | |
|---|---|---|---|---|---|---|---|---|
| | Bayesian Model Averaging | | | | Super Learner | | | |
| DA-Method | Kul | Exp | Log | SQ | Kul | Exp | Log | SQ |
| MMDA | 0.781(±0.004) | 0.685(±0.098) | 0.773(±0.012) | 0.741(±0.008) | 0.784(±0.004) | 0.696(±0.088) | 0.773(±0.008) | 0.747(±0.005) |
| CoDATS | 0.724(±0.126) | 0.790(±0.020) | 0.779(±0.020) | 0.744(±0.013) | 0.723(±0.125) | 0.803(±0.011) | 0.779(±0.015) | 0.751(±0.010) |
| DANN | 0.697(±0.175) | 0.802(±0.023) | 0.795(±0.020) | 0.761(±0.013) | 0.701(±0.175) | 0.809(±0.017) | 0.797(±0.016) | 0.765(±0.010) |
| CDAN | 0.793(±0.029) | 0.724(±0.166) | 0.788(±0.028) | 0.752(±0.024) | 0.795(±0.028) | 0.735(±0.157) | 0.791(±0.024) | 0.763(±0.021) |
| DSAN | 0.755(±0.127) | 0.545(±0.227) | 0.792(±0.014) | 0.758(±0.016) | 0.759(±0.127) | 0.555(±0.224) | 0.795(±0.013) | 0.763(±0.013) |
| DIRT | 0.732(±0.081) | 0.806(±0.039) | 0.790(±0.022) | 0.753(±0.016) | 0.734(±0.081) | 0.817(±0.030) | 0.791(±0.018) | 0.759(±0.013) |
| AdvSKM | 0.752(±0.004) | 0.762(±0.012) | 0.746(±0.017) | 0.713(±0.010) | 0.757(±0.003) | 0.771(±0.005) | 0.748(±0.013) | 0.717(±0.007) |
| HoMM | 0.759(±0.005) | 0.679(±0.095) | 0.754(±0.020) | 0.723(±0.013) | 0.761(±0.005) | 0.690(±0.086) | 0.755(±0.017) | 0.721(±0.010) |
| DDC | 0.749(±0.014) | 0.471(±0.215) | 0.744(±0.018) | 0.710(±0.011) | 0.752(±0.012) | 0.478(±0.209) | 0.746(±0.014) | 0.713(±0.009) |
| DeepCoral | 0.701(±0.113) | 0.768(±0.020) | 0.758(±0.014) | 0.727(±0.008) | 0.705(±0.112) | 0.779(±0.014) | 0.758(±0.012) | 0.730(±0.006) |
| CMD | 0.671(±0.166) | 0.785(±0.014) | 0.770(±0.016) | 0.736(±0.010) | 0.674(±0.165) | 0.797(±0.005) | 0.772(±0.011) | 0.742(±0.007) |
| Avg. | 0.738(±0.077) | 0.711(±0.084) | 0.772(±0.018) | 0.738(±0.013) | 0.740(±0.076) | 0.721(±0.077) | 0.773(±0.015) | 0.743(±0.010) |

Table 8: Mean and standard deviation (after ±) of target classification accuracy on HHAR over three different random initialization of model weights and several domain adaptation scenarios.

| | Cross-Validation for Binary Classifier | | | | Aggregation | | | |
|---|---|---|---|---|---|---|---|---|
| Scenario | KuLSIF | Exp | LR | SQ | KuLSIF | Exp | LR | SQ |
| R → C | 0.585(±0.013) | 0.590(±0.016) | 0.589(±0.024) | 0.578(±0.022) | **0.595(±0.011)** | 0.598(±0.015) | 0.596(±0.017) | 0.592(±0.021) |
| R → I | 0.377(±0.009) | 0.372(±0.008) | 0.378(±0.007) | 0.365(±0.005) | **0.393(±0.007)** | 0.385(±0.008) | 0.394(±0.001) | 0.384(±0.004) |
| R → P | 0.722(±0.004) | 0.729(±0.006) | **0.725(±0.006)** | 0.723(±0.004) | **0.730(±0.002)** | 0.743(±0.006) | 0.721(±0.000) | 0.728(±0.003) |
| R → Q | 0.354(±0.008) | 0.348(±0.016) | 0.349(±0.014) | 0.328(±0.013) | **0.356(±0.006)** | 0.356(±0.016) | 0.358(±0.008) | 0.351(±0.012) |
| R → S | 0.596(±0.009) | 0.599(±0.010) | 0.600(±0.009) | 0.596(±0.007) | 0.606(±0.007) | 0.611(±0.009) | 0.610(±0.002) | 0.618(±0.006) |
| Avg. | 0.527(±0.009) | 0.528(±0.011) | 0.528(±0.012) | 0.518(±0.010) | **0.536(±0.007)** | 0.539(±0.011) | 0.536(±0.006) | 0.535(±0.009) |

Table 9: Mean and standard deviation (after ±) of target classification accuracy on MiniDomainNet computed with domain adaptation method MMDA.

| | Cross-Validation for Binary Classifier | | | | Aggregation | | | |
|---|---|---|---|---|---|---|---|---|
| Scenario | KuLSIF | Exp | LR | SQ | KuLSIF | Exp | LR | SQ |
| R → C | 0.597(±0.011) | 0.577(±0.010) | 0.594(±0.010) | 0.576(±0.008) | **0.602(±0.009)** | 0.587(±0.008) | 0.601(±0.005) | 0.593(±0.006) |
| R → I | 0.366(±0.012) | 0.368(±0.023) | 0.349(±0.038) | 0.338(±0.036) | **0.373(±0.011)** | 0.385(±0.023) | 0.357(±0.031) | 0.356(±0.035) |
| R → P | 0.739(±0.015) | 0.733(±0.018) | 0.737(±0.017) | 0.726(±0.015) | **0.747(±0.014)** | 0.745(±0.017) | 0.750(±0.011) | 0.742(±0.014) |
| R → Q | 0.361(±0.013) | 0.364(±0.014) | 0.355(±0.027) | 0.342(±0.025) | **0.370(±0.010)** | 0.377(±0.013) | 0.369(±0.021) | 0.364(±0.023) |
| R → S | **0.619(±0.008)** | 0.616(±0.008) | 0.616(±0.009) | 0.601(±0.008) | 0.617(±0.006) | 0.624(±0.008) | 0.622(±0.002) | 0.607(±0.008) |
| Avg. | 0.536(±0.012) | 0.532(±0.015) | 0.530(±0.020) | 0.517(±0.018) | **0.542(±0.010)** | 0.543(±0.014) | 0.540(±0.014) | 0.533(±0.017) |

Table 10: Mean and standard deviation (after ±) of target classification accuracy on MiniDomainNet computed with domain adaptation method CoDATS.

| | Cross-Validation for Binary Classifier | | | | Aggregation | | | |
|---|---|---|---|---|---|---|---|---|
| Scenario | KuLSIF | Exp | LR | SQ | KuLSIF | Exp | LR | SQ |
| R → C | 0.591(±0.015) | 0.576(±0.013) | 0.579(±0.012) | 0.560(±0.009) | **0.597(±0.013)** | 0.583(±0.011) | 0.587(±0.005) | 0.568(±0.007) |
| R → I | 0.374(±0.009) | 0.372(±0.011) | 0.369(±0.026) | 0.354(±0.024) | **0.379(±0.007)** | 0.387(±0.009) | 0.375(±0.021) | 0.368(±0.022) |
| R → P | 0.720(±0.007) | 0.716(±0.008) | 0.699(±0.032) | 0.692(±0.030) | **0.724(±0.006)** | 0.727(±0.008) | 0.702(±0.026) | 0.699(±0.029) |
| R → Q | 0.357(±0.009) | 0.340(±0.016) | 0.337(±0.016) | 0.322(±0.014) | **0.362(±0.005)** | 0.355(±0.015) | 0.348(±0.009) | 0.326(±0.013) |
| R → S | 0.613(±0.007) | 0.607(±0.010) | 0.614(±0.007) | 0.602(±0.005) | **0.619(±0.005)** | 0.624(±0.010) | 0.621(±0.001) | 0.632(±0.004) |
| Avg. | 0.531(±0.009) | 0.522(±0.012) | 0.520(±0.019) | 0.506(±0.016) | **0.536(±0.007)** | 0.535(±0.011) | 0.526(±0.013) | 0.519(±0.015) |

Table 11: Mean and standard deviation (after ±) of target classification accuracy on MiniDomainNet computed with domain adaptation method DANN.

| | Cross-Validation for Binary Classifier | | | | Aggregation | | | |
|---|---|---|---|---|---|---|---|---|
| Scenario | KuLSIF | Exp | LR | SQ | KuLSIF | Exp | LR | SQ |
| R → C | 0.592(±0.018) | 0.595(±0.029) | 0.594(±0.023) | 0.578(±0.021) | 0.594(±0.015) | 0.616(±0.028) | 0.608(±0.017) | 0.598(±0.019) |
| R → I | 0.372(±0.013) | 0.381(±0.012) | 0.337(±0.056) | 0.324(±0.053) | **0.384(±0.011)** | 0.384(±0.011) | 0.346(±0.050) | 0.343(±0.052) |
| R → P | 0.729(±0.004) | 0.720(±0.015) | 0.725(±0.011) | 0.713(±0.009) | **0.735(±0.001)** | 0.734(±0.014) | 0.731(±0.005) | 0.722(±0.008) |
| R → Q | 0.353(±0.018) | 0.352(±0.020) | 0.354(±0.017) | 0.351(±0.014) | 0.354(±0.014) | 0.369(±0.020) | 0.364(±0.011) | 0.363(±0.012) |
| R → S | 0.609(±0.008) | 0.607(±0.012) | 0.611(±0.009) | 0.594(±0.007) | **0.618(±0.005)** | 0.618(±0.010) | 0.623(±0.003) | 0.603(±0.006) |
| Avg. | 0.531(±0.012) | 0.531(±0.017) | 0.524(±0.023) | 0.512(±0.021) | **0.537(±0.009)** | 0.544(±0.017) | 0.535(±0.017) | 0.526(±0.020) |

Table 12: Mean and standard deviation (after ±) of target classification accuracy on MiniDomainNet computed with domain adaptation method CDAN.

| Scenario | Cross-Validation for Binary Classifier | | | | Aggregation | | | |
|---|---|---|---|---|---|---|---|---|
| | KuLSIF | Exp | LR | SQ | KuLSIF | Exp | LR | SQ |
| R → C | **0.623**(±**0.015**) | 0.603(±0.026) | 0.596(±0.019) | 0.589(±0.017) | 0.622(±0.013) | 0.614(±0.026) | 0.606(±0.013) | 0.597(±0.016) |
| R → I | 0.374(±0.010) | 0.367(±0.008) | 0.350(±0.014) | 0.339(±0.012) | **0.383**(±**0.008**) | **0.391**(±**0.006**) | **0.363**(±**0.010**) | **0.344**(±**0.011**) |
| R → P | 0.720(±0.007) | 0.724(±0.008) | 0.720(±0.008) | 0.708(±0.005) | **0.726**(±**0.004**) | **0.738**(±**0.007**) | **0.729**(±**0.001**) | **0.723**(±**0.004**) |
| R → Q | 0.364(±0.012) | 0.355(±0.018) | 0.354(±0.018) | 0.334(±0.016) | **0.366**(±**0.010**) | **0.364**(±**0.016**) | **0.359**(±**0.012**) | **0.347**(±**0.015**) |
| R → S | 0.616(±0.012) | 0.609(±0.015) | 0.615(±0.016) | 0.598(±0.014) | **0.621**(±**0.011**) | **0.622**(±**0.014**) | **0.618**(±**0.010**) | **0.612**(±**0.013**) |
| Avg. | 0.539(±0.011) | 0.532(±0.015) | 0.527(±0.015) | 0.513(±0.013) | **0.544**(±**0.009**) | **0.546**(±**0.014**) | **0.535**(±**0.009**) | **0.525**(±**0.012**) |

Table 13: Mean and standard deviation (after ±) of target classification accuracy on MiniDomainNet computed with domain adaptation method DSAN.

| Scenario | Cross-Validation for Binary Classifier | | | | Aggregation | | | |
|---|---|---|---|---|---|---|---|---|
| | KuLSIF | Exp | LR | SQ | KuLSIF | Exp | LR | SQ |
| R → C | 0.572(±0.021) | 0.474(±0.176) | 0.566(±0.022) | 0.555(±0.020) | **0.580**(±**0.019**) | 0.478(±0.175) | **0.582**(±**0.016**) | **0.575**(±**0.018**) |
| R → I | 0.376(±0.047) | 0.236(±0.135) | 0.406(±0.025) | 0.391(±0.022) | **0.387**(±**0.045**) | 0.246(±0.133) | **0.407**(±**0.018**) | **0.402**(±**0.022**) |
| R → P | 0.714(±0.026) | 0.542(±0.271) | **0.713**(±**0.025**) | 0.700(±0.024) | **0.718**(±**0.024**) | 0.550(±0.270) | 0.711(±0.020) | **0.724**(±**0.023**) |
| R → Q | 0.336(±0.018) | 0.248(±0.055) | 0.329(±0.021) | 0.315(±0.019) | **0.340**(±**0.016**) | 0.258(±0.054) | **0.338**(±**0.016**) | **0.325**(±**0.018**) |
| R → S | 0.586(±0.020) | 0.431(±0.245) | 0.587(±0.021) | 0.585(±0.019) | **0.591**(±**0.019**) | 0.441(±0.244) | **0.595**(±**0.015**) | **0.598**(±**0.018**) |
| Avg. | 0.517(±0.026) | 0.386(±0.177) | 0.520(±0.023) | 0.509(±0.021) | **0.523**(±**0.025**) | 0.395(±0.175) | **0.526**(±**0.017**) | **0.525**(±**0.020**) |

Table 14: Mean and standard deviation (after ±) of target classification accuracy on MiniDomainNet computed with domain adaptation method DIRT.

| Scenario | Cross-Validation for Binary Classifier | | | | Aggregation | | | |
|---|---|---|---|---|---|---|---|---|
| | KuLSIF | Exp | LR | SQ | KuLSIF | Exp | LR | SQ |
| R → C | 0.568(±0.008) | 0.561(±0.013) | 0.552(±0.006) | 0.535(±0.004) | **0.580**(±**0.006**) | **0.573**(±**0.012**) | **0.564**(±**0.000**) | **0.548**(±**0.003**) |
| R → I | **0.382**(±**0.010**) | 0.389(±0.007) | **0.385**(±**0.026**) | 0.369(±0.024) | 0.381(±0.008) | **0.415**(±**0.006**) | 0.384(±0.019) | **0.384**(±**0.023**) |
| R → P | 0.717(±0.005) | 0.712(±0.014) | 0.712(±0.010) | 0.705(±0.008) | 0.717(±0.004) | **0.725**(±**0.013**) | **0.724**(±**0.004**) | **0.723**(±**0.006**) |
| R → Q | 0.332(±0.005) | 0.325(±0.007) | 0.324(±0.011) | 0.315(±0.008) | **0.342**(±**0.003**) | **0.329**(±**0.006**) | **0.331**(±**0.004**) | **0.332**(±**0.007**) |
| R → S | 0.584(±0.004) | 0.586(±0.003) | 0.588(±0.005) | 0.575(±0.004) | **0.587**(±**0.003**) | **0.602**(±**0.002**) | **0.602**(±**0.001**) | **0.598**(±**0.002**) |
| Avg. | 0.516(±0.006) | 0.515(±0.009) | 0.512(±0.012) | 0.500(±0.010) | **0.522**(±**0.005**) | **0.529**(±**0.008**) | **0.521**(±**0.006**) | **0.517**(±**0.008**) |

Table 15: Mean and standard deviation (after ±) of target classification accuracy on MiniDomainNet computed with domain adaptation method AdvSKM.

| Scenario | Cross-Validation for Binary Classifier | | | | Aggregation | | | |
|---|---|---|---|---|---|---|---|---|
| | KuLSIF | Exp | LR | SQ | KuLSIF | Exp | LR | SQ |
| R → C | 0.590(±0.020) | 0.588(±0.020) | 0.577(±0.020) | 0.567(±0.018) | **0.604**(±**0.018**) | **0.604**(±**0.019**) | **0.589**(±**0.013**) | **0.591**(±**0.017**) |
| R → I | 0.399(±0.006) | 0.407(±0.009) | 0.378(±0.025) | 0.365(±0.024) | **0.400**(±**0.004**) | **0.419**(±**0.008**) | **0.386**(±**0.018**) | **0.385**(±**0.023**) |
| R → P | 0.729(±0.002) | 0.726(±0.009) | 0.726(±0.010) | 0.704(±0.008) | **0.736**(±**0.000**) | **0.744**(±**0.008**) | **0.737**(±**0.004**) | **0.725**(±**0.007**) |
| R → Q | 0.351(±0.005) | 0.333(±0.013) | 0.333(±0.013) | 0.318(±0.012) | **0.356**(±**0.002**) | **0.345**(±**0.011**) | **0.341**(±**0.008**) | **0.345**(±**0.011**) |
| R → S | 0.588(±0.008) | 0.593(±0.011) | 0.592(±0.008) | 0.572(±0.005) | **0.599**(±**0.007**) | **0.610**(±**0.009**) | **0.594**(±**0.002**) | **0.595**(±**0.003**) |
| Avg. | 0.531(±0.008) | 0.529(±0.012) | 0.521(±0.015) | 0.505(±0.013) | **0.539**(±**0.007**) | **0.544**(±**0.011**) | **0.529**(±**0.009**) | **0.528**(±**0.012**) |

Table 16: Mean and standard deviation (after ±) of target classification accuracy on MiniDomainNet computed with domain adaptation method HoMM.

| Scenario | Cross-Validation for Binary Classifier | | | | Aggregation | | | |
|---|---|---|---|---|---|---|---|---|
| | KuLSIF | Exp | LR | SQ | KuLSIF | Exp | LR | SQ |
| R → C | 0.569(±0.027) | 0.571(±0.029) | 0.563(±0.028) | 0.555(±0.027) | **0.584**(±**0.025**) | **0.590**(±**0.029**) | **0.577**(±**0.023**) | **0.573**(±**0.027**) |
| R → I | 0.389(±0.011) | 0.391(±0.016) | 0.390(±0.011) | 0.369(±0.010) | **0.394**(±**0.009**) | **0.399**(±**0.015**) | **0.398**(±**0.006**) | **0.388**(±**0.009**) |
| R → P | 0.717(±0.002) | 0.721(±0.003) | 0.715(±0.002) | 0.701(±0.000) | **0.727**(±**0.000**) | **0.726**(±**0.002**) | **0.723**(±**0.003**) | **0.711**(±**0.000**) |
| R → Q | 0.333(±0.004) | 0.323(±0.013) | 0.324(±0.013) | 0.313(±0.010) | **0.337**(±**0.002**) | **0.335**(±**0.012**) | **0.336**(±**0.006**) | **0.331**(±**0.009**) |
| R → S | 0.580(±0.005) | 0.578(±0.005) | 0.579(±0.004) | 0.562(±0.003) | **0.594**(±**0.003**) | **0.595**(±**0.004**) | **0.586**(±**0.002**) | **0.574**(±**0.002**) |
| Avg. | 0.517(±0.010) | 0.517(±0.013) | 0.514(±0.012) | 0.500(±0.010) | **0.527**(±**0.008**) | **0.529**(±**0.012**) | **0.524**(±**0.008**) | **0.515**(±**0.009**) |

Table 17: Mean and standard deviation (after ±) of target classification accuracy on MiniDomainNet computed with domain adaptation method DDC.

| Scenario | Cross-Validation for Binary Classifier | | | | Aggregation | | | |
|---|---|---|---|---|---|---|---|---|
| | KuLSIF | Exp | LR | SQ | KuLSIF | Exp | LR | SQ |
| R → C | 0.588(±0.021) | 0.569(±0.016) | 0.567(±0.012) | 0.543(±0.010) | **0.593**(±**0.020**) | **0.578**(±**0.015**) | **0.578**(±**0.006**) | **0.548**(±**0.009**) |
| R → I | 0.369(±0.012) | 0.372(±0.008) | **0.381**(±**0.018**) | 0.370(±0.017) | **0.371**(±**0.010**) | **0.387**(±**0.008**) | 0.380(±0.013) | **0.379**(±**0.016**) |
| R → P | 0.735(±0.011) | 0.737(±0.010) | 0.734(±0.005) | 0.722(±0.004) | **0.749**(±**0.010**) | **0.749**(±**0.010**) | **0.737**(±**0.001**) | **0.732**(±**0.003**) |
| R → Q | 0.363(±0.011) | 0.348(±0.021) | 0.349(±0.016) | 0.334(±0.014) | 0.363(±0.008) | **0.359**(±**0.020**) | **0.365**(±**0.011**) | **0.355**(±**0.013**) |
| R → S | 0.621(±0.006) | 0.616(±0.009) | 0.617(±0.009) | 0.604(±0.008) | **0.637**(±**0.003**) | **0.629**(±**0.008**) | **0.619**(±**0.003**) | **0.620**(±**0.006**) |
| Avg. | 0.535(±0.012) | 0.528(±0.013) | 0.530(±0.012) | 0.514(±0.011) | **0.543**(±**0.010**) | **0.540**(±**0.012**) | **0.536**(±**0.007**) | **0.527**(±**0.009**) |

Table 18: Mean and standard deviation (after ±) of target classification accuracy on MiniDomainNet computed with domain adaptation method DeepCoral.

| Scenario | Cross-Validation for Binary Classifier | | | | Aggregation | | | |
|---|---|---|---|---|---|---|---|---|
| | KuLSIF | Exp | LR | SQ | KuLSIF | Exp | LR | SQ |
| R → C | 0.576(±0.015) | 0.560(±0.026) | 0.552(±0.017) | 0.538(±0.015) | **0.585(±0.013)** | 0.574(±0.025) | 0.560(±0.011) | 0.541(±0.014) |
| R → I | 0.383(±0.009) | 0.391(±0.021) | 0.381(±0.033) | 0.375(±0.032) | **0.386(±0.008)** | 0.400(±0.019) | 0.384(±0.026) | 0.397(±0.031) |
| R → P | 0.739(±0.008) | 0.739(±0.016) | 0.742(±0.011) | 0.723(±0.008) | **0.753(±0.006)** | 0.747(±0.016) | 0.747(±0.005) | 0.732(±0.006) |
| R → Q | 0.354(±0.013) | 0.336(±0.030) | 0.335(±0.028) | 0.321(±0.026) | **0.362(±0.011)** | 0.352(±0.029) | 0.341(±0.022) | 0.333(±0.025) |
| R → S | **0.595(±0.005)** | 0.594(±0.010) | 0.597(±0.014) | 0.590(±0.013) | 0.594(±0.004) | 0.617(±0.010) | 0.603(±0.009) | 0.605(±0.013) |
| Avg. | 0.529(±0.010) | 0.524(±0.021) | 0.521(±0.021) | 0.510(±0.019) | **0.536(±0.008)** | 0.538(±0.020) | 0.527(±0.015) | 0.521(±0.018) |

Table 19: Mean and standard deviation (after ±) of target classification accuracy on MiniDomainNet computed with domain adaptation method CMD.

| Scenario | Bayesian Model Averaging | | | | Super Learner | | | |
|---|---|---|---|---|---|---|---|---|
| | KuLSIF | Exp | LR | SQ | KuLSIF | Exp | LR | SQ |
| R → C | 0.583(±0.008) | 0.589(±0.015) | 0.589(±0.022) | 0.565(±0.023) | 0.585(±0.009) | 0.588(±0.014) | 0.588(±0.019) | 0.569(±0.021) |
| R → I | 0.371(±0.001) | 0.374(±0.006) | 0.378(±0.006) | 0.365(±0.004) | 0.373(±0.003) | 0.376(±0.006) | 0.377(±0.001) | 0.369(±0.003) |
| R → P | 0.721(±0.003) | 0.731(±0.005) | 0.725(±0.004) | 0.712(±0.004) | 0.723(±0.004) | 0.733(±0.003) | 0.727(±0.001) | 0.716(±0.002) |
| R → Q | 0.360(±0.007) | 0.350(±0.014) | 0.349(±0.011) | 0.338(±0.012) | 0.362(±0.009) | 0.348(±0.013) | 0.351(±0.008) | 0.342(±0.010) |
| R → S | 0.591(±0.007) | 0.601(±0.007) | 0.600(±0.006) | 0.580(±0.007) | 0.593(±0.009) | 0.603(±0.008) | 0.602(±0.002) | 0.584(±0.004) |
| Avg. | 0.525(±0.005) | 0.529(±0.009) | 0.528(±0.010) | 0.512(±0.010) | 0.527(±0.007) | 0.530(±0.009) | 0.529(±0.006) | 0.516(±0.008) |

Table 20: Mean and standard deviation (after ±) of target classification accuracy on MiniDomainNet computed with domain adaptation method MMDA.

| Scenario | Bayesian Model Averaging | | | | Super Learner | | | |
|---|---|---|---|---|---|---|---|---|
| | KuLSIF | Exp | LR | SQ | KuLSIF | Exp | LR | SQ |
| R → C | 0.602(±0.010) | 0.576(±0.009) | 0.594(±0.008) | 0.572(±0.008) | 0.604(±0.013) | 0.577(±0.008) | 0.593(±0.004) | 0.576(±0.006) |
| R → I | 0.358(±0.003) | 0.370(±0.021) | 0.349(±0.035) | 0.332(±0.036) | 0.360(±0.005) | 0.366(±0.021) | 0.348(±0.031) | 0.336(±0.033) |
| R → P | 0.741(±0.015) | 0.735(±0.016) | 0.737(±0.015) | 0.723(±0.015) | 0.743(±0.016) | 0.737(±0.016) | 0.739(±0.011) | 0.727(±0.013) |
| R → Q | 0.364(±0.008) | 0.363(±0.011) | 0.355(±0.026) | 0.342(±0.025) | 0.366(±0.010) | 0.368(±0.012) | 0.354(±0.020) | 0.346(±0.023) |
| R → S | 0.620(±0.007) | 0.618(±0.006) | 0.616(±0.006) | 0.598(±0.006) | 0.622(±0.009) | 0.620(±0.007) | 0.618(±0.004) | 0.602(±0.004) |
| Avg. | 0.537(±0.009) | 0.533(±0.013) | 0.530(±0.018) | 0.514(±0.018) | 0.539(±0.011) | 0.534(±0.013) | 0.531(±0.014) | 0.518(±0.016) |

Table 21: Mean and standard deviation (after ±) of target classification accuracy on MiniDomainNet computed with domain adaptation method CoDATS.

| Scenario | Bayesian Model Averaging | | | | Super Learner | | | |
|---|---|---|---|---|---|---|---|---|
| | KuLSIF | Exp | LR | SQ | KuLSIF | Exp | LR | SQ |
| R → C | 0.605(±0.007) | 0.575(±0.012) | 0.579(±0.009) | 0.570(±0.010) | 0.607(±0.008) | 0.575(±0.011) | 0.579(±0.006) | 0.574(±0.008) |
| R → I | 0.372(±0.000) | 0.374(±0.009) | 0.369(±0.024) | 0.352(±0.024) | 0.374(±0.003) | 0.372(±0.008) | 0.369(±0.019) | 0.356(±0.022) |
| R → P | 0.719(±0.006) | 0.715(±0.007) | 0.699(±0.031) | 0.681(±0.030) | 0.721(±0.008) | 0.718(±0.007) | 0.698(±0.026) | 0.685(±0.028) |
| R → Q | 0.372(±0.001) | 0.342(±0.015) | 0.337(±0.013) | 0.314(±0.014) | 0.374(±0.003) | 0.338(±0.013) | 0.339(±0.009) | 0.318(±0.011) |
| R → S | 0.615(±0.003) | 0.606(±0.007) | 0.614(±0.005) | 0.598(±0.005) | 0.617(±0.005) | 0.605(±0.009) | 0.616(±0.001) | 0.602(±0.003) |
| Avg. | 0.537(±0.003) | 0.522(±0.010) | 0.520(±0.016) | 0.503(±0.017) | 0.539(±0.005) | 0.522(±0.010) | 0.520(±0.012) | 0.507(±0.014) |

Table 22: Mean and standard deviation (after ±) of target classification accuracy on MiniDomainNet computed with domain adaptation method DANN.

| Scenario | Bayesian Model Averaging | | | | Super Learner | | | |
|---|---|---|---|---|---|---|---|---|
| | KuLSIF | Exp | LR | SQ | KuLSIF | Exp | LR | SQ |
| R → C | 0.596(±0.020) | 0.595(±0.027) | 0.594(±0.020) | 0.574(±0.022) | 0.598(±0.022) | 0.599(±0.027) | 0.593(±0.017) | 0.578(±0.020) |
| R → I | 0.369(±0.014) | 0.382(±0.010) | 0.337(±0.054) | 0.313(±0.054) | 0.371(±0.016) | 0.385(±0.011) | 0.339(±0.050) | 0.317(±0.052) |
| R → P | 0.729(±0.002) | 0.722(±0.012) | 0.725(±0.010) | 0.705(±0.009) | 0.731(±0.004) | 0.718(±0.014) | 0.724(±0.005) | 0.709(±0.007) |
| R → Q | 0.354(±0.011) | 0.351(±0.019) | 0.354(±0.015) | 0.333(±0.015) | 0.356(±0.012) | 0.354(±0.020) | 0.356(±0.011) | 0.337(±0.013) |
| R → S | 0.608(±0.006) | 0.608(±0.009) | 0.611(±0.007) | 0.599(±0.006) | 0.610(±0.008) | 0.611(±0.011) | 0.611(±0.003) | 0.603(±0.004) |
| Avg. | 0.531(±0.010) | 0.531(±0.015) | 0.524(±0.021) | 0.505(±0.021) | 0.533(±0.012) | 0.533(±0.016) | 0.524(±0.017) | 0.509(±0.019) |

Table 23: Mean and standard deviation (after ±) of target classification accuracy on MiniDomainNet computed with domain adaptation method CDAN.

| Scenario | Bayesian Model Averaging | | | | Super Learner | | | |
|---|---|---|---|---|---|---|---|---|
| | KuLSIF | Exp | LR | SQ | KuLSIF | Exp | LR | SQ |
| R → C | 0.637(±0.006) | 0.605(±0.024) | 0.596(±0.017) | 0.583(±0.017) | 0.639(±0.008) | 0.604(±0.024) | 0.598(±0.013) | 0.587(±0.014) |
| R → I | 0.374(±0.009) | 0.368(±0.006) | 0.350(±0.013) | 0.329(±0.012) | 0.376(±0.012) | 0.371(±0.006) | 0.352(±0.008) | 0.333(±0.011) |
| R → P | 0.717(±0.006) | 0.723(±0.006) | 0.720(±0.005) | 0.714(±0.007) | 0.719(±0.007) | 0.724(±0.006) | 0.722(±0.001) | 0.718(±0.005) |
| R → Q | 0.370(±0.010) | 0.354(±0.016) | 0.354(±0.016) | 0.336(±0.015) | 0.372(±0.011) | 0.353(±0.016) | 0.355(±0.011) | 0.340(±0.013) |
| R → S | 0.620(±0.011) | 0.611(±0.013) | 0.615(±0.013) | 0.600(±0.014) | 0.622(±0.013) | 0.613(±0.013) | 0.617(±0.009) | 0.604(±0.011) |
| Avg. | 0.544(±0.008) | 0.532(±0.013) | 0.527(±0.013) | 0.513(±0.013) | 0.546(±0.010) | 0.533(±0.013) | 0.529(±0.009) | 0.517(±0.011) |

Table 24: Mean and standard deviation (after ±) of target classification accuracy on MiniDomainNet computed with domain adaptation method DSAN.

| Scenario | Bayesian Model Averaging | | | | Super Learner | | | |
|---|---|---|---|---|---|---|---|---|
| | KuLSIF | Exp | LR | SQ | KuLSIF | Exp | LR | SQ |
| R → C | 0.575(±0.023) | 0.476(±0.174) | 0.566(±0.019) | 0.553(±0.020) | 0.577(±0.025) | 0.478(±0.174) | 0.568(±0.015) | 0.557(±0.018) |
| R → I | 0.375(±0.032) | 0.237(±0.134) | 0.406(±0.022) | 0.385(±0.023) | 0.377(±0.034) | 0.239(±0.133) | 0.405(±0.018) | 0.389(±0.021) |
| R → P | 0.711(±0.027) | 0.542(±0.270) | 0.713(±0.023) | 0.694(±0.023) | 0.713(±0.030) | 0.545(±0.269) | 0.715(±0.020) | 0.698(±0.021) |
| R → Q | 0.323(±0.034) | 0.250(±0.053) | 0.329(±0.019) | 0.317(±0.020) | 0.325(±0.036) | 0.246(±0.053) | 0.331(±0.015) | 0.321(±0.019) |
| R → S | 0.581(±0.014) | 0.430(±0.244) | 0.587(±0.020) | 0.565(±0.020) | 0.583(±0.016) | 0.435(±0.243) | 0.589(±0.015) | 0.569(±0.018) |
| Avg. | 0.513(±0.026) | 0.387(±0.175) | 0.520(±0.021) | 0.503(±0.021) | 0.515(±0.028) | 0.389(±0.175) | 0.522(±0.017) | 0.507(±0.019) |

Table 25: Mean and standard deviation (after ±) of target classification accuracy on MiniDomainNet computed with domain adaptation method DIRT.

| Scenario | Bayesian Model Averaging | | | | Super Learner | | | |
|---|---|---|---|---|---|---|---|---|
| | KuLSIF | Exp | LR | SQ | KuLSIF | Exp | LR | SQ |
| R → C | 0.566(±0.007) | 0.563(±0.009) | 0.552(±0.004) | 0.538(±0.005) | 0.568(±0.009) | 0.559(±0.012) | 0.554(±0.001) | 0.542(±0.003) |
| R → I | 0.375(±0.004) | 0.389(±0.005) | 0.385(±0.023) | 0.372(±0.024) | 0.377(±0.013) | 0.393(±0.005) | 0.387(±0.019) | 0.376(±0.021) |
| R → P | 0.719(±0.000) | 0.714(±0.012) | 0.712(±0.009) | 0.704(±0.008) | 0.721(±0.003) | 0.710(±0.012) | 0.714(±0.005) | 0.708(±0.006) |
| R → Q | 0.331(±0.004) | 0.327(±0.005) | 0.324(±0.009) | 0.303(±0.008) | 0.333(±0.005) | 0.325(±0.005) | 0.323(±0.004) | 0.307(±0.006) |
| R → S | 0.580(±0.002) | 0.585(±0.001) | 0.588(±0.003) | 0.584(±0.004) | 0.582(±0.004) | 0.584(±0.001) | 0.587(±0.000) | 0.588(±0.003) |
| Avg. | 0.514(±0.005) | 0.516(±0.007) | 0.512(±0.009) | 0.500(±0.010) | 0.516(±0.007) | 0.514(±0.007) | 0.513(±0.006) | 0.504(±0.008) |

Table 26: Mean and standard deviation (after ±) of target classification accuracy on MiniDomainNet computed with domain adaptation method AdvSKM.

| Scenario | Bayesian Model Averaging | | | | Super Learner | | | |
|---|---|---|---|---|---|---|---|---|
| | KuLSIF | Exp | LR | SQ | KuLSIF | Exp | LR | SQ |
| R → C | 0.594(±0.007) | 0.590(±0.018) | 0.577(±0.018) | 0.566(±0.017) | 0.596(±0.009) | 0.586(±0.018) | 0.576(±0.014) | 0.570(±0.015) |
| R → I | 0.390(±0.006) | 0.409(±0.006) | 0.378(±0.023) | 0.359(±0.023) | 0.392(±0.007) | 0.408(±0.006) | 0.380(±0.020) | 0.363(±0.021) |
| R → P | 0.728(±0.001) | 0.728(±0.006) | 0.726(±0.007) | 0.710(±0.008) | 0.730(±0.003) | 0.725(±0.006) | 0.726(±0.003) | 0.714(±0.005) |
| R → Q | 0.351(±0.001) | 0.335(±0.011) | 0.333(±0.012) | 0.318(±0.011) | 0.353(±0.003) | 0.336(±0.011) | 0.335(±0.007) | 0.322(±0.009) |
| R → S | 0.584(±0.006) | 0.592(±0.010) | 0.592(±0.006) | 0.577(±0.006) | 0.586(±0.008) | 0.593(±0.010) | 0.591(±0.003) | 0.581(±0.005) |
| Avg. | 0.529(±0.004) | 0.531(±0.010) | 0.521(±0.013) | 0.506(±0.013) | 0.531(±0.006) | 0.530(±0.010) | 0.522(±0.009) | 0.510(±0.011) |

Table 27: Mean and standard deviation (after ±) of target classification accuracy on MiniDomainNet computed with domain adaptation method HoMM.

| Scenario | Bayesian Model Averaging | | | | Super Learner | | | |
|---|---|---|---|---|---|---|---|---|
| | KuLSIF | Exp | LR | SQ | KuLSIF | Exp | LR | SQ |
| R → C | 0.566(±0.024) | 0.573(±0.028) | 0.563(±0.027) | 0.558(±0.026) | 0.568(±0.025) | 0.575(±0.028) | 0.562(±0.023) | 0.562(±0.024) |
| R → I | 0.383(±0.006) | 0.393(±0.013) | 0.390(±0.010) | 0.369(±0.009) | 0.385(±0.008) | 0.389(±0.014) | 0.392(±0.005) | 0.373(±0.008) |
| R → P | 0.713(±0.000) | 0.720(±0.001) | 0.715(±0.000) | 0.701(±0.000) | 0.715(±0.002) | 0.721(±0.001) | 0.717(±0.004) | 0.705(±0.003) |
| R → Q | 0.333(±0.002) | 0.322(±0.011) | 0.324(±0.011) | 0.308(±0.011) | 0.335(±0.003) | 0.327(±0.011) | 0.326(±0.006) | 0.312(±0.008) |
| R → S | 0.577(±0.005) | 0.579(±0.003) | 0.579(±0.003) | 0.569(±0.002) | 0.579(±0.007) | 0.582(±0.003) | 0.581(±0.001) | 0.573(±0.001) |
| Avg. | 0.514(±0.007) | 0.517(±0.011) | 0.514(±0.010) | 0.501(±0.010) | 0.516(±0.009) | 0.519(±0.011) | 0.516(±0.008) | 0.505(±0.009) |

Table 28: Mean and standard deviation (after ±) of target classification accuracy on MiniDomainNet computed with domain adaptation method DDC.

| Scenario | Bayesian Model Averaging | | | | Super Learner | | | |
|---|---|---|---|---|---|---|---|---|
| | KuLSIF | Exp | LR | SQ | KuLSIF | Exp | LR | SQ |
| R → C | 0.588(±0.020) | 0.571(±0.014) | 0.567(±0.009) | 0.554(±0.010) | 0.590(±0.022) | 0.573(±0.014) | 0.569(±0.007) | 0.558(±0.008) |
| R → I | 0.365(±0.010) | 0.374(±0.006) | 0.381(±0.017) | 0.370(±0.016) | 0.367(±0.012) | 0.370(±0.006) | 0.383(±0.012) | 0.374(±0.014) |
| R → P | 0.736(±0.010) | 0.737(±0.008) | 0.734(±0.004) | 0.717(±0.003) | 0.738(±0.012) | 0.741(±0.008) | 0.736(±0.002) | 0.721(±0.002) |
| R → Q | 0.378(±0.000) | 0.349(±0.019) | 0.349(±0.015) | 0.336(±0.015) | 0.380(±0.002) | 0.352(±0.019) | 0.350(±0.010) | 0.340(±0.012) |
| R → S | 0.620(±0.000) | 0.616(±0.006) | 0.617(±0.007) | 0.590(±0.007) | 0.622(±0.001) | 0.618(±0.006) | 0.616(±0.004) | 0.594(±0.006) |
| Avg. | 0.537(±0.008) | 0.530(±0.011) | 0.530(±0.010) | 0.513(±0.010) | 0.539(±0.010) | 0.531(±0.011) | 0.531(±0.007) | 0.517(±0.008) |

Table 29: Mean and standard deviation (after ±) of target classification accuracy on MiniDomainNet computed with domain adaptation method DeepCoral.

| Scenario | Bayesian Model Averaging | | | | Super Learner | | | |
|---|---|---|---|---|---|---|---|---|
| | KuLSIF | Exp | LR | SQ | KuLSIF | Exp | LR | SQ |
| R → C | 0.591(±0.007) | 0.562(±0.024) | 0.552(±0.016) | 0.539(±0.015) | 0.593(±0.009) | 0.562(±0.024) | 0.554(±0.011) | 0.543(±0.014) |
| R → I | 0.377(±0.005) | 0.390(±0.019) | 0.381(±0.031) | 0.364(±0.030) | 0.379(±0.007) | 0.395(±0.019) | 0.382(±0.029) | 0.368(±0.028) |
| R → P | 0.738(±0.002) | 0.741(±0.014) | 0.742(±0.008) | 0.723(±0.009) | 0.740(±0.004) | 0.739(±0.014) | 0.741(±0.005) | 0.727(±0.006) |
| R → Q | 0.362(±0.005) | 0.338(±0.028) | 0.335(±0.025) | 0.315(±0.025) | 0.364(±0.007) | 0.340(±0.028) | 0.334(±0.028) | 0.319(±0.023) |
| R → S | 0.590(±0.001) | 0.593(±0.009) | 0.597(±0.012) | 0.581(±0.012) | 0.592(±0.003) | 0.592(±0.009) | 0.599(±0.008) | 0.585(±0.011) |
| Avg. | 0.531(±0.004) | 0.525(±0.019) | 0.521(±0.019) | 0.504(±0.018) | 0.533(±0.006) | 0.525(±0.019) | 0.522(±0.015) | 0.508(±0.016) |

Table 30: Mean and standard deviation (after ±) of target classification accuracy on MiniDomainNet computed with domain adaptation method CMD.

| Scenario | Cross-Validation for Binary Classifier | | | | Aggregation | | | |
|---|---|---|---|---|---|---|---|---|
| | KuLSIF | Exp | LR | SQ | KuLSIF | Exp | LR | SQ |
| B → D | 0.795(±0.003) | 0.784(±0.002) | 0.796(±0.003) | 0.792(±0.001) | **0.799(±0.001)** | **0.794(±0.002)** | **0.802(±0.003)** | **0.796(±0.000)** |
| B → E | 0.776(±0.018) | 0.752(±0.015) | 0.766(±0.016) | 0.762(±0.014) | **0.784(±0.016)** | **0.764(±0.015)** | **0.770(±0.016)** | **0.773(±0.012)** |
| B → K | 0.793(±0.019) | 0.791(±0.020) | 0.794(±0.021) | 0.792(±0.019) | **0.806(±0.017)** | **0.797(±0.018)** | **0.804(±0.021)** | **0.800(±0.017)** |
| D → B | 0.799(±0.007) | 0.770(±0.005) | **0.793(±0.006)** | 0.794(±0.004) | **0.805(±0.004)** | **0.790(±0.004)** | 0.788(±0.006) | **0.805(±0.001)** |
| D → E | 0.791(±0.008) | **0.785(±0.010)** | 0.790(±0.011) | 0.792(±0.010) | **0.797(±0.007)** | 0.783(±0.010) | **0.800(±0.011)** | **0.805(±0.008)** |
| D → K | 0.802(±0.010) | 0.791(±0.008) | 0.801(±0.009) | 0.797(±0.007) | **0.810(±0.008)** | **0.803(±0.007)** | **0.808(±0.009)** | **0.812(±0.005)** |
| E → B | 0.709(±0.015) | 0.693(±0.010) | **0.709(±0.011)** | 0.705(±0.010) | **0.720(±0.014)** | **0.708(±0.009)** | 0.708(±0.011) | **0.718(±0.007)** |
| E → D | 0.746(±0.003) | 0.729(±0.008) | 0.737(±0.009) | 0.727(±0.006) | **0.761(±0.002)** | **0.731(±0.007)** | **0.744(±0.009)** | **0.731(±0.004)** |
| E → K | 0.878(±0.009) | 0.867(±0.008) | 0.876(±0.008) | 0.881(±0.005) | **0.883(±0.006)** | **0.878(±0.007)** | **0.879(±0.008)** | **0.896(±0.004)** |
| K → B | 0.729(±0.005) | 0.723(±0.005) | 0.732(±0.006) | 0.727(±0.004) | **0.740(±0.003)** | **0.734(±0.003)** | **0.742(±0.006)** | 0.727(±0.002) |
| K → D | 0.757(±0.015) | 0.740(±0.011) | 0.756(±0.012) | 0.750(±0.010) | **0.767(±0.013)** | **0.756(±0.011)** | **0.766(±0.012)** | **0.758(±0.008)** |
| K → E | **0.857(±0.008)** | 0.832(±0.007) | 0.858(±0.007) | 0.848(±0.005) | 0.855(±0.005) | 0.850(±0.005) | 0.865(±0.007) | 0.854(±0.004) |
| Avg. | 0.786(±0.010) | 0.771(±0.009) | 0.784(±0.010) | 0.780(±0.008) | **0.794(±0.008)** | **0.782(±0.008)** | **0.790(±0.010)** | **0.789(±0.006)** |

Table 31: Mean and standard deviation (after ±) of target classification accuracy on Amazon Reviews computed with domain adaptation method MMDA.

| Scenario | Cross-Validation for Binary Classifier | | | | Aggregation | | | |
|---|---|---|---|---|---|---|---|---|
| | KuLSIF | Exp | LR | SQ | KuLSIF | Exp | LR | SQ |
| B → D | 0.804(±0.008) | 0.778(±0.006) | **0.806(±0.007)** | 0.796(±0.004) | **0.807(±0.006)** | **0.787(±0.004)** | 0.804(±0.007) | **0.803(±0.002)** |
| B → E | 0.785(±0.012) | 0.763(±0.009) | **0.782(±0.010)** | 0.773(±0.008) | **0.789(±0.011)** | **0.777(±0.007)** | 0.777(±0.010) | **0.777(±0.005)** |
| B → K | 0.807(±0.009) | 0.784(±0.005) | 0.807(±0.007) | 0.797(±0.005) | **0.809(±0.006)** | **0.799(±0.004)** | **0.811(±0.007)** | **0.806(±0.003)** |
| D → B | 0.801(±0.005) | 0.790(±0.004) | 0.802(±0.005) | 0.807(±0.003) | **0.817(±0.003)** | **0.798(±0.003)** | **0.806(±0.005)** | **0.812(±0.001)** |
| D → E | 0.806(±0.011) | 0.787(±0.015) | 0.804(±0.015) | 0.797(±0.013) | **0.815(±0.009)** | **0.797(±0.014)** | **0.808(±0.015)** | **0.805(±0.010)** |
| D → K | 0.819(±0.012) | 0.801(±0.014) | 0.819(±0.016) | 0.810(±0.013) | **0.830(±0.010)** | **0.824(±0.013)** | **0.826(±0.016)** | **0.812(±0.012)** |
| E → B | 0.719(±0.025) | 0.704(±0.031) | 0.721(±0.031) | 0.717(±0.030) | **0.732(±0.024)** | **0.716(±0.030)** | **0.723(±0.031)** | **0.718(±0.029)** |
| E → D | 0.748(±0.024) | 0.729(±0.017) | 0.743(±0.017) | 0.743(±0.015) | **0.755(±0.022)** | **0.737(±0.016)** | **0.750(±0.017)** | **0.748(±0.013)** |
| E → K | 0.883(±0.012) | 0.868(±0.008) | 0.881(±0.009) | 0.874(±0.007) | **0.885(±0.010)** | **0.880(±0.008)** | **0.888(±0.009)** | **0.877(±0.005)** |
| K → B | 0.741(±0.008) | 0.730(±0.007) | 0.734(±0.008) | 0.730(±0.006) | **0.747(±0.007)** | **0.735(±0.005)** | **0.737(±0.008)** | **0.736(±0.004)** |
| K → D | 0.766(±0.007) | 0.745(±0.006) | 0.762(±0.007) | 0.760(±0.004) | **0.773(±0.005)** | **0.762(±0.006)** | **0.765(±0.007)** | **0.766(±0.002)** |
| K → E | 0.866(±0.008) | 0.849(±0.006) | 0.865(±0.007) | 0.855(±0.005) | **0.881(±0.006)** | **0.863(±0.005)** | **0.873(±0.007)** | **0.862(±0.004)** |
| Avg. | 0.795(±0.012) | 0.777(±0.011) | 0.794(±0.012) | 0.788(±0.009) | **0.803(±0.010)** | **0.790(±0.010)** | **0.797(±0.012)** | **0.793(±0.008)** |

Table 32: Mean and standard deviation (after ±) of target classification accuracy on Amazon Reviews computed with domain adaptation method CoDATS.

| Scenario | Cross-Validation for Binary Classifier | | | | Aggregation | | | |
|---|---|---|---|---|---|---|---|---|
| | KuLSIF | Exp | LR | SQ | KuLSIF | Exp | LR | SQ |
| B → D | 0.802(±0.006) | 0.793(±0.008) | **0.802(±0.010)** | 0.805(±0.007) | **0.814(±0.004)** | **0.812(±0.007)** | 0.800(±0.010) | **0.818(±0.005)** |
| B → E | 0.784(±0.012) | 0.743(±0.011) | 0.763(±0.013) | 0.755(±0.010) | **0.793(±0.010)** | **0.749(±0.010)** | **0.770(±0.013)** | **0.762(±0.009)** |
| B → K | 0.801(±0.009) | 0.783(±0.007) | 0.803(±0.008) | 0.802(±0.005) | **0.812(±0.007)** | **0.795(±0.006)** | **0.813(±0.008)** | **0.818(±0.004)** |
| D → B | 0.804(±0.003) | 0.788(±0.002) | 0.799(±0.003) | 0.798(±0.001) | **0.812(±0.001)** | **0.797(±0.001)** | **0.802(±0.003)** | **0.805(±0.001)** |
| D → E | 0.808(±0.007) | 0.787(±0.008) | **0.811(±0.009)** | 0.807(±0.007) | **0.814(±0.004)** | **0.801(±0.007)** | 0.810(±0.009) | **0.819(±0.005)** |
| D → K | 0.814(±0.013) | 0.806(±0.017) | 0.814(±0.018) | 0.813(±0.015) | **0.822(±0.011)** | **0.818(±0.016)** | **0.822(±0.018)** | **0.818(±0.013)** |
| E → B | 0.713(±0.017) | 0.706(±0.014) | 0.717(±0.014) | 0.710(±0.012) | **0.724(±0.015)** | **0.722(±0.012)** | **0.727(±0.014)** | **0.725(±0.010)** |
| E → D | 0.748(±0.009) | 0.733(±0.005) | 0.748(±0.006) | 0.751(±0.005) | **0.758(±0.007)** | **0.752(±0.003)** | **0.758(±0.006)** | **0.758(±0.003)** |
| E → K | 0.878(±0.015) | 0.866(±0.013) | 0.874(±0.015) | 0.871(±0.012) | **0.894(±0.013)** | **0.877(±0.012)** | **0.884(±0.015)** | **0.882(±0.010)** |
| K → B | 0.739(±0.014) | 0.739(±0.003) | 0.750(±0.003) | 0.740(±0.002) | **0.749(±0.012)** | **0.745(±0.002)** | **0.760(±0.003)** | **0.747(±0.001)** |
| K → D | 0.770(±0.020) | 0.752(±0.037) | 0.766(±0.039) | 0.760(±0.036) | **0.782(±0.019)** | **0.767(±0.036)** | **0.776(±0.039)** | **0.764(±0.033)** |
| K → E | 0.863(±0.011) | 0.853(±0.007) | 0.865(±0.007) | 0.870(±0.005) | **0.879(±0.008)** | **0.864(±0.006)** | **0.874(±0.007)** | **0.878(±0.003)** |
| Avg. | 0.794(±0.011) | 0.779(±0.011) | 0.793(±0.012) | 0.790(±0.010) | **0.804(±0.009)** | **0.792(±0.010)** | **0.800(±0.012)** | **0.799(±0.008)** |

Table 33: Mean and standard deviation (after ±) of target classification accuracy on Amazon Reviews computed with domain adaptation method DANN.

| Scenario | Cross-Validation for Binary Classifier | | | | Aggregation | | | |
|---|---|---|---|---|---|---|---|---|
| | KuLSIF | Exp | LR | SQ | KuLSIF | Exp | LR | SQ |
| B → D | 0.802(±0.009) | 0.786(±0.010) | 0.800(±0.011) | 0.800(±0.008) | **0.813(±0.007)** | **0.793(±0.009)** | **0.808(±0.011)** | **0.805(±0.007)** |
| B → E | **0.779(±0.009)** | 0.759(±0.007) | 0.777(±0.007) | 0.777(±0.004) | 0.778(±0.009) | **0.770(±0.006)** | **0.779(±0.007)** | **0.784(±0.001)** |
| B → K | 0.797(±0.007) | 0.776(±0.017) | 0.789(±0.018) | 0.785(±0.016) | **0.800(±0.005)** | **0.783(±0.016)** | **0.794(±0.018)** | **0.791(±0.014)** |
| D → B | 0.797(±0.008) | 0.787(±0.006) | 0.796(±0.007) | 0.797(±0.004) | **0.813(±0.006)** | **0.809(±0.006)** | 0.796(±0.007) | **0.814(±0.003)** |
| D → E | 0.798(±0.008) | 0.791(±0.004) | 0.800(±0.005) | 0.798(±0.003) | **0.799(±0.006)** | **0.801(±0.003)** | **0.802(±0.005)** | **0.804(±0.002)** |
| D → K | 0.804(±0.015) | 0.794(±0.015) | 0.802(±0.016) | 0.803(±0.014) | **0.813(±0.013)** | **0.796(±0.014)** | **0.806(±0.016)** | **0.805(±0.012)** |
| E → B | 0.707(±0.020) | 0.681(±0.014) | 0.705(±0.016) | 0.695(±0.014) | **0.711(±0.018)** | **0.695(±0.014)** | **0.708(±0.016)** | **0.705(±0.012)** |
| E → D | 0.738(±0.011) | 0.724(±0.012) | 0.737(±0.013) | 0.740(±0.010) | **0.744(±0.008)** | **0.732(±0.010)** | **0.744(±0.013)** | 0.740(±0.009) |
| E → K | 0.879(±0.011) | 0.859(±0.013) | 0.875(±0.013) | 0.865(±0.011) | **0.880(±0.009)** | **0.877(±0.010)** | **0.885(±0.013)** | **0.877(±0.009)** |
| K → B | 0.727(±0.014) | 0.717(±0.006) | 0.735(±0.007) | 0.733(±0.006) | **0.741(±0.013)** | **0.730(±0.005)** | **0.737(±0.007)** | **0.744(±0.004)** |
| K → D | **0.754(±0.026)** | 0.755(±0.008) | 0.765(±0.009) | 0.765(±0.007) | 0.752(±0.023) | **0.758(±0.006)** | **0.769(±0.009)** | **0.770(±0.005)** |
| K → E | 0.859(±0.005) | 0.841(±0.004) | **0.858(±0.005)** | 0.859(±0.003) | **0.862(±0.003)** | **0.850(±0.004)** | 0.854(±0.005) | **0.865(±0.000)** |
| Avg. | 0.787(±0.012) | 0.772(±0.010) | 0.787(±0.010) | 0.785(±0.009) | **0.792(±0.010)** | **0.783(±0.009)** | **0.790(±0.010)** | **0.792(±0.007)** |

Table 34: Mean and standard deviation (after ±) of target classification accuracy on Amazon Reviews computed with domain adaptation method CDAN.

|          | **Cross-Validation for Binary Classifier** | | | | **Aggregation** | | | |
| Scenario | KuLSIF | Exp | LR | SQ | KuLSIF | Exp | LR | SQ |
|---|---|---|---|---|---|---|---|---|
| B → D | 0.803(±0.015) | 0.790(±0.011) | 0.803(±0.011) | 0.795(±0.009) | **0.816**(±0.013) | **0.798**(±0.009) | **0.813**(±0.011) | **0.804**(±0.007) |
| B → E | 0.788(±0.004) | 0.772(±0.015) | 0.786(±0.014) | 0.782(±0.014) | **0.794**(±0.003) | **0.792**(±0.014) | **0.793**(±0.016) | **0.796**(±0.012) |
| B → K | 0.804(±0.012) | 0.797(±0.014) | 0.807(±0.015) | 0.803(±0.013) | **0.812**(±0.009) | **0.812**(±0.013) | **0.809**(±0.015) | **0.805**(±0.011) |
| D → B | 0.801(±0.005) | 0.783(±0.003) | 0.802(±0.005) | 0.805(±0.003) | **0.813**(±0.003) | **0.794**(±0.002) | **0.810**(±0.005) | **0.820**(±0.001) |
| D → E | 0.803(±0.008) | 0.786(±0.008) | 0.801(±0.009) | 0.804(±0.007) | **0.815**(±0.006) | **0.791**(±0.007) | **0.805**(±0.009) | **0.821**(±0.005) |
| D → K | 0.803(±0.029) | 0.793(±0.015) | 0.805(±0.017) | 0.810(±0.014) | **0.813**(±0.028) | **0.809**(±0.014) | **0.807**(±0.017) | **0.812**(±0.014) |
| E → B | 0.719(±0.021) | 0.704(±0.021) | 0.719(±0.022) | 0.714(±0.020) | **0.733**(±0.020) | **0.717**(±0.020) | **0.725**(±0.022) | **0.724**(±0.018) |
| E → D | 0.749(±0.004) | 0.735(±0.008) | 0.746(±0.009) | 0.738(±0.007) | **0.764**(±0.002) | **0.754**(±0.008) | **0.753**(±0.009) | **0.740**(±0.005) |
| E → K | 0.883(±0.007) | 0.854(±0.007) | 0.881(±0.008) | **0.873**(±0.008) | **0.898**(±0.008) | **0.862**(±0.007) | **0.890**(±0.008) | 0.871(±0.004) |
| K → B | **0.739**(±0.010) | 0.720(±0.017) | 0.732(±0.018) | 0.727(±0.016) | 0.738(±0.008) | **0.731**(±0.016) | 0.732(±0.018) | **0.732**(±0.014) |
| K → D | 0.767(±0.008) | 0.753(±0.014) | **0.770**(±0.015) | 0.770(±0.012) | **0.773**(±0.005) | **0.764**(±0.013) | 0.766(±0.015) | **0.782**(±0.010) |
| K → E | 0.864(±0.006) | 0.847(±0.006) | 0.863(±0.007) | 0.858(±0.004) | **0.870**(±0.004) | **0.858**(±0.005) | **0.870**(±0.007) | **0.870**(±0.002) |
| Avg. | 0.794(±0.011) | 0.778(±0.012) | 0.793(±0.013) | 0.790(±0.011) | **0.803**(±0.009) | **0.790**(±0.011) | **0.798**(±0.013) | **0.798**(±0.009) |

Table 35: Mean and standard deviation (after ±) of target classification accuracy on Amazon Reviews computed with domain adaptation method DSAN.

|          | **Cross-Validation for Binary Classifier** | | | | **Aggregation** | | | |
| Scenario | KuLSIF | Exp | LR | SQ | KuLSIF | Exp | LR | SQ |
|---|---|---|---|---|---|---|---|---|
| B → D | 0.812(±0.004) | 0.795(±0.004) | **0.811**(±0.005) | 0.804(±0.003) | **0.825**(±0.003) | 0.804(±0.003) | 0.810(±0.005) | **0.817**(±0.000) |
| B → E | 0.786(±0.007) | 0.769(±0.008) | 0.779(±0.008) | 0.771(±0.005) | **0.795**(±0.005) | **0.784**(±0.007) | **0.786**(±0.008) | 0.784(±0.003) |
| B → K | 0.809(±0.006) | 0.793(±0.006) | 0.797(±0.007) | 0.794(±0.005) | **0.816**(±0.004) | **0.801**(±0.004) | 0.797(±0.007) | **0.805**(±0.002) |
| D → B | 0.806(±0.009) | 0.801(±0.008) | **0.811**(±0.009) | 0.807(±0.007) | **0.809**(±0.007) | **0.812**(±0.008) | 0.809(±0.009) | **0.812**(±0.005) |
| D → E | 0.804(±0.012) | 0.789(±0.021) | 0.802(±0.022) | 0.800(±0.021) | **0.810**(±0.011) | **0.813**(±0.020) | 0.806(±0.022) | 0.805(±0.020) |
| D → K | 0.830(±0.012) | 0.809(±0.017) | 0.823(±0.018) | 0.817(±0.016) | **0.844**(±0.010) | **0.826**(±0.017) | **0.832**(±0.018) | **0.829**(±0.014) |
| E → B | 0.686(±0.016) | 0.675(±0.018) | 0.685(±0.019) | 0.680(±0.017) | **0.696**(±0.013) | **0.687**(±0.017) | **0.689**(±0.019) | **0.684**(±0.016) |
| E → D | 0.711(±0.017) | 0.706(±0.016) | 0.715(±0.016) | 0.708(±0.015) | **0.727**(±0.016) | **0.711**(±0.015) | 0.715(±0.016) | **0.726**(±0.013) |
| E → K | 0.885(±0.009) | 0.868(±0.006) | 0.880(±0.008) | 0.876(±0.005) | **0.898**(±0.008) | **0.878**(±0.005) | **0.886**(±0.008) | **0.885**(±0.002) |
| K → B | 0.720(±0.011) | 0.717(±0.010) | **0.731**(±0.011) | 0.732(±0.009) | **0.730**(±0.010) | **0.722**(±0.008) | 0.729(±0.011) | **0.737**(±0.007) |
| K → D | 0.729(±0.013) | 0.732(±0.009) | **0.748**(±0.010) | 0.750(±0.009) | **0.742**(±0.011) | **0.733**(±0.008) | 0.746(±0.010) | **0.764**(±0.007) |
| K → E | 0.863(±0.013) | 0.857(±0.008) | 0.870(±0.009) | 0.867(±0.008) | **0.872**(±0.011) | **0.872**(±0.008) | **0.880**(±0.009) | **0.876**(±0.005) |
| Avg. | 0.787(±0.011) | 0.776(±0.011) | 0.788(±0.012) | 0.784(±0.010) | **0.797**(±0.009) | **0.787**(±0.010) | **0.790**(±0.012) | **0.794**(±0.008) |

Table 36: Mean and standard deviation (after ±) of target classification accuracy on Amazon Reviews computed with domain adaptation method DIRT.

|          | **Cross-Validation for Binary Classifier** | | | | **Aggregation** | | | |
| Scenario | KuLSIF | Exp | LR | SQ | KuLSIF | Exp | LR | SQ |
|---|---|---|---|---|---|---|---|---|
| B → D | 0.794(±0.005) | 0.770(±0.005) | 0.786(±0.006) | 0.778(±0.004) | **0.801**(±0.003) | **0.787**(±0.003) | **0.793**(±0.006) | **0.784**(±0.002) |
| B → E | 0.760(±0.012) | 0.744(±0.010) | 0.756(±0.011) | 0.755(±0.009) | **0.764**(±0.010) | **0.750**(±0.010) | **0.761**(±0.011) | **0.762**(±0.007) |
| B → K | 0.782(±0.018) | 0.765(±0.015) | 0.781(±0.017) | 0.776(±0.014) | **0.784**(±0.015) | **0.778**(±0.015) | **0.786**(±0.017) | **0.785**(±0.012) |
| D → B | 0.794(±0.010) | 0.780(±0.004) | 0.794(±0.005) | 0.785(±0.003) | **0.798**(±0.008) | **0.787**(±0.004) | **0.795**(±0.005) | **0.795**(±0.001) |
| D → E | 0.783(±0.006) | 0.768(±0.004) | 0.776(±0.005) | 0.776(±0.004) | **0.790**(±0.004) | **0.791**(±0.004) | **0.779**(±0.005) | **0.792**(±0.002) |
| D → K | **0.789**(±0.013) | 0.771(±0.011) | 0.789(±0.011) | 0.779(±0.009) | 0.786(±0.011) | **0.780**(±0.010) | **0.796**(±0.011) | **0.785**(±0.007) |
| E → B | 0.711(±0.021) | 0.694(±0.017) | 0.706(±0.018) | 0.702(±0.016) | **0.721**(±0.019) | **0.709**(±0.016) | 0.706(±0.018) | **0.717**(±0.014) |
| E → D | 0.732(±0.007) | 0.722(±0.007) | 0.731(±0.007) | 0.726(±0.005) | **0.746**(±0.004) | **0.729**(±0.007) | **0.734**(±0.007) | **0.728**(±0.003) |
| E → K | 0.875(±0.011) | 0.861(±0.010) | 0.874(±0.012) | 0.870(±0.010) | **0.882**(±0.008) | **0.885**(±0.009) | **0.880**(±0.012) | **0.876**(±0.008) |
| K → B | 0.723(±0.007) | 0.701(±0.004) | 0.723(±0.005) | 0.725(±0.003) | **0.726**(±0.005) | **0.706**(±0.003) | **0.729**(±0.005) | **0.742**(±0.001) |
| K → D | 0.748(±0.014) | 0.726(±0.014) | 0.750(±0.014) | 0.743(±0.012) | 0.748(±0.013) | **0.740**(±0.013) | **0.757**(±0.014) | **0.754**(±0.010) |
| K → E | 0.858(±0.008) | 0.833(±0.006) | 0.854(±0.007) | 0.849(±0.005) | **0.867**(±0.006) | **0.838**(±0.005) | **0.858**(±0.007) | **0.858**(±0.003) |
| Avg. | 0.779(±0.011) | 0.761(±0.009) | 0.777(±0.010) | 0.772(±0.008) | **0.784**(±0.009) | **0.774**(±0.008) | **0.781**(±0.010) | **0.782**(±0.006) |

Table 37: Mean and standard deviation (after ±) of target classification accuracy on Amazon Reviews computed with domain adaptation method AdvSKM.

|          | **Cross-Validation for Binary Classifier** | | | | **Aggregation** | | | |
| Scenario | KuLSIF | Exp | LR | SQ | KuLSIF | Exp | LR | SQ |
|---|---|---|---|---|---|---|---|---|
| B → D | 0.794(±0.007) | 0.777(±0.008) | 0.792(±0.010) | 0.782(±0.008) | **0.794**(±0.006) | **0.788**(±0.007) | **0.798**(±0.010) | **0.794**(±0.006) |
| B → E | 0.761(±0.012) | 0.746(±0.018) | 0.756(±0.018) | 0.755(±0.017) | **0.770**(±0.010) | **0.766**(±0.017) | 0.756(±0.018) | **0.773**(±0.014) |
| B → K | 0.781(±0.017) | 0.764(±0.021) | 0.776(±0.022) | 0.776(±0.020) | **0.792**(±0.015) | **0.781**(±0.020) | **0.781**(±0.022) | **0.787**(±0.017) |
| D → B | 0.790(±0.002) | 0.776(±0.002) | 0.789(±0.003) | 0.783(±0.001) | **0.795**(±0.000) | **0.792**(±0.002) | **0.790**(±0.003) | **0.795**(±0.002) |
| D → E | 0.781(±0.004) | 0.756(±0.003) | 0.772(±0.004) | 0.777(±0.002) | **0.790**(±0.002) | **0.767**(±0.002) | **0.778**(±0.004) | **0.783**(±0.000) |
| D → K | 0.789(±0.015) | 0.770(±0.014) | 0.788(±0.015) | 0.785(±0.014) | **0.800**(±0.013) | **0.783**(±0.013) | **0.798**(±0.015) | **0.792**(±0.011) |
| E → B | 0.699(±0.012) | 0.679(±0.013) | 0.693(±0.014) | 0.689(±0.013) | **0.710**(±0.011) | **0.698**(±0.012) | **0.701**(±0.014) | **0.699**(±0.011) |
| E → D | 0.735(±0.011) | 0.722(±0.009) | 0.732(±0.010) | 0.722(±0.009) | **0.744**(±0.008) | **0.724**(±0.008) | **0.734**(±0.010) | **0.727**(±0.007) |
| E → K | 0.873(±0.009) | 0.857(±0.007) | 0.872(±0.008) | 0.862(±0.006) | **0.884**(±0.007) | **0.865**(±0.007) | **0.881**(±0.008) | **0.863**(±0.005) |
| K → B | 0.722(±0.006) | 0.703(±0.012) | **0.717**(±0.013) | 0.711(±0.011) | **0.723**(±0.003) | **0.709**(±0.011) | 0.713(±0.013) | **0.724**(±0.009) |
| K → D | 0.748(±0.008) | 0.734(±0.008) | 0.752(±0.009) | 0.753(±0.006) | **0.760**(±0.006) | **0.742**(±0.006) | **0.762**(±0.009) | **0.759**(±0.005) |
| K → E | 0.856(±0.006) | 0.834(±0.007) | 0.854(±0.007) | 0.847(±0.006) | **0.864**(±0.004) | **0.849**(±0.006) | **0.860**(±0.007) | **0.857**(±0.003) |
| Avg. | 0.777(±0.009) | 0.760(±0.010) | 0.774(±0.011) | 0.770(±0.009) | **0.786**(±0.007) | **0.772**(±0.009) | **0.779**(±0.011) | **0.779**(±0.008) |

Table 38: Mean and standard deviation (after ±) of target classification accuracy on Amazon Reviews computed with domain adaptation method HoMM.

|  | Cross-Validation for Binary Classifier | | | | Aggregation | | | |
|---|---|---|---|---|---|---|---|---|
| Scenario | KuLSIF | Exp | LR | SQ | KuLSIF | Exp | LR | SQ |
| B → D | 0.790(±0.009) | 0.775(±0.006) | 0.790(±0.008) | 0.783(±0.006) | **0.794(±0.007)** | **0.797(±0.005)** | **0.796(±0.008)** | **0.789(±0.004)** |
| B → E | 0.763(±0.014) | 0.749(±0.015) | 0.760(±0.016) | 0.750(±0.014) | **0.774(±0.012)** | **0.759(±0.013)** | **0.763(±0.016)** | **0.752(±0.012)** |
| B → K | 0.783(±0.014) | 0.774(±0.022) | 0.779(±0.022) | 0.772(±0.020) | **0.793(±0.012)** | **0.787(±0.021)** | **0.786(±0.022)** | **0.780(±0.018)** |
| D → B | 0.795(±0.006) | 0.780(±0.007) | 0.791(±0.008) | 0.783(±0.006) | **0.796(±0.004)** | **0.799(±0.006)** | **0.793(±0.008)** | **0.790(±0.005)** |
| D → E | 0.784(±0.004) | 0.757(±0.007) | 0.778(±0.008) | 0.782(±0.005) | **0.790(±0.001)** | **0.776(±0.007)** | **0.784(±0.008)** | **0.792(±0.004)** |
| D → K | 0.789(±0.021) | 0.780(±0.014) | 0.790(±0.015) | 0.781(±0.014) | **0.796(±0.018)** | **0.787(±0.014)** | **0.792(±0.015)** | **0.789(±0.011)** |
| E → B | 0.705(±0.024) | 0.680(±0.022) | 0.699(±0.024) | 0.702(±0.022) | **0.713(±0.022)** | **0.692(±0.021)** | **0.700(±0.024)** | **0.713(±0.020)** |
| E → D | 0.736(±0.007) | 0.727(±0.007) | **0.740(±0.007)** | 0.739(±0.005) | **0.746(±0.005)** | **0.744(±0.005)** | 0.737(±0.007) | **0.742(±0.003)** |
| E → K | 0.877(±0.011) | 0.862(±0.007) | 0.874(±0.009) | 0.879(±0.007) | **0.885(±0.009)** | **0.880(±0.006)** | **0.884(±0.009)** | **0.890(±0.004)** |
| K → B | 0.726(±0.003) | 0.714(±0.006) | 0.722(±0.007) | 0.719(±0.005) | **0.734(±0.001)** | **0.721(±0.006)** | **0.731(±0.007)** | **0.737(±0.004)** |
| K → D | 0.751(±0.013) | 0.736(±0.007) | 0.752(±0.009) | 0.750(±0.006) | **0.756(±0.011)** | **0.750(±0.007)** | **0.760(±0.009)** | **0.759(±0.004)** |
| K → E | 0.856(±0.008) | 0.837(±0.006) | 0.856(±0.007) | 0.852(±0.005) | **0.862(±0.005)** | 0.837(±0.004) | **0.865(±0.007)** | **0.864(±0.003)** |
| Avg. | 0.780(±0.011) | 0.764(±0.011) | 0.778(±0.012) | 0.774(±0.010) | **0.787(±0.009)** | **0.777(±0.010)** | **0.782(±0.012)** | **0.783(±0.008)** |

Table 39: Mean and standard deviation (after ±) of target classification accuracy on Amazon Reviews computed with domain adaptation method DDC.

|  | Cross-Validation for Binary Classifier | | | | Aggregation | | | |
|---|---|---|---|---|---|---|---|---|
| Scenario | KuLSIF | Exp | LR | SQ | KuLSIF | Exp | LR | SQ |
| B → D | 0.801(±0.010) | 0.795(±0.009) | 0.799(±0.011) | 0.790(±0.009) | **0.814(±0.008)** | **0.812(±0.008)** | **0.804(±0.011)** | **0.801(±0.006)** |
| B → E | 0.773(±0.008) | 0.753(±0.012) | 0.763(±0.012) | 0.756(±0.010) | **0.781(±0.006)** | **0.762(±0.011)** | **0.772(±0.012)** | **0.765(±0.008)** |
| B → K | 0.785(±0.017) | 0.765(±0.017) | 0.781(±0.018) | 0.773(±0.016) | **0.797(±0.016)** | **0.782(±0.017)** | **0.789(±0.018)** | **0.779(±0.013)** |
| D → B | 0.797(±0.004) | 0.783(±0.002) | 0.794(±0.003) | 0.789(±0.001) | **0.803(±0.002)** | **0.794(±0.000)** | **0.804(±0.003)** | **0.803(±0.000)** |
| D → E | 0.785(±0.003) | 0.769(±0.009) | **0.779(±0.011)** | 0.769(±0.009) | **0.793(±0.002)** | **0.778(±0.008)** | 0.777(±0.011) | **0.775(±0.007)** |
| D → K | 0.792(±0.014) | 0.774(±0.016) | 0.791(±0.016) | 0.785(±0.013) | **0.799(±0.012)** | **0.787(±0.015)** | **0.792(±0.016)** | 0.785(±0.010) |
| E → B | 0.710(±0.024) | 0.695(±0.022) | 0.709(±0.024) | 0.706(±0.022) | **0.713(±0.022)** | **0.704(±0.022)** | **0.717(±0.024)** | **0.709(±0.020)** |
| E → D | 0.739(±0.004) | 0.724(±0.002) | 0.739(±0.003) | 0.729(±0.000) | **0.747(±0.002)** | **0.731(±0.002)** | **0.748(±0.003)** | **0.738(±0.001)** |
| E → K | 0.878(±0.007) | 0.863(±0.007) | 0.876(±0.009) | 0.875(±0.007) | **0.894(±0.005)** | **0.882(±0.006)** | **0.878(±0.009)** | **0.886(±0.005)** |
| K → B | 0.739(±0.009) | 0.720(±0.007) | **0.735(±0.008)** | 0.738(±0.005) | **0.754(±0.007)** | **0.735(±0.007)** | 0.733(±0.008) | **0.756(±0.003)** |
| K → D | 0.755(±0.015) | 0.733(±0.016) | 0.749(±0.017) | 0.746(±0.014) | **0.770(±0.013)** | **0.743(±0.015)** | **0.754(±0.017)** | **0.753(±0.011)** |
| K → E | 0.857(±0.010) | 0.830(±0.004) | **0.855(±0.005)** | 0.851(±0.004) | **0.864(±0.008)** | **0.842(±0.003)** | 0.850(±0.005) | **0.863(±0.002)** |
| Avg. | 0.784(±0.010) | 0.767(±0.010) | 0.781(±0.011) | 0.776(±0.009) | **0.794(±0.008)** | **0.779(±0.009)** | **0.785(±0.011)** | **0.784(±0.007)** |

Table 40: Mean and standard deviation (after ±) of target classification accuracy on Amazon Reviews computed with domain adaptation method DeepCoral.

|  | Cross-Validation for Binary Classifier | | | | Aggregation | | | |
|---|---|---|---|---|---|---|---|---|
| Scenario | KuLSIF | Exp | LR | SQ | KuLSIF | Exp | LR | SQ |
| B → D | 0.799(±0.005) | 0.785(±0.007) | 0.794(±0.009) | **0.792(±0.007)** | **0.809(±0.002)** | **0.796(±0.007)** | **0.803(±0.009)** | 0.790(±0.004) |
| B → E | 0.779(±0.014) | 0.747(±0.020) | **0.761(±0.021)** | 0.756(±0.019) | **0.787(±0.012)** | **0.759(±0.018)** | 0.756(±0.021) | **0.758(±0.017)** |
| B → K | 0.781(±0.029) | 0.782(±0.010) | 0.793(±0.011) | 0.783(±0.009) | **0.788(±0.026)** | **0.791(±0.009)** | **0.799(±0.011)** | **0.791(±0.007)** |
| D → B | 0.798(±0.012) | 0.791(±0.006) | 0.802(±0.007) | 0.792(±0.005) | **0.799(±0.010)** | **0.796(±0.005)** | **0.806(±0.007)** | **0.798(±0.003)** |
| D → E | 0.795(±0.009) | 0.777(±0.009) | 0.802(±0.010) | 0.792(±0.008) | **0.811(±0.007)** | **0.795(±0.008)** | **0.812(±0.010)** | **0.800(±0.006)** |
| D → K | 0.799(±0.018) | 0.786(±0.013) | **0.804(±0.015)** | 0.796(±0.013) | **0.812(±0.016)** | **0.797(±0.013)** | 0.803(±0.015) | **0.804(±0.011)** |
| E → B | 0.719(±0.014) | 0.714(±0.016) | 0.721(±0.017) | 0.726(±0.016) | **0.732(±0.012)** | **0.727(±0.015)** | **0.727(±0.017)** | **0.732(±0.013)** |
| E → D | **0.750(±0.009)** | 0.730(±0.007) | 0.744(±0.008) | 0.742(±0.007) | 0.746(±0.007) | **0.738(±0.006)** | **0.752(±0.008)** | **0.752(±0.005)** |
| E → K | 0.875(±0.015) | 0.857(±0.009) | **0.875(±0.010)** | 0.867(±0.007) | **0.879(±0.013)** | **0.875(±0.008)** | 0.872(±0.010) | **0.880(±0.004)** |
| K → B | 0.745(±0.004) | 0.719(±0.004) | 0.731(±0.004) | 0.731(±0.003) | **0.748(±0.003)** | **0.727(±0.003)** | **0.733(±0.004)** | **0.732(±0.001)** |
| K → D | 0.775(±0.019) | 0.723(±0.030) | 0.740(±0.031) | 0.730(±0.029) | **0.783(±0.018)** | **0.732(±0.030)** | **0.749(±0.031)** | **0.740(±0.027)** |
| K → E | 0.864(±0.010) | 0.859(±0.005) | 0.867(±0.006) | 0.865(±0.004) | **0.868(±0.008)** | **0.866(±0.004)** | **0.870(±0.006)** | **0.866(±0.002)** |
| Avg. | 0.790(±0.013) | 0.773(±0.011) | 0.786(±0.012) | 0.781(±0.011) | **0.797(±0.011)** | **0.783(±0.011)** | **0.790(±0.012)** | **0.787(±0.008)** |

Table 41: Mean and standard deviation (after ±) of target classification accuracy on Amazon Reviews computed with domain adaptation method CMD.

| Scenario | Bayesian Model Averaging | | | | Super Learner | | | |
|---|---|---|---|---|---|---|---|---|
| | KuLSIF | Exp | LR | SQ | KuLSIF | Exp | LR | SQ |
| B → D | 0.790(±0.001) | 0.783(±0.002) | 0.796(±0.003) | 0.791(±0.001) | 0.799(±0.002) | 0.788(±0.001) | 0.795(±0.002) | 0.792(±0.001) |
| B → E | 0.772(±0.028) | 0.753(±0.015) | 0.766(±0.016) | 0.764(±0.013) | 0.777(±0.026) | 0.756(±0.014) | 0.765(±0.015) | 0.766(±0.011) |
| B → K | 0.794(±0.021) | 0.790(±0.020) | 0.794(±0.021) | 0.792(±0.018) | 0.796(±0.018) | 0.795(±0.019) | 0.793(±0.020) | 0.798(±0.017) |
| D → B | 0.803(±0.001) | 0.772(±0.005) | 0.793(±0.006) | 0.796(±0.002) | 0.803(±0.002) | 0.768(±0.004) | 0.795(±0.005) | 0.792(±0.002) |
| D → E | 0.784(±0.012) | 0.784(±0.010) | 0.790(±0.011) | 0.794(±0.009) | 0.790(±0.010) | 0.789(±0.009) | 0.792(±0.011) | 0.793(±0.008) |
| D → K | 0.808(±0.003) | 0.792(±0.008) | 0.801(±0.009) | 0.799(±0.006) | 0.813(±0.001) | 0.795(±0.007) | 0.803(±0.009) | 0.803(±0.006) |
| E → B | 0.715(±0.011) | 0.695(±0.010) | 0.709(±0.011) | 0.707(±0.008) | 0.717(±0.009) | 0.693(±0.010) | 0.711(±0.011) | 0.709(±0.008) |
| E → D | 0.747(±0.006) | 0.728(±0.008) | 0.737(±0.009) | 0.720(±0.006) | 0.745(±0.004) | 0.727(±0.007) | 0.736(±0.008) | 0.719(±0.005) |
| E → K | 0.877(±0.006) | 0.867(±0.008) | 0.876(±0.008) | 0.885(±0.006) | 0.880(±0.004) | 0.871(±0.006) | 0.875(±0.007) | 0.886(±0.005) |
| K → B | 0.729(±0.004) | 0.723(±0.005) | 0.732(±0.006) | 0.726(±0.003) | 0.719(±0.002) | 0.723(±0.004) | 0.731(±0.006) | 0.724(±0.001) |
| K → D | 0.761(±0.009) | 0.742(±0.011) | 0.756(±0.012) | 0.750(±0.010) | 0.764(±0.007) | 0.739(±0.009) | 0.757(±0.011) | 0.756(±0.009) |
| K → E | 0.852(±0.010) | 0.834(±0.007) | 0.858(±0.007) | 0.845(±0.005) | 0.861(±0.008) | 0.836(±0.005) | 0.860(±0.006) | 0.851(±0.003) |
| Avg. | 0.786(±0.009) | 0.772(±0.009) | 0.784(±0.010) | 0.781(±0.007) | 0.789(±0.008) | 0.773(±0.008) | 0.784(±0.009) | 0.782(±0.006) |

Table 42: Mean and standard deviation (after ±) of target classification accuracy on Amazon Reviews computed with domain adaptation method MMDA.

| Scenario | Bayesian Model Averaging | | | | Super Learner | | | |
|---|---|---|---|---|---|---|---|---|
| | KuLSIF | Exp | LR | SQ | KuLSIF | Exp | LR | SQ |
| B → D | 0.800(±0.004) | 0.798(±0.006) | 0.806(±0.007) | 0.805(±0.004) | 0.800(±0.002) | 0.803(±0.005) | 0.807(±0.006) | 0.801(±0.003) |
| B → E | 0.785(±0.020) | 0.780(±0.009) | 0.782(±0.010) | 0.781(±0.007) | 0.784(±0.018) | 0.778(±0.008) | 0.784(±0.010) | 0.781(±0.005) |
| B → K | 0.815(±0.005) | 0.792(±0.007) | 0.807(±0.007) | 0.802(±0.004) | 0.817(±0.003) | 0.797(±0.007) | 0.808(±0.006) | 0.804(±0.004) |
| D → B | 0.797(±0.004) | 0.789(±0.004) | 0.802(±0.005) | 0.801(±0.002) | 0.800(±0.002) | 0.790(±0.003) | 0.801(±0.004) | 0.799(±0.001) |
| D → E | 0.802(±0.011) | 0.791(±0.014) | 0.804(±0.015) | 0.802(±0.013) | 0.795(±0.010) | 0.795(±0.013) | 0.803(±0.015) | 0.800(±0.011) |
| D → K | 0.815(±0.014) | 0.801(±0.015) | 0.819(±0.016) | 0.822(±0.014) | 0.823(±0.012) | 0.800(±0.014) | 0.818(±0.014) | 0.818(±0.013) |
| E → B | 0.723(±0.037) | 0.712(±0.030) | 0.721(±0.031) | 0.726(±0.029) | 0.731(±0.035) | 0.714(±0.029) | 0.723(±0.029) | 0.730(±0.028) |
| E → D | 0.755(±0.018) | 0.726(±0.015) | 0.743(±0.017) | 0.744(±0.013) | 0.759(±0.016) | 0.728(±0.014) | 0.744(±0.017) | 0.739(±0.013) |
| E → K | 0.880(±0.009) | 0.863(±0.008) | 0.881(±0.009) | 0.881(±0.006) | 0.891(±0.007) | 0.868(±0.007) | 0.883(±0.008) | 0.887(±0.006) |
| K → B | 0.752(±0.009) | 0.723(±0.007) | 0.734(±0.008) | 0.731(±0.005) | 0.759(±0.008) | 0.721(±0.005) | 0.736(±0.007) | 0.732(±0.005) |
| K → D | 0.768(±0.010) | 0.751(±0.006) | 0.762(±0.007) | 0.762(±0.003) | 0.773(±0.009) | 0.747(±0.005) | 0.761(±0.006) | 0.757(±0.003) |
| K → E | 0.863(±0.008) | 0.847(±0.007) | 0.865(±0.007) | 0.862(±0.005) | 0.870(±0.006) | 0.849(±0.006) | 0.867(±0.006) | 0.859(±0.004) |
| Avg. | 0.796(±0.012) | 0.781(±0.011) | 0.794(±0.012) | 0.793(±0.009) | 0.800(±0.011) | 0.782(±0.010) | 0.794(±0.011) | 0.792(±0.008) |

Table 43: Mean and standard deviation (after ±) of target classification accuracy on Amazon Reviews computed with domain adaptation method CoDATS.

| Scenario | Bayesian Model Averaging | | | | Super Learner | | | |
|---|---|---|---|---|---|---|---|---|
| | KuLSIF | Exp | LR | SQ | KuLSIF | Exp | LR | SQ |
| B → D | 0.804(±0.000) | 0.783(±0.009) | 0.802(±0.010) | 0.804(±0.007) | 0.806(±0.003) | 0.782(±0.008) | 0.804(±0.009) | 0.805(±0.006) |
| B → E | 0.774(±0.016) | 0.757(±0.011) | 0.763(±0.013) | 0.759(±0.010) | 0.776(±0.015) | 0.762(±0.010) | 0.765(±0.011) | 0.765(±0.009) |
| B → K | 0.796(±0.009) | 0.793(±0.007) | 0.803(±0.008) | 0.802(±0.005) | 0.797(±0.006) | 0.798(±0.006) | 0.802(±0.007) | 0.799(±0.004) |
| D → B | 0.802(±0.001) | 0.779(±0.001) | 0.799(±0.003) | 0.782(±0.001) | 0.811(±0.003) | 0.777(±0.001) | 0.801(±0.002) | 0.789(±0.000) |
| D → E | 0.804(±0.005) | 0.802(±0.008) | 0.811(±0.009) | 0.806(±0.006) | 0.812(±0.002) | 0.801(±0.007) | 0.811(±0.009) | 0.813(±0.005) |
| D → K | 0.812(±0.015) | 0.798(±0.016) | 0.814(±0.018) | 0.818(±0.015) | 0.817(±0.014) | 0.803(±0.015) | 0.813(±0.016) | 0.816(±0.015) |
| E → B | 0.716(±0.024) | 0.709(±0.013) | 0.717(±0.014) | 0.711(±0.011) | 0.722(±0.022) | 0.710(±0.012) | 0.719(±0.013) | 0.714(±0.011) |
| E → D | 0.749(±0.005) | 0.735(±0.004) | 0.748(±0.006) | 0.749(±0.003) | 0.752(±0.004) | 0.740(±0.004) | 0.750(±0.005) | 0.744(±0.002) |
| E → K | 0.878(±0.014) | 0.869(±0.013) | 0.874(±0.015) | 0.864(±0.013) | 0.880(±0.013) | 0.865(±0.013) | 0.876(±0.014) | 0.859(±0.009) |
| K → B | 0.717(±0.031) | 0.732(±0.003) | 0.750(±0.003) | 0.745(±0.001) | 0.713(±0.028) | 0.731(±0.002) | 0.749(±0.003) | 0.743(±0.001) |
| K → D | 0.758(±0.033) | 0.760(±0.037) | 0.766(±0.039) | 0.768(±0.035) | 0.758(±0.030) | 0.761(±0.036) | 0.765(±0.037) | 0.775(±0.035) |
| K → E | 0.855(±0.010) | 0.856(±0.006) | 0.865(±0.007) | 0.864(±0.005) | 0.860(±0.008) | 0.858(±0.005) | 0.867(±0.006) | 0.869(±0.004) |
| Avg. | 0.789(±0.014) | 0.781(±0.011) | 0.793(±0.012) | 0.790(±0.009) | 0.792(±0.012) | 0.782(±0.010) | 0.793(±0.011) | 0.791(±0.009) |

Table 44: Mean and standard deviation (after ±) of target classification accuracy on Amazon Reviews computed with domain adaptation method DANN.

| Scenario | Bayesian Model Averaging | | | | Super Learner | | | |
|---|---|---|---|---|---|---|---|---|
| | KuLSIF | Exp | LR | SQ | KuLSIF | Exp | LR | SQ |
| B → D | 0.805(±0.006) | 0.783(±0.009) | 0.800(±0.011) | 0.795(±0.007) | 0.805(±0.004) | 0.788(±0.009) | 0.800(±0.009) | 0.799(±0.006) |
| B → E | 0.784(±0.007) | 0.769(±0.006) | 0.777(±0.007) | 0.771(±0.004) | 0.782(±0.005) | 0.765(±0.005) | 0.776(±0.006) | 0.766(±0.003) |
| B → K | 0.801(±0.002) | 0.785(±0.016) | 0.789(±0.018) | 0.789(±0.014) | 0.805(±0.004) | 0.788(±0.015) | 0.788(±0.016) | 0.790(±0.014) |
| D → B | 0.793(±0.009) | 0.787(±0.006) | 0.796(±0.007) | 0.791(±0.004) | 0.799(±0.006) | 0.790(±0.005) | 0.795(±0.006) | 0.790(±0.004) |
| D → E | 0.796(±0.014) | 0.789(±0.005) | 0.800(±0.005) | 0.797(±0.004) | 0.793(±0.012) | 0.785(±0.004) | 0.800(±0.004) | 0.804(±0.003) |
| D → K | 0.815(±0.018) | 0.790(±0.015) | 0.802(±0.016) | 0.798(±0.014) | 0.817(±0.016) | 0.792(±0.014) | 0.801(±0.015) | 0.795(±0.012) |
| E → B | 0.708(±0.020) | 0.695(±0.015) | 0.705(±0.016) | 0.699(±0.014) | 0.704(±0.020) | 0.695(±0.014) | 0.707(±0.016) | 0.706(±0.012) |
| E → D | 0.737(±0.019) | 0.727(±0.012) | 0.737(±0.013) | 0.732(±0.010) | 0.730(±0.016) | 0.730(±0.011) | 0.739(±0.011) | 0.728(±0.010) |
| E → K | 0.885(±0.008) | 0.862(±0.011) | 0.875(±0.013) | 0.872(±0.009) | 0.882(±0.006) | 0.864(±0.011) | 0.875(±0.011) | 0.877(±0.008) |
| K → B | 0.720(±0.030) | 0.720(±0.006) | 0.735(±0.007) | 0.735(±0.004) | 0.723(±0.027) | 0.719(±0.005) | 0.734(±0.006) | 0.733(±0.003) |
| K → D | 0.742(±0.053) | 0.747(±0.008) | 0.765(±0.009) | 0.768(±0.007) | 0.744(±0.052) | 0.752(±0.007) | 0.764(±0.009) | 0.771(±0.006) |
| K → E | 0.861(±0.001) | 0.847(±0.004) | 0.858(±0.005) | 0.856(±0.002) | 0.865(±0.002) | 0.847(±0.003) | 0.860(±0.004) | 0.852(±0.002) |
| Avg. | 0.787(±0.016) | 0.775(±0.010) | 0.787(±0.010) | 0.784(±0.008) | 0.787(±0.014) | 0.776(±0.009) | 0.787(±0.009) | 0.784(±0.007) |

Table 45: Mean and standard deviation (after ±) of target classification accuracy on Amazon Reviews computed with domain adaptation method CDAN.

| Scenario | Bayesian Model Averaging | | | | Super Learner | | | |
|---|---|---|---|---|---|---|---|---|
| | KuLSIF | Exp | LR | SQ | KuLSIF | Exp | LR | SQ |
| B → D | 0.804(±0.013) | 0.789(±0.011) | 0.803(±0.011) | 0.801(±0.008) | 0.804(±0.012) | 0.791(±0.010) | 0.805(±0.010) | 0.806(±0.008) |
| B → E | 0.790(±0.000) | 0.771(±0.015) | 0.786(±0.016) | 0.785(±0.014) | 0.789(±0.003) | 0.776(±0.014) | 0.785(±0.015) | 0.789(±0.013) |
| B → K | 0.808(±0.012) | 0.792(±0.013) | 0.807(±0.015) | 0.804(±0.011) | 0.812(±0.010) | 0.797(±0.012) | 0.806(±0.014) | 0.808(±0.011) |
| D → B | 0.805(±0.001) | 0.794(±0.003) | 0.802(±0.005) | 0.799(±0.002) | 0.802(±0.002) | 0.796(±0.002) | 0.801(±0.004) | 0.802(±0.001) |
| D → E | 0.800(±0.015) | 0.788(±0.007) | 0.801(±0.009) | 0.789(±0.006) | 0.799(±0.013) | 0.789(±0.007) | 0.803(±0.008) | 0.786(±0.004) |
| D → K | 0.801(±0.057) | 0.788(±0.015) | 0.805(±0.017) | 0.805(±0.014) | 0.803(±0.056) | 0.788(±0.014) | 0.804(±0.016) | 0.809(±0.014) |
| E → B | 0.719(±0.021) | 0.709(±0.021) | 0.719(±0.022) | 0.709(±0.019) | 0.723(±0.020) | 0.713(±0.020) | 0.721(±0.021) | 0.704(±0.018) |
| E → D | 0.749(±0.001) | 0.736(±0.009) | 0.746(±0.009) | 0.747(±0.006) | 0.749(±0.003) | 0.732(±0.008) | 0.746(±0.009) | 0.747(±0.005) |
| E → K | 0.889(±0.006) | 0.868(±0.007) | 0.881(±0.008) | 0.878(±0.006) | 0.892(±0.004) | 0.864(±0.007) | 0.880(±0.007) | 0.876(±0.005) |
| K → B | 0.736(±0.012) | 0.718(±0.017) | 0.732(±0.018) | 0.718(±0.015) | 0.729(±0.010) | 0.720(±0.016) | 0.731(±0.017) | 0.719(±0.014) |
| K → D | 0.756(±0.011) | 0.753(±0.013) | 0.770(±0.015) | 0.762(±0.011) | 0.758(±0.008) | 0.753(±0.012) | 0.772(±0.013) | 0.762(±0.010) |
| K → E | 0.869(±0.000) | 0.852(±0.005) | 0.863(±0.007) | 0.859(±0.004) | 0.868(±0.001) | 0.853(±0.003) | 0.865(±0.006) | 0.859(±0.002) |
| Avg. | 0.794(±0.012) | 0.780(±0.011) | 0.793(±0.013) | 0.788(±0.010) | 0.794(±0.012) | 0.781(±0.010) | 0.793(±0.012) | 0.789(±0.009) |

Table 46: Mean and standard deviation (after ±) of target classification accuracy on Amazon Reviews computed with domain adaptation method DSAN.

| Scenario | Bayesian Model Averaging | | | | Super Learner | | | |
|---|---|---|---|---|---|---|---|---|
| | KuLSIF | Exp | LR | SQ | KuLSIF | Exp | LR | SQ |
| B → D | 0.814(±0.001) | 0.799(±0.005) | 0.811(±0.005) | 0.797(±0.002) | 0.818(±0.001) | 0.804(±0.003) | 0.811(±0.005) | 0.800(±0.001) |
| B → E | 0.783(±0.003) | 0.761(±0.007) | 0.779(±0.008) | 0.779(±0.006) | 0.787(±0.001) | 0.762(±0.006) | 0.781(±0.007) | 0.780(±0.004) |
| B → K | 0.803(±0.009) | 0.782(±0.006) | 0.797(±0.007) | 0.793(±0.005) | 0.807(±0.008) | 0.785(±0.007) | 0.799(±0.007) | 0.791(±0.004) |
| D → B | 0.798(±0.004) | 0.803(±0.008) | 0.811(±0.009) | 0.800(±0.006) | 0.800(±0.003) | 0.804(±0.006) | 0.813(±0.007) | 0.805(±0.005) |
| D → E | 0.787(±0.015) | 0.792(±0.020) | 0.802(±0.022) | 0.793(±0.020) | 0.788(±0.014) | 0.790(±0.020) | 0.801(±0.021) | 0.790(±0.019) |
| D → K | 0.826(±0.019) | 0.809(±0.016) | 0.823(±0.018) | 0.824(±0.014) | 0.830(±0.017) | 0.813(±0.015) | 0.822(±0.017) | 0.820(±0.013) |
| E → B | 0.702(±0.017) | 0.674(±0.018) | 0.685(±0.019) | 0.679(±0.016) | 0.701(±0.014) | 0.670(±0.018) | 0.687(±0.017) | 0.678(±0.015) |
| E → D | 0.725(±0.006) | 0.699(±0.015) | 0.715(±0.016) | 0.707(±0.013) | 0.731(±0.004) | 0.698(±0.015) | 0.717(±0.015) | 0.708(±0.012) |
| E → K | 0.891(±0.005) | 0.869(±0.007) | 0.880(±0.008) | 0.868(±0.005) | 0.893(±0.004) | 0.871(±0.006) | 0.879(±0.007) | 0.874(±0.004) |
| K → B | 0.744(±0.015) | 0.719(±0.010) | 0.731(±0.011) | 0.729(±0.008) | 0.746(±0.013) | 0.721(±0.010) | 0.730(±0.010) | 0.736(±0.008) |
| K → D | 0.740(±0.025) | 0.738(±0.009) | 0.748(±0.010) | 0.741(±0.007) | 0.742(±0.022) | 0.737(±0.008) | 0.749(±0.010) | 0.742(±0.006) |
| K → E | 0.858(±0.017) | 0.860(±0.007) | 0.870(±0.009) | 0.876(±0.007) | 0.854(±0.015) | 0.860(±0.006) | 0.869(±0.009) | 0.882(±0.005) |
| Avg. | 0.789(±0.011) | 0.775(±0.011) | 0.788(±0.012) | 0.782(±0.009) | 0.791(±0.010) | 0.776(±0.010) | 0.788(±0.011) | 0.784(±0.008) |

Table 47: Mean and standard deviation (after ±) of target classification accuracy on Amazon Reviews computed with domain adaptation method DIRT.

| Scenario | Bayesian Model Averaging | | | | Super Learner | | | |
|---|---|---|---|---|---|---|---|---|
| | KuLSIF | Exp | LR | SQ | KuLSIF | Exp | LR | SQ |
| B → D | 0.798(±0.002) | 0.777(±0.004) | 0.786(±0.006) | 0.782(±0.002) | 0.790(±0.000) | 0.778(±0.004) | 0.787(±0.005) | 0.788(±0.002) |
| B → E | 0.761(±0.010) | 0.744(±0.010) | 0.756(±0.011) | 0.750(±0.009) | 0.759(±0.009) | 0.741(±0.009) | 0.758(±0.010) | 0.757(±0.007) |
| B → K | 0.785(±0.015) | 0.769(±0.015) | 0.781(±0.017) | 0.769(±0.014) | 0.785(±0.012) | 0.768(±0.014) | 0.782(±0.015) | 0.771(±0.012) |
| D → B | 0.787(±0.006) | 0.783(±0.004) | 0.794(±0.005) | 0.780(±0.003) | 0.798(±0.005) | 0.786(±0.004) | 0.796(±0.005) | 0.786(±0.002) |
| D → E | 0.787(±0.006) | 0.753(±0.005) | 0.776(±0.005) | 0.765(±0.002) | 0.781(±0.003) | 0.749(±0.003) | 0.776(±0.005) | 0.766(±0.001) |
| D → K | 0.795(±0.010) | 0.772(±0.010) | 0.789(±0.011) | 0.781(±0.009) | 0.797(±0.009) | 0.773(±0.009) | 0.788(±0.011) | 0.784(±0.008) |
| E → B | 0.715(±0.018) | 0.701(±0.017) | 0.706(±0.018) | 0.716(±0.015) | 0.708(±0.016) | 0.703(±0.015) | 0.705(±0.018) | 0.723(±0.015) |
| E → D | 0.729(±0.004) | 0.717(±0.006) | 0.731(±0.007) | 0.732(±0.005) | 0.740(±0.002) | 0.722(±0.005) | 0.730(±0.006) | 0.731(±0.004) |
| E → K | 0.874(±0.008) | 0.858(±0.011) | 0.874(±0.012) | 0.869(±0.008) | 0.883(±0.006) | 0.860(±0.010) | 0.876(±0.011) | 0.872(±0.008) |
| K → B | 0.717(±0.006) | 0.709(±0.005) | 0.723(±0.005) | 0.722(±0.003) | 0.720(±0.004) | 0.714(±0.003) | 0.725(±0.004) | 0.726(±0.001) |
| K → D | 0.750(±0.018) | 0.731(±0.013) | 0.750(±0.014) | 0.741(±0.011) | 0.749(±0.015) | 0.730(±0.012) | 0.749(±0.013) | 0.743(±0.010) |
| K → E | 0.857(±0.005) | 0.845(±0.006) | 0.854(±0.007) | 0.851(±0.004) | 0.860(±0.003) | 0.849(±0.005) | 0.853(±0.007) | 0.853(±0.003) |
| Avg. | 0.780(±0.009) | 0.763(±0.009) | 0.777(±0.010) | 0.772(±0.007) | 0.781(±0.007) | 0.764(±0.008) | 0.777(±0.009) | 0.775(±0.006) |

Table 48: Mean and standard deviation (after ±) of target classification accuracy on Amazon Reviews computed with domain adaptation method AdvSKM.

| Scenario | Bayesian Model Averaging | | | | Super Learner | | | |
|---|---|---|---|---|---|---|---|---|
| | KuLSIF | Exp | LR | SQ | KuLSIF | Exp | LR | SQ |
| B → D | 0.800(±0.003) | 0.778(±0.009) | 0.792(±0.010) | 0.783(±0.007) | 0.796(±0.002) | 0.774(±0.009) | 0.794(±0.009) | 0.787(±0.006) |
| B → E | 0.758(±0.009) | 0.759(±0.018) | 0.756(±0.018) | 0.756(±0.015) | 0.767(±0.007) | 0.760(±0.018) | 0.758(±0.018) | 0.758(±0.015) |
| B → K | 0.785(±0.015) | 0.759(±0.021) | 0.776(±0.022) | 0.774(±0.019) | 0.781(±0.014) | 0.760(±0.020) | 0.778(±0.021) | 0.778(±0.019) |
| D → B | 0.795(±0.002) | 0.776(±0.002) | 0.789(±0.003) | 0.791(±0.001) | 0.792(±0.004) | 0.778(±0.001) | 0.788(±0.002) | 0.795(±0.000) |
| D → E | 0.786(±0.000) | 0.757(±0.003) | 0.772(±0.004) | 0.765(±0.002) | 0.784(±0.002) | 0.754(±0.002) | 0.774(±0.004) | 0.765(±0.001) |
| D → K | 0.794(±0.012) | 0.769(±0.014) | 0.788(±0.015) | 0.785(±0.013) | 0.787(±0.009) | 0.774(±0.013) | 0.789(±0.015) | 0.789(±0.011) |
| E → B | 0.701(±0.011) | 0.679(±0.013) | 0.693(±0.014) | 0.685(±0.012) | 0.704(±0.008) | 0.679(±0.012) | 0.695(±0.013) | 0.691(±0.011) |
| E → D | 0.738(±0.008) | 0.717(±0.010) | 0.732(±0.010) | 0.726(±0.007) | 0.737(±0.006) | 0.718(±0.008) | 0.734(±0.010) | 0.724(±0.006) |
| E → K | 0.873(±0.006) | 0.867(±0.007) | 0.872(±0.008) | 0.873(±0.005) | 0.881(±0.003) | 0.863(±0.005) | 0.873(±0.008) | 0.871(±0.004) |
| K → B | 0.720(±0.003) | 0.701(±0.012) | 0.717(±0.013) | 0.703(±0.010) | 0.721(±0.001) | 0.706(±0.012) | 0.716(±0.012) | 0.705(±0.009) |
| K → D | 0.745(±0.009) | 0.736(±0.008) | 0.752(±0.009) | 0.742(±0.007) | 0.743(±0.007) | 0.738(±0.007) | 0.754(±0.008) | 0.740(±0.006) |
| K → E | 0.862(±0.003) | 0.838(±0.007) | 0.854(±0.007) | 0.849(±0.006) | 0.852(±0.001) | 0.840(±0.006) | 0.853(±0.006) | 0.847(±0.005) |
| Avg. | 0.780(±0.007) | 0.761(±0.010) | 0.774(±0.011) | 0.769(±0.009) | 0.779(±0.005) | 0.762(±0.009) | 0.775(±0.010) | 0.771(±0.008) |

Table 49: Mean and standard deviation (after ±) of target classification accuracy on Amazon Reviews computed with domain adaptation method HoMM.

| Scenario | Bayesian Model Averaging | | | | Super Learner | | | |
|---|---|---|---|---|---|---|---|---|
| | KuLSIF | Exp | LR | SQ | KuLSIF | Exp | LR | SQ |
| B → D | 0.796(±0.006) | 0.772(±0.006) | 0.790(±0.008) | 0.788(±0.005) | 0.794(±0.005) | 0.773(±0.006) | 0.789(±0.008) | 0.792(±0.005) |
| B → E | 0.760(±0.011) | 0.740(±0.015) | 0.760(±0.016) | 0.751(±0.013) | 0.764(±0.010) | 0.738(±0.013) | 0.759(±0.016) | 0.752(±0.013) |
| B → K | 0.790(±0.011) | 0.770(±0.020) | 0.779(±0.022) | 0.768(±0.012) | 0.790(±0.009) | 0.770(±0.019) | 0.781(±0.021) | 0.766(±0.018) |
| D → B | 0.798(±0.002) | 0.777(±0.006) | 0.791(±0.008) | 0.787(±0.005) | 0.793(±0.001) | 0.773(±0.005) | 0.793(±0.007) | 0.791(±0.004) |
| D → E | 0.780(±0.001) | 0.765(±0.007) | 0.778(±0.008) | 0.766(±0.005) | 0.791(±0.003) | 0.768(±0.006) | 0.780(±0.007) | 0.764(±0.003) |
| D → K | 0.784(±0.025) | 0.776(±0.014) | 0.790(±0.015) | 0.779(±0.012) | 0.782(±0.023) | 0.778(±0.014) | 0.792(±0.014) | 0.786(±0.013) |
| E → B | 0.708(±0.018) | 0.697(±0.023) | 0.699(±0.024) | 0.694(±0.021) | 0.714(±0.016) | 0.701(±0.022) | 0.698(±0.022) | 0.696(±0.020) |
| E → D | 0.737(±0.002) | 0.730(±0.007) | 0.740(±0.007) | 0.737(±0.005) | 0.735(±0.001) | 0.728(±0.006) | 0.739(±0.006) | 0.738(±0.004) |
| E → K | 0.878(±0.007) | 0.871(±0.007) | 0.874(±0.009) | 0.858(±0.006) | 0.884(±0.006) | 0.873(±0.006) | 0.876(±0.007) | 0.863(±0.005) |
| K → B | 0.725(±0.000) | 0.726(±0.005) | 0.722(±0.007) | 0.718(±0.004) | 0.733(±0.002) | 0.728(±0.004) | 0.724(±0.006) | 0.718(±0.003) |
| K → D | 0.751(±0.011) | 0.738(±0.008) | 0.752(±0.009) | 0.750(±0.006) | 0.753(±0.010) | 0.739(±0.007) | 0.752(±0.008) | 0.753(±0.005) |
| K → E | 0.860(±0.005) | 0.848(±0.007) | 0.856(±0.007) | 0.857(±0.005) | 0.855(±0.004) | 0.844(±0.006) | 0.857(±0.007) | 0.860(±0.003) |
| Avg. | 0.781(±0.008) | 0.767(±0.010) | 0.778(±0.012) | 0.771(±0.009) | 0.782(±0.007) | 0.768(±0.009) | 0.778(±0.011) | 0.773(±0.008) |

Table 50: Mean and standard deviation (after ±) of target classification accuracy on Amazon Reviews computed with domain adaptation method DDC.

| Scenario | Bayesian Model Averaging | | | | Super Learner | | | |
|---|---|---|---|---|---|---|---|---|
| | KuLSIF | Exp | LR | SQ | KuLSIF | Exp | LR | SQ |
| B → D | 0.805(±0.007) | 0.788(±0.009) | 0.799(±0.011) | 0.789(±0.008) | 0.802(±0.006) | 0.787(±0.009) | 0.801(±0.010) | 0.787(±0.007) |
| B → E | 0.778(±0.004) | 0.737(±0.012) | 0.763(±0.012) | 0.761(±0.009) | 0.777(±0.002) | 0.742(±0.011) | 0.762(±0.012) | 0.763(±0.009) |
| B → K | 0.790(±0.012) | 0.776(±0.017) | 0.781(±0.018) | 0.770(±0.015) | 0.785(±0.009) | 0.772(±0.015) | 0.783(±0.017) | 0.773(±0.013) |
| D → B | 0.801(±0.001) | 0.778(±0.002) | 0.794(±0.003) | 0.783(±0.000) | 0.805(±0.001) | 0.777(±0.002) | 0.793(±0.001) | 0.787(±0.001) |
| D → E | 0.790(±0.002) | 0.761(±0.010) | 0.779(±0.011) | 0.776(±0.008) | 0.792(±0.000) | 0.761(±0.009) | 0.778(±0.009) | 0.780(±0.007) |
| D → K | 0.791(±0.013) | 0.775(±0.016) | 0.791(±0.016) | 0.788(±0.013) | 0.793(±0.012) | 0.773(±0.015) | 0.793(±0.016) | 0.788(±0.013) |
| E → B | 0.711(±0.022) | 0.702(±0.022) | 0.709(±0.024) | 0.711(±0.021) | 0.713(±0.020) | 0.706(±0.022) | 0.711(±0.023) | 0.718(±0.020) |
| E → D | 0.738(±0.001) | 0.717(±0.002) | 0.739(±0.003) | 0.736(±0.000) | 0.746(±0.000) | 0.722(±0.002) | 0.741(±0.002) | 0.743(±0.001) |
| E → K | 0.882(±0.004) | 0.854(±0.008) | 0.876(±0.009) | 0.864(±0.007) | 0.886(±0.002) | 0.857(±0.007) | 0.878(±0.008) | 0.862(±0.007) |
| K → B | 0.737(±0.003) | 0.724(±0.007) | 0.735(±0.008) | 0.725(±0.004) | 0.743(±0.001) | 0.728(±0.006) | 0.734(±0.006) | 0.727(±0.004) |
| K → D | 0.760(±0.013) | 0.740(±0.015) | 0.749(±0.017) | 0.751(±0.014) | 0.759(±0.011) | 0.741(±0.014) | 0.751(±0.016) | 0.758(±0.013) |
| K → E | 0.862(±0.006) | 0.837(±0.004) | 0.855(±0.005) | 0.855(±0.002) | 0.861(±0.006) | 0.833(±0.003) | 0.857(±0.004) | 0.856(±0.000) |
| Avg. | 0.787(±0.008) | 0.766(±0.010) | 0.781(±0.011) | 0.776(±0.008) | 0.788(±0.006) | 0.767(±0.010) | 0.782(±0.010) | 0.778(±0.008) |

Table 51: Mean and standard deviation (after ±) of target classification accuracy on Amazon Reviews computed with domain adaptation method DeepCoral.

| Scenario | Bayesian Model Averaging | | | | Super Learner | | | |
|---|---|---|---|---|---|---|---|---|
| | KuLSIF | Exp | LR | SQ | KuLSIF | Exp | LR | SQ |
| B → D | 0.800(±0.005) | 0.776(±0.009) | 0.794(±0.009) | 0.791(±0.006) | 0.798(±0.004) | 0.775(±0.009) | 0.796(±0.008) | 0.798(±0.005) |
| B → E | 0.780(±0.019) | 0.746(±0.020) | 0.761(±0.021) | 0.754(±0.018) | 0.770(±0.017) | 0.750(±0.019) | 0.761(±0.020) | 0.761(±0.017) |
| B → K | 0.748(±0.069) | 0.778(±0.010) | 0.793(±0.011) | 0.795(±0.008) | 0.748(±0.067) | 0.780(±0.010) | 0.795(±0.010) | 0.796(±0.006) |
| D → B | 0.789(±0.017) | 0.797(±0.006) | 0.802(±0.007) | 0.796(±0.003) | 0.797(±0.015) | 0.799(±0.005) | 0.804(±0.006) | 0.796(±0.002) |
| D → E | 0.779(±0.015) | 0.790(±0.009) | 0.802(±0.010) | 0.794(±0.007) | 0.787(±0.013) | 0.792(±0.008) | 0.804(±0.010) | 0.793(±0.006) |
| D → K | 0.796(±0.028) | 0.786(±0.014) | 0.804(±0.015) | 0.808(±0.012) | 0.793(±0.025) | 0.791(±0.013) | 0.806(±0.014) | 0.803(±0.010) |
| E → B | 0.716(±0.012) | 0.712(±0.016) | 0.721(±0.017) | 0.726(±0.014) | 0.723(±0.010) | 0.717(±0.015) | 0.720(±0.016) | 0.726(±0.013) |
| E → D | 0.746(±0.018) | 0.737(±0.008) | 0.744(±0.008) | 0.746(±0.005) | 0.747(±0.016) | 0.741(±0.006) | 0.746(±0.007) | 0.750(±0.005) |
| E → K | 0.876(±0.026) | 0.852(±0.008) | 0.875(±0.010) | 0.865(±0.008) | 0.881(±0.023) | 0.857(±0.007) | 0.874(±0.008) | 0.863(±0.007) |
| K → B | 0.741(±0.004) | 0.717(±0.004) | 0.731(±0.004) | 0.723(±0.002) | 0.747(±0.000) | 0.713(±0.003) | 0.733(±0.003) | 0.726(±0.000) |
| K → D | 0.782(±0.022) | 0.721(±0.030) | 0.740(±0.031) | 0.732(±0.027) | 0.780(±0.019) | 0.718(±0.030) | 0.739(±0.030) | 0.737(±0.027) |
| K → E | 0.859(±0.010) | 0.847(±0.004) | 0.867(±0.006) | 0.853(±0.003) | 0.864(±0.008) | 0.846(±0.004) | 0.869(±0.006) | 0.860(±0.003) |
| Avg. | 0.784(±0.020) | 0.772(±0.012) | 0.786(±0.012) | 0.782(±0.009) | 0.786(±0.018) | 0.773(±0.011) | 0.787(±0.011) | 0.784(±0.008) |

Table 52: Mean and standard deviation (after ±) of target classification accuracy on Amazon Reviews computed with domain adaptation method CMD.

| Scenario | Cross-Validation for Binary Classifier | | | | Aggregation | | | |
|---|---|---|---|---|---|---|---|---|
| | KuLSIF | Exp | LR | SQ | KuLSIF | Exp | LR | SQ |
| 0 → 6 | 0.746(±0.008) | 0.701(±0.031) | 0.716(±0.026) | 0.678(±0.012) | **0.793(±0.022)** | **0.740(±0.001)** | **0.728(±0.019)** | **0.702(±0.011)** |
| 1 → 6 | 0.897(±0.002) | 0.864(±0.009) | 0.867(±0.006) | 0.829(±0.008) | **0.953(±0.029)** | **0.906(±0.022)** | **0.891(±0.000)** | **0.858(±0.007)** |
| 2 → 7 | 0.488(±0.010) | 0.484(±0.009) | 0.493(±0.009) | 0.460(±0.005) | **0.532(±0.021)** | **0.528(±0.022)** | **0.494(±0.003)** | **0.475(±0.004)** |
| 3 → 8 | 0.839(±0.012) | 0.845(±0.032) | 0.864(±0.012) | 0.826(±0.002) | **0.877(±0.019)** | **0.883(±0.000)** | **0.877(±0.006)** | **0.859(±0.001)** |
| 4 → 5 | 0.928(±0.006) | 0.456(±0.410) | 0.923(±0.012) | 0.888(±0.002) | **0.975(±0.025)** | **0.497(±0.379)** | **0.938(±0.006)** | **0.920(±0.001)** |
| Avg. | 0.780(±0.008) | 0.670(±0.098) | 0.773(±0.013) | 0.736(±0.006) | **0.826(±0.023)** | **0.711(±0.085)** | **0.786(±0.007)** | **0.763(±0.005)** |

Table 53: Mean and standard deviation (after ±) of target classification accuracy on HHAR computed with domain adaptation method MMDA.

| Scenario | Cross-Validation for Binary Classifier | | | | Aggregation | | | |
|---|---|---|---|---|---|---|---|---|
| | KuLSIF | Exp | LR | SQ | KuLSIF | Exp | LR | SQ |
| $0 \to 6$ | 0.596($\pm$0.048) | 0.634($\pm$0.045) | 0.631($\pm$0.043) | 0.593($\pm$0.029) | **0.638($\pm$0.017)** | **0.674($\pm$0.012)** | **0.643($\pm$0.038)** | **0.620($\pm$0.028)** |
| $1 \to 6$ | 0.939($\pm$0.006) | 0.908($\pm$0.011) | 0.906($\pm$0.013) | 0.868($\pm$0.001) | **0.982($\pm$0.024)** | **0.955($\pm$0.019)** | **0.919($\pm$0.006)** | **0.896($\pm$0.000)** |
| $2 \to 7$ | 0.472($\pm$0.005) | 0.470($\pm$0.008) | 0.476($\pm$0.008) | 0.438($\pm$0.006) | **0.516($\pm$0.025)** | **0.511($\pm$0.024)** | **0.493($\pm$0.002)** | **0.464($\pm$0.005)** |
| $3 \to 8$ | 0.960($\pm$0.030) | 0.921($\pm$0.028) | 0.934($\pm$0.030) | 0.896($\pm$0.016) | **1.005($\pm$0.002)** | **0.964($\pm$0.005)** | **0.954($\pm$0.024)** | **0.928($\pm$0.014)** |
| $4 \to 5$ | 0.648($\pm$0.562) | 0.947($\pm$0.013) | 0.947($\pm$0.012) | 0.909($\pm$0.002) | **0.682($\pm$0.530)** | **0.984($\pm$0.018)** | **0.958($\pm$0.006)** | **0.925($\pm$0.002)** |
| Avg. | 0.723($\pm$0.130) | 0.776($\pm$0.021) | 0.779($\pm$0.021) | 0.741($\pm$0.011) | **0.765($\pm$0.120)** | **0.818($\pm$0.016)** | **0.793($\pm$0.015)** | **0.767($\pm$0.010)** |

Table 54: Mean and standard deviation (after $\pm$) of target classification accuracy on HHAR computed with domain adaptation method CoDATS.

| Scenario | Cross-Validation for Binary Classifier | | | | Aggregation | | | |
|---|---|---|---|---|---|---|---|---|
| | KuLSIF | Exp | LR | SQ | KuLSIF | Exp | LR | SQ |
| $0 \to 6$ | 0.604($\pm$0.036) | 0.652($\pm$0.043) | 0.657($\pm$0.027) | 0.619($\pm$0.013) | **0.647($\pm$0.006)** | **0.690($\pm$0.010)** | **0.669($\pm$0.021)** | **0.637($\pm$0.012)** |
| $1 \to 6$ | 0.936($\pm$0.002) | 0.890($\pm$0.038) | 0.906($\pm$0.010) | 0.868($\pm$0.004) | **0.977($\pm$0.028)** | **0.935($\pm$0.006)** | **0.916($\pm$0.004)** | **0.891($\pm$0.002)** |
| $2 \to 7$ | 0.327($\pm$0.284) | 0.511($\pm$0.026) | 0.533($\pm$0.045) | 0.495($\pm$0.031) | **0.367($\pm$0.253)** | **0.552($\pm$0.006)** | **0.541($\pm$0.038)** | **0.513($\pm$0.030)** |
| $3 \to 8$ | 0.964($\pm$0.006) | 0.919($\pm$0.007) | 0.923($\pm$0.015) | 0.885($\pm$0.001) | **1.005($\pm$0.026)** | **0.964($\pm$0.026)** | **0.939($\pm$0.008)** | **0.900($\pm$0.001)** |
| $4 \to 5$ | 0.654($\pm$0.566) | 0.951($\pm$0.005) | 0.957($\pm$0.010) | 0.919($\pm$0.004) | **0.696($\pm$0.536)** | **0.998($\pm$0.026)** | **0.969($\pm$0.004)** | **0.960($\pm$0.003)** |
| Avg. | 0.697($\pm$0.179) | 0.785($\pm$0.024) | 0.795($\pm$0.021) | 0.757($\pm$0.011) | **0.738($\pm$0.170)** | **0.828($\pm$0.015)** | **0.807($\pm$0.015)** | **0.780($\pm$0.010)** |

Table 55: Mean and standard deviation (after $\pm$) of target classification accuracy on HHAR computed with domain adaptation method DANN.

| Scenario | Cross-Validation for Binary Classifier | | | | Aggregation | | | |
|---|---|---|---|---|---|---|---|---|
| | KuLSIF | Exp | LR | SQ | KuLSIF | Exp | LR | SQ |
| $0 \to 6$ | 0.622($\pm$0.009) | 0.507($\pm$0.281) | 0.666($\pm$0.014) | 0.628($\pm$0.000) | **0.660($\pm$0.022)** | **0.558($\pm$0.249)** | **0.685($\pm$0.008)** | **0.658($\pm$0.002)** |
| $1 \to 6$ | 0.933($\pm$0.000) | 0.906($\pm$0.008) | 0.908($\pm$0.006) | 0.870($\pm$0.008) | **0.982($\pm$0.031)** | **0.940($\pm$0.025)** | **0.922($\pm$0.001)** | **0.899($\pm$0.007)** |
| $2 \to 7$ | 0.550($\pm$0.061) | 0.547($\pm$0.061) | 0.552($\pm$0.066) | 0.514($\pm$0.052) | **0.592($\pm$0.030)** | **0.592($\pm$0.029)** | **0.562($\pm$0.060)** | **0.533($\pm$0.050)** |
| $3 \to 8$ | 0.874($\pm$0.076) | 0.863($\pm$0.059) | 0.862($\pm$0.056) | 0.829($\pm$0.042) | **0.911($\pm$0.043)** | **0.910($\pm$0.028)** | **0.878($\pm$0.050)** | **0.850($\pm$0.041)** |
| $4 \to 5$ | 0.980($\pm$0.000) | 0.707($\pm$0.423) | 0.954($\pm$0.006) | 0.916($\pm$0.008) | **1.025($\pm$0.031)** | **0.749($\pm$0.391)** | **0.972($\pm$0.001)** | **0.935($\pm$0.007)** |
| Avg. | 0.792($\pm$0.029) | 0.706($\pm$0.166) | 0.788($\pm$0.029) | 0.751($\pm$0.022) | **0.834($\pm$0.031)** | **0.750($\pm$0.144)** | **0.804($\pm$0.024)** | **0.775($\pm$0.021)** |

Table 56: Mean and standard deviation (after $\pm$) of target classification accuracy on HHAR computed with domain adaptation method CDAN.

| Scenario | Cross-Validation for Binary Classifier | | | | Aggregation | | | |
|---|---|---|---|---|---|---|---|---|
| | KuLSIF | Exp | LR | SQ | KuLSIF | Exp | LR | SQ |
| $0 \to 6$ | 0.719($\pm$0.070) | 0.350($\pm$0.289) | 0.705($\pm$0.043) | 0.667($\pm$0.029) | **0.753($\pm$0.039)** | **0.391($\pm$0.257)** | **0.720($\pm$0.038)** | **0.686($\pm$0.028)** |
| $1 \to 6$ | 0.929($\pm$0.000) | 0.410($\pm$0.429) | 0.893($\pm$0.023) | 0.855($\pm$0.009) | **0.975($\pm$0.031)** | **0.464($\pm$0.396)** | **0.911($\pm$0.018)** | **0.870($\pm$0.008)** |
| $2 \to 7$ | 0.496($\pm$0.000) | 0.495($\pm$0.002) | 0.495($\pm$0.001) | 0.457($\pm$0.013) | **0.538($\pm$0.031)** | **0.538($\pm$0.030)** | **0.509($\pm$0.006)** | **0.482($\pm$0.012)** |
| $3 \to 8$ | 0.971($\pm$0.005) | 0.926($\pm$0.005) | 0.927($\pm$0.005) | 0.889($\pm$0.009) | **1.009($\pm$0.027)** | **0.965($\pm$0.028)** | **0.937($\pm$0.001)** | **0.913($\pm$0.008)** |
| $4 \to 5$ | 0.654($\pm$0.566) | 0.460($\pm$0.417) | 0.939($\pm$0.005) | 0.901($\pm$0.009) | **0.694($\pm$0.535)** | **0.503($\pm$0.385)** | **0.947($\pm$0.001)** | **0.925($\pm$0.008)** |
| Avg. | 0.754($\pm$0.128) | 0.528($\pm$0.228) | 0.792($\pm$0.015) | 0.754($\pm$0.014) | **0.794($\pm$0.133)** | **0.572($\pm$0.219)** | **0.805($\pm$0.013)** | **0.775($\pm$0.013)** |

Table 57: Mean and standard deviation (after $\pm$) of target classification accuracy on HHAR computed with domain adaptation method DSAN.

| Scenario | Cross-Validation for Binary Classifier | | | | Aggregation | | | |
|---|---|---|---|---|---|---|---|---|
| | KuLSIF | Exp | LR | SQ | KuLSIF | Exp | LR | SQ |
| $0 \to 6$ | 0.693($\pm$0.072) | 0.694($\pm$0.050) | 0.709($\pm$0.043) | 0.671($\pm$0.029) | **0.737($\pm$0.041)** | **0.729($\pm$0.018)** | **0.729($\pm$0.037)** | **0.701($\pm$0.027)** |
| $1 \to 6$ | 0.938($\pm$0.000) | 0.884($\pm$0.029) | 0.904($\pm$0.010) | 0.866($\pm$0.004) | **0.985($\pm$0.031)** | **0.923($\pm$0.004)** | **0.909($\pm$0.004)** | **0.887($\pm$0.004)** |
| $2 \to 7$ | 0.194($\pm$0.336) | 0.582($\pm$0.090) | 0.545($\pm$0.041) | 0.507($\pm$0.027) | **0.234($\pm$0.306)** | **0.620($\pm$0.058)** | **0.548($\pm$0.036)** | **0.541($\pm$0.026)** |
| $3 \to 8$ | 0.848($\pm$0.007) | 0.839($\pm$0.022) | 0.842($\pm$0.011) | 0.804($\pm$0.003) | **0.887($\pm$0.024)** | **0.879($\pm$0.011)** | **0.856($\pm$0.006)** | **0.823($\pm$0.002)** |
| $4 \to 5$ | 0.984($\pm$0.004) | 0.949($\pm$0.008) | 0.951($\pm$0.009) | 0.919($\pm$0.005) | **1.026($\pm$0.026)** | **0.989($\pm$0.024)** | **0.964($\pm$0.002)** | **0.941($\pm$0.004)** |
| Avg. | 0.731($\pm$0.084) | 0.790($\pm$0.040) | 0.790($\pm$0.023) | 0.753($\pm$0.014) | **0.774($\pm$0.086)** | **0.828($\pm$0.023)** | **0.801($\pm$0.017)** | **0.779($\pm$0.013)** |

Table 58: Mean and standard deviation (after $\pm$) of target classification accuracy on HHAR computed with domain adaptation method DIRT.

| Scenario | Cross-Validation for Binary Classifier | | | | Aggregation | | | |
|---|---|---|---|---|---|---|---|---|
| | KuLSIF | Exp | LR | SQ | KuLSIF | Exp | LR | SQ |
| $0 \to 6$ | 0.718($\pm$0.009) | 0.714($\pm$0.010) | 0.705($\pm$0.020) | 0.667($\pm$0.006) | **0.766($\pm$0.022)** | **0.755($\pm$0.022)** | **0.721($\pm$0.014)** | **0.690($\pm$0.005)** |
| $1 \to 6$ | 0.857($\pm$0.006) | 0.835($\pm$0.026) | 0.836($\pm$0.025) | 0.798($\pm$0.011) | **0.893($\pm$0.025)** | **0.894($\pm$0.007)** | **0.840($\pm$0.019)** | **0.823($\pm$0.010)** |
| $2 \to 7$ | 0.490($\pm$0.014) | 0.497($\pm$0.010) | 0.502($\pm$0.024) | 0.464($\pm$0.010) | **0.526($\pm$0.017)** | **0.541($\pm$0.022)** | **0.519($\pm$0.018)** | **0.492($\pm$0.008)** |
| $3 \to 8$ | 0.810($\pm$0.002) | 0.806($\pm$0.005) | 0.807($\pm$0.010) | 0.769($\pm$0.009) | **0.856($\pm$0.029)** | **0.840($\pm$0.027)** | **0.819($\pm$0.001)** | **0.796($\pm$0.008)** |
| $4 \to 5$ | 0.884($\pm$0.005) | 0.875($\pm$0.014) | 0.878($\pm$0.017) | 0.840($\pm$0.013) | **0.924($\pm$0.026)** | **0.921($\pm$0.019)** | **0.897($\pm$0.011)** | **0.874($\pm$0.002)** |
| Avg. | 0.752($\pm$0.007) | 0.746($\pm$0.013) | 0.746($\pm$0.018) | 0.708($\pm$0.008) | **0.793($\pm$0.024)** | **0.790($\pm$0.019)** | **0.759($\pm$0.012)** | **0.735($\pm$0.006)** |

Table 59: Mean and standard deviation (after $\pm$) of target classification accuracy on HHAR computed with domain adaptation method AdvSKM.

| Scenario | Cross-Validation for Binary Classifier | | | | Aggregation | | | |
|---|---|---|---|---|---|---|---|---|
| | KuLSIF | Exp | LR | SQ | KuLSIF | Exp | LR | SQ |
| 0 → 6 | 0.732(±0.010) | 0.719(±0.025) | 0.725(±0.024) | 0.687(±0.010) | **0.782**(±**0.021**) | 0.764(±0.007) | 0.742(±0.018) | 0.721(±0.008) |
| 1 → 6 | 0.879(±0.011) | 0.856(±0.019) | 0.853(±0.024) | 0.815(±0.010) | **0.917**(±**0.020**) | 0.899(±0.013) | 0.866(±0.017) | 0.845(±0.009) |
| 2 → 7 | 0.455(±0.015) | 0.466(±0.029) | 0.474(±0.024) | 0.436(±0.010) | 0.499(±0.016) | **0.515**(±**0.002**) | 0.491(±0.018) | 0.466(±0.009) |
| 3 → 8 | 0.818(±0.005) | 0.820(±0.008) | 0.822(±0.004) | 0.784(±0.010) | 0.859(±0.026) | **0.863**(±**0.024**) | 0.835(±0.002) | 0.820(±0.009) |
| 4 → 5 | 0.911(±0.005) | 0.449(±0.399) | 0.899(±0.031) | 0.861(±0.017) | **0.958**(±**0.026**) | 0.496(±0.366) | 0.905(±0.025) | 0.894(±0.016) |
| Avg. | 0.759(±0.009) | 0.662(±0.096) | 0.754(±0.021) | 0.716(±0.011) | **0.803**(±**0.022**) | 0.707(±0.083) | 0.768(±0.016) | 0.749(±0.010) |

Table 60: Mean and standard deviation (after ±) of target classification accuracy on HHAR computed with domain adaptation method HoMM.

| Scenario | Cross-Validation for Binary Classifier | | | | Aggregation | | | |
|---|---|---|---|---|---|---|---|---|
| | KuLSIF | Exp | LR | SQ | KuLSIF | Exp | LR | SQ |
| 0 → 6 | 0.697(±0.013) | 0.349(±0.286) | 0.690(±0.011) | 0.652(±0.003) | **0.748**(±**0.018**) | 0.397(±0.255) | 0.709(±0.005) | 0.680(±0.003) |
| 1 → 6 | 0.882(±0.017) | 0.639(±0.413) | 0.873(±0.014) | 0.835(±0.000) | **0.924**(±**0.014**) | 0.677(±0.381) | 0.886(±0.009) | 0.864(±0.001) |
| 2 → 7 | 0.439(±0.041) | 0.440(±0.036) | 0.439(±0.037) | 0.401(±0.023) | **0.486**(±**0.010**) | 0.476(±0.004) | 0.452(±0.032) | 0.422(±0.023) |
| 3 → 8 | 0.818(±0.008) | 0.621(±0.342) | 0.823(±0.006) | 0.785(±0.008) | **0.858**(±**0.022**) | 0.663(±0.310) | 0.840(±0.001) | 0.813(±0.007) |
| 4 → 5 | 0.905(±0.008) | 0.219(±0.000) | 0.895(±0.025) | 0.857(±0.011) | **0.955**(±**0.023**) | 0.266(±0.032) | 0.908(±0.019) | 0.879(±0.010) |
| Avg. | 0.748(±0.017) | 0.454(±0.215) | 0.744(±0.019) | 0.706(±0.009) | **0.794**(±**0.018**) | 0.496(±0.196) | 0.759(±0.013) | 0.732(±0.009) |

Table 61: Mean and standard deviation (after ±) of target classification accuracy on HHAR computed with domain adaptation method DDC.

| Scenario | Cross-Validation for Binary Classifier | | | | Aggregation | | | |
|---|---|---|---|---|---|---|---|---|
| | KuLSIF | Exp | LR | SQ | KuLSIF | Exp | LR | SQ |
| 0 → 6 | 0.728(±0.019) | 0.700(±0.032) | 0.717(±0.025) | 0.679(±0.011) | **0.768**(±**0.012**) | 0.739(±0.001) | 0.733(±0.019) | 0.694(±0.010) |
| 1 → 6 | 0.886(±0.009) | 0.845(±0.039) | 0.861(±0.014) | 0.823(±0.000) | **0.924**(±**0.022**) | 0.890(±0.008) | 0.865(±0.009) | 0.842(±0.001) |
| 2 → 7 | 0.460(±0.012) | 0.470(±0.014) | 0.480(±0.020) | 0.442(±0.006) | 0.509(±0.019) | **0.512**(±**0.017**) | 0.488(±0.015) | 0.462(±0.005) |
| 3 → 8 | 0.812(±0.000) | 0.810(±0.003) | 0.811(±0.001) | 0.773(±0.013) | 0.849(±0.031) | **0.852**(±**0.029**) | 0.832(±0.004) | 0.804(±0.011) |
| 4 → 5 | 0.618(±0.536) | 0.919(±0.020) | 0.923(±0.016) | 0.885(±0.002) | 0.672(±0.504) | **0.963**(±**0.012**) | 0.929(±0.009) | 0.911(±0.000) |
| Avg. | 0.701(±0.115) | 0.749(±0.021) | 0.758(±0.015) | 0.720(±0.006) | 0.745(±0.118) | **0.791**(±**0.013**) | 0.769(±0.011) | 0.743(±0.005) |

Table 62: Mean and standard deviation (after ±) of target classification accuracy on HHAR computed with domain adaptation method Deep Coral.

| Scenario | Cross-Validation for Binary Classifier | | | | Aggregation | | | |
|---|---|---|---|---|---|---|---|---|
| | KuLSIF | Exp | LR | SQ | KuLSIF | Exp | LR | SQ |
| 0 → 6 | 0.693(±0.012) | 0.713(±0.012) | 0.693(±0.010) | 0.655(±0.004) | 0.739(±0.019) | **0.757**(±**0.020**) | 0.706(±0.004) | 0.667(±0.002) |
| 1 → 6 | 0.907(±0.006) | 0.885(±0.018) | 0.896(±0.008) | 0.859(±0.006) | **0.947**(±**0.024**) | 0.924(±0.015) | 0.904(±0.001) | 0.891(±0.005) |
| 2 → 7 | 0.320(±0.277) | 0.503(±0.027) | 0.513(±0.040) | 0.475(±0.026) | 0.370(±0.247) | **0.542**(±**0.003**) | 0.523(±0.035) | 0.493(±0.024) |
| 3 → 8 | 0.811(±0.011) | 0.813(±0.009) | 0.813(±0.011) | 0.775(±0.003) | 0.856(±0.019) | **0.860**(±**0.023**) | 0.825(±0.005) | 0.797(±0.002) |
| 4 → 5 | 0.625(±0.541) | 0.937(±0.011) | 0.934(±0.013) | 0.896(±0.001) | 0.660(±0.511) | **0.989**(±**0.021**) | 0.945(±0.007) | 0.921(±0.001) |
| Avg. | 0.671(±0.170) | 0.770(±0.015) | 0.770(±0.016) | 0.732(±0.008) | 0.714(±0.164) | **0.814**(±**0.017**) | 0.780(±0.010) | 0.754(±0.007) |

Table 63: Mean and standard deviation (after ±) of target classification accuracy on HHAR computed with domain adaptation method CMD.

| Scenario | Bayesian Model Averaging | | | | Super Learner | | | |
|---|---|---|---|---|---|---|---|---|
| | KuLSIF | Exp | LR | SQ | KuLSIF | Exp | LR | SQ |
| 0 → 6 | 0.748(±0.005) | 0.714(±0.030) | 0.716(±0.024) | 0.681(±0.014) | 0.754(±0.003) | 0.727(±0.020) | 0.715(±0.020) | 0.686(±0.011) |
| 1 → 6 | 0.899(±0.002) | 0.880(±0.009) | 0.867(±0.005) | 0.837(±0.010) | 0.899(±0.002) | 0.891(±0.001) | 0.871(±0.002) | 0.838(±0.006) |
| 2 → 7 | 0.489(±0.006) | 0.502(±0.009) | 0.493(±0.008) | 0.456(±0.007) | 0.490(±0.005) | 0.511(±0.001) | 0.493(±0.004) | 0.473(±0.004) |
| 3 → 8 | 0.838(±0.008) | 0.857(±0.031) | 0.864(±0.011) | 0.834(±0.004) | 0.842(±0.007) | 0.875(±0.020) | 0.866(±0.007) | 0.841(±0.001) |
| 4 → 5 | 0.930(±0.002) | 0.471(±0.410) | 0.923(±0.011) | 0.896(±0.004) | 0.935(±0.002) | 0.474(±0.399) | 0.921(±0.008) | 0.895(±0.000) |
| Avg. | 0.781(±0.004) | 0.685(±0.098) | 0.773(±0.012) | 0.741(±0.008) | 0.784(±0.004) | 0.696(±0.088) | 0.773(±0.008) | 0.747(±0.005) |

Table 64: Mean and standard deviation (after ±) of target classification accuracy on HHAR computed with domain adaptation method MMDA.

| Scenario | Bayesian Model Averaging | | | | Super Learner | | | |
|---|---|---|---|---|---|---|---|---|
| | KuLSIF | Exp | LR | SQ | KuLSIF | Exp | LR | SQ |
| 0 → 6 | 0.598(±0.043) | 0.650(±0.044) | 0.631(±0.041) | 0.596(±0.031) | 0.593(±0.042) | 0.665(±0.034) | 0.629(±0.038) | 0.610(±0.028) |
| 1 → 6 | 0.941(±0.001) | 0.922(±0.011) | 0.906(±0.011) | 0.870(±0.003) | 0.942(±0.000) | 0.936(±0.000) | 0.910(±0.008) | 0.868(±0.000) |
| 2 → 7 | 0.471(±0.001) | 0.483(±0.008) | 0.476(±0.008) | 0.436(±0.008) | 0.468(±0.000) | 0.497(±0.003) | 0.475(±0.003) | 0.449(±0.005) |
| 3 → 8 | 0.962(±0.026) | 0.935(±0.026) | 0.934(±0.028) | 0.903(±0.018) | 0.956(±0.024) | 0.938(±0.016) | 0.932(±0.023) | 0.909(±0.015) |
| 4 → 5 | 0.648(±0.558) | 0.962(±0.013) | 0.947(±0.012) | 0.915(±0.004) | 0.656(±0.556) | 0.978(±0.002) | 0.950(±0.006) | 0.919(±0.001) |
| Avg. | 0.724(±0.126) | 0.790(±0.020) | 0.779(±0.020) | 0.744(±0.013) | 0.723(±0.125) | 0.803(±0.011) | 0.779(±0.015) | 0.751(±0.010) |

Table 65: Mean and standard deviation (after ±) of target classification accuracy on HHAR computed with domain adaptation method CoDATS.

| Scenario | Bayesian Model Averaging | | | | Super Learner | | | |
|---|---|---|---|---|---|---|---|---|
| | KuLSIF | Exp | LR | SQ | KuLSIF | Exp | LR | SQ |
| $0 \to 6$ | 0.606(±0.031) | 0.663(±0.042) | 0.657(±0.026) | 0.621(±0.015) | 0.610(±0.032) | 0.673(±0.032) | 0.658(±0.022) | 0.628(±0.013) |
| $1 \to 6$ | 0.938(±0.002) | 0.905(±0.037) | 0.906(±0.009) | 0.876(±0.006) | 0.944(±0.001) | 0.908(±0.027) | 0.910(±0.005) | 0.880(±0.003) |
| $2 \to 7$ | 0.326(±0.279) | 0.527(±0.024) | 0.533(±0.044) | 0.500(±0.033) | 0.332(±0.278) | 0.539(±0.015) | 0.536(±0.040) | 0.500(±0.030) |
| $3 \to 8$ | 0.963(±0.001) | 0.944(±0.007) | 0.923(±0.013) | 0.885(±0.003) | 0.964(±0.002) | 0.946(±0.004) | 0.926(±0.009) | 0.890(±0.000) |
| $4 \to 5$ | 0.653(±0.562) | 0.972(±0.004) | 0.957(±0.009) | 0.923(±0.006) | 0.653(±0.561) | 0.978(±0.006) | 0.955(±0.004) | 0.925(±0.004) |
| Avg. | 0.697(±0.175) | 0.802(±0.023) | 0.795(±0.020) | 0.761(±0.013) | 0.701(±0.175) | 0.809(±0.017) | 0.797(±0.016) | 0.765(±0.010) |

Table 66: Mean and standard deviation (after ±) of target classification accuracy on HHAR computed with domain adaptation method DANN.

| Scenario | Bayesian Model Averaging | | | | Super Learner | | | |
|---|---|---|---|---|---|---|---|---|
| | KuLSIF | Exp | LR | SQ | KuLSIF | Exp | LR | SQ |
| $0 \to 6$ | 0.624(±0.005) | 0.530(±0.280) | 0.666(±0.013) | 0.638(±0.002) | 0.624(±0.004) | 0.539(±0.270) | 0.666(±0.009) | 0.636(±0.000) |
| $1 \to 6$ | 0.932(±0.004) | 0.920(±0.007) | 0.908(±0.005) | 0.866(±0.010) | 0.941(±0.006) | 0.925(±0.004) | 0.910(±0.000) | 0.879(±0.008) |
| $2 \to 7$ | 0.549(±0.058) | 0.574(±0.061) | 0.552(±0.064) | 0.510(±0.054) | 0.554(±0.056) | 0.579(±0.050) | 0.556(±0.060) | 0.531(±0.050) |
| $3 \to 8$ | 0.876(±0.072) | 0.874(±0.058) | 0.862(±0.055) | 0.831(±0.044) | 0.870(±0.071) | 0.890(±0.048) | 0.866(±0.051) | 0.835(±0.042) |
| $4 \to 5$ | 0.981(±0.005) | 0.722(±0.423) | 0.954(±0.005) | 0.916(±0.010) | 0.987(±0.006) | 0.741(±0.412) | 0.958(±0.000) | 0.933(±0.006) |
| Avg. | 0.793(±0.029) | 0.724(±0.166) | 0.788(±0.028) | 0.752(±0.024) | 0.795(±0.028) | 0.735(±0.157) | 0.791(±0.024) | 0.763(±0.021) |

Table 67: Mean and standard deviation (after ±) of target classification accuracy on HHAR computed with domain adaptation method CDAN.

| Scenario | Bayesian Model Averaging | | | | Super Learner | | | |
|---|---|---|---|---|---|---|---|---|
| | KuLSIF | Exp | LR | SQ | KuLSIF | Exp | LR | SQ |
| $0 \to 6$ | 0.721(±0.066) | 0.371(±0.288) | 0.705(±0.042) | 0.667(±0.031) | 0.727(±0.065) | 0.383(±0.278) | 0.709(±0.039) | 0.679(±0.030) |
| $1 \to 6$ | 0.929(±0.004) | 0.427(±0.427) | 0.893(±0.022) | 0.863(±0.011) | 0.931(±0.005) | 0.433(±0.418) | 0.895(±0.018) | 0.868(±0.008) |
| $2 \to 7$ | 0.498(±0.004) | 0.515(±0.001) | 0.495(±0.000) | 0.462(±0.015) | 0.502(±0.005) | 0.515(±0.009) | 0.497(±0.004) | 0.461(±0.013) |
| $3 \to 8$ | 0.972(±0.000) | 0.943(±0.005) | 0.927(±0.004) | 0.891(±0.011) | 0.972(±0.000) | 0.952(±0.006) | 0.931(±0.001) | 0.901(±0.009) |
| $4 \to 5$ | 0.656(±0.561) | 0.470(±0.416) | 0.939(±0.003) | 0.909(±0.011) | 0.662(±0.561) | 0.492(±0.406) | 0.940(±0.000) | 0.907(±0.008) |
| Avg. | 0.755(±0.127) | 0.545(±0.227) | 0.792(±0.014) | 0.758(±0.016) | 0.759(±0.127) | 0.555(±0.224) | 0.795(±0.013) | 0.763(±0.013) |

Table 68: Mean and standard deviation (after ±) of target classification accuracy on HHAR computed with domain adaptation method DSAN.

| Scenario | Bayesian Model Averaging | | | | Super Learner | | | |
|---|---|---|---|---|---|---|---|---|
| | KuLSIF | Exp | LR | SQ | KuLSIF | Exp | LR | SQ |
| $0 \to 6$ | 0.692(±0.068) | 0.707(±0.048) | 0.709(±0.041) | 0.667(±0.031) | 0.698(±0.067) | 0.723(±0.039) | 0.712(±0.038) | 0.679(±0.027) |
| $1 \to 6$ | 0.936(±0.003) | 0.902(±0.029) | 0.904(±0.009) | 0.864(±0.006) | 0.934(±0.005) | 0.908(±0.018) | 0.902(±0.004) | 0.870(±0.002) |
| $2 \to 7$ | 0.196(±0.333) | 0.599(±0.091) | 0.545(±0.040) | 0.506(±0.029) | 0.202(±0.330) | 0.617(±0.080) | 0.543(±0.037) | 0.513(±0.026) |
| $3 \to 8$ | 0.847(±0.002) | 0.855(±0.020) | 0.842(±0.011) | 0.812(±0.005) | 0.852(±0.002) | 0.868(±0.011) | 0.846(±0.006) | 0.811(±0.002) |
| $4 \to 5$ | 0.986(±0.001) | 0.967(±0.007) | 0.951(±0.007) | 0.919(±0.007) | 0.983(±0.000) | 0.968(±0.002) | 0.955(±0.004) | 0.921(±0.005) |
| Avg. | 0.732(±0.081) | 0.806(±0.039) | 0.790(±0.022) | 0.753(±0.016) | 0.734(±0.081) | 0.817(±0.030) | 0.791(±0.018) | 0.759(±0.013) |

Table 69: Mean and standard deviation (after ±) of target classification accuracy on HHAR computed with domain adaptation method DIRT.

| Scenario | Bayesian Model Averaging | | | | Super Learner | | | |
|---|---|---|---|---|---|---|---|---|
| | KuLSIF | Exp | LR | SQ | KuLSIF | Exp | LR | SQ |
| $0 \to 6$ | 0.720(±0.004) | 0.733(±0.008) | 0.705(±0.019) | 0.679(±0.008) | 0.720(±0.004) | 0.745(±0.000) | 0.703(±0.015) | 0.671(±0.005) |
| $1 \to 6$ | 0.859(±0.003) | 0.854(±0.025) | 0.836(±0.024) | 0.800(±0.013) | 0.862(±0.002) | 0.863(±0.015) | 0.838(±0.020) | 0.804(±0.010) |
| $2 \to 7$ | 0.489(±0.010) | 0.510(±0.010) | 0.502(±0.023) | 0.469(±0.012) | 0.495(±0.009) | 0.524(±0.001) | 0.504(±0.018) | 0.478(±0.008) |
| $3 \to 8$ | 0.809(±0.002) | 0.821(±0.004) | 0.807(±0.004) | 0.770(±0.011) | 0.818(±0.002) | 0.826(±0.007) | 0.811(±0.000) | 0.779(±0.008) |
| $4 \to 5$ | 0.883(±0.001) | 0.889(±0.013) | 0.878(±0.015) | 0.848(±0.005) | 0.892(±0.000) | 0.896(±0.003) | 0.882(±0.012) | 0.854(±0.002) |
| Avg. | 0.752(±0.004) | 0.762(±0.012) | 0.746(±0.017) | 0.713(±0.010) | 0.757(±0.003) | 0.771(±0.005) | 0.748(±0.013) | 0.717(±0.007) |

Table 70: Mean and standard deviation (after ±) of target classification accuracy on HHAR computed with domain adaptation method AdvSKM.

| Scenario | Bayesian Model Averaging | | | | Super Learner | | | |
|---|---|---|---|---|---|---|---|---|
| | KuLSIF | Exp | LR | SQ | KuLSIF | Exp | LR | SQ |
| $0 \to 6$ | 0.733(±0.005) | 0.738(±0.023) | 0.725(±0.022) | 0.695(±0.012) | 0.740(±0.004) | 0.744(±0.014) | 0.729(±0.019) | 0.693(±0.009) |
| $1 \to 6$ | 0.878(±0.006) | 0.872(±0.018) | 0.853(±0.023) | 0.823(±0.012) | 0.879(±0.007) | 0.884(±0.008) | 0.851(±0.018) | 0.821(±0.008) |
| $2 \to 7$ | 0.454(±0.012) | 0.477(±0.029) | 0.474(±0.024) | 0.444(±0.012) | 0.451(±0.011) | 0.492(±0.018) | 0.477(±0.019) | 0.442(±0.009) |
| $3 \to 8$ | 0.817(±0.001) | 0.841(±0.007) | 0.822(±0.003) | 0.792(±0.012) | 0.824(±0.000) | 0.847(±0.002) | 0.820(±0.001) | 0.783(±0.009) |
| $4 \to 5$ | 0.910(±0.000) | 0.467(±0.398) | 0.899(±0.030) | 0.862(±0.019) | 0.911(±0.002) | 0.481(±0.388) | 0.897(±0.027) | 0.869(±0.016) |
| Avg. | 0.759(±0.005) | 0.679(±0.095) | 0.754(±0.020) | 0.723(±0.013) | 0.761(±0.005) | 0.690(±0.086) | 0.755(±0.017) | 0.721(±0.010) |

Table 71: Mean and standard deviation (after ±) of target classification accuracy on HHAR computed with domain adaptation method HoMM.

| Scenario | Bayesian Model Averaging | | | | Super Learner | | | |
|---|---|---|---|---|---|---|---|---|
| | KuLSIF | Exp | LR | SQ | KuLSIF | Exp | LR | SQ |
| $0 \to 6$ | 0.699($\pm$0.009) | 0.369($\pm$0.285) | 0.690($\pm$0.010) | 0.648($\pm$0.005) | 0.700($\pm$0.007) | 0.369($\pm$0.276) | 0.694($\pm$0.006) | 0.660($\pm$0.002) |
| $1 \to 6$ | 0.882($\pm$0.014) | 0.654($\pm$0.412) | 0.873($\pm$0.013) | 0.842($\pm$0.002) | 0.881($\pm$0.011) | 0.664($\pm$0.401) | 0.877($\pm$0.008) | 0.838($\pm$0.001) |
| $2 \to 7$ | 0.441($\pm$0.037) | 0.460($\pm$0.035) | 0.439($\pm$0.035) | 0.410($\pm$0.025) | 0.441($\pm$0.036) | 0.470($\pm$0.025) | 0.439($\pm$0.032) | 0.404($\pm$0.023) |
| $3 \to 8$ | 0.817($\pm$0.004) | 0.633($\pm$0.341) | 0.823($\pm$0.005) | 0.783($\pm$0.010) | 0.826($\pm$0.003) | 0.644($\pm$0.331) | 0.827($\pm$0.001) | 0.797($\pm$0.006) |
| $4 \to 5$ | 0.907($\pm$0.005) | 0.237($\pm$0.001) | 0.895($\pm$0.025) | 0.865($\pm$0.013) | 0.911($\pm$0.003) | 0.244($\pm$0.011) | 0.893($\pm$0.021) | 0.869($\pm$0.011) |
| Avg. | 0.749($\pm$0.014) | 0.471($\pm$0.215) | 0.744($\pm$0.018) | 0.710($\pm$0.011) | 0.752($\pm$0.012) | 0.478($\pm$0.209) | 0.746($\pm$0.014) | 0.713($\pm$0.009) |

Table 72: Mean and standard deviation (after $\pm$) of target classification accuracy on HHAR computed with domain adaptation method DDC.

| Scenario | Bayesian Model Averaging | | | | Super Learner | | | |
|---|---|---|---|---|---|---|---|---|
| | KuLSIF | Exp | LR | SQ | KuLSIF | Exp | LR | SQ |
| $0 \to 6$ | 0.728($\pm$0.015) | 0.710($\pm$0.031) | 0.717($\pm$0.024) | 0.687($\pm$0.013) | 0.736($\pm$0.014) | 0.725($\pm$0.020) | 0.719($\pm$0.019) | 0.688($\pm$0.010) |
| $1 \to 6$ | 0.888($\pm$0.005) | 0.867($\pm$0.037) | 0.861($\pm$0.014) | 0.827($\pm$0.002) | 0.889($\pm$0.003) | 0.868($\pm$0.028) | 0.859($\pm$0.009) | 0.836($\pm$0.001) |
| $2 \to 7$ | 0.459($\pm$0.007) | 0.486($\pm$0.013) | 0.480($\pm$0.019) | 0.450($\pm$0.008) | 0.457($\pm$0.007) | 0.497($\pm$0.003) | 0.478($\pm$0.015) | 0.446($\pm$0.006) |
| $3 \to 8$ | 0.812($\pm$0.004) | 0.840($\pm$0.002) | 0.811($\pm$0.000) | 0.778($\pm$0.015) | 0.820($\pm$0.004) | 0.846($\pm$0.009) | 0.809($\pm$0.004) | 0.788($\pm$0.012) |
| $4 \to 5$ | 0.620($\pm$0.531) | 0.935($\pm$0.018) | 0.923($\pm$0.015) | 0.893($\pm$0.004) | 0.625($\pm$0.530) | 0.957($\pm$0.009) | 0.927($\pm$0.011) | 0.893($\pm$0.001) |
| Avg. | 0.701($\pm$0.113) | 0.768($\pm$0.020) | 0.758($\pm$0.014) | 0.727($\pm$0.008) | 0.705($\pm$0.112) | 0.779($\pm$0.014) | 0.758($\pm$0.012) | 0.730($\pm$0.006) |

Table 73: Mean and standard deviation (after $\pm$) of target classification accuracy on HHAR computed with domain adaptation method DeepCoral.

| Scenario | Bayesian Model Averaging | | | | Super Learner | | | |
|---|---|---|---|---|---|---|---|---|
| | KuLSIF | Exp | LR | SQ | KuLSIF | Exp | LR | SQ |
| $0 \to 6$ | 0.693($\pm$0.008) | 0.729($\pm$0.011) | 0.693($\pm$0.010) | 0.663($\pm$0.006) | 0.695($\pm$0.008) | 0.743($\pm$0.001) | 0.697($\pm$0.005) | 0.663($\pm$0.003) |
| $1 \to 6$ | 0.909($\pm$0.002) | 0.897($\pm$0.016) | 0.896($\pm$0.006) | 0.859($\pm$0.008) | 0.903($\pm$0.001) | 0.913($\pm$0.007) | 0.897($\pm$0.003) | 0.866($\pm$0.005) |
| $2 \to 7$ | 0.320($\pm$0.274) | 0.512($\pm$0.027) | 0.513($\pm$0.040) | 0.482($\pm$0.028) | 0.322($\pm$0.272) | 0.531($\pm$0.017) | 0.511($\pm$0.035) | 0.489($\pm$0.025) |
| $3 \to 8$ | 0.810($\pm$0.007) | 0.831($\pm$0.008) | 0.813($\pm$0.010) | 0.775($\pm$0.005) | 0.815($\pm$0.007) | 0.837($\pm$0.002) | 0.817($\pm$0.006) | 0.788($\pm$0.002) |
| $4 \to 5$ | 0.624($\pm$0.537) | 0.956($\pm$0.010) | 0.934($\pm$0.012) | 0.904($\pm$0.003) | 0.633($\pm$0.536) | 0.963($\pm$0.000) | 0.938($\pm$0.008) | 0.902($\pm$0.001) |
| Avg. | 0.671($\pm$0.166) | 0.785($\pm$0.014) | 0.770($\pm$0.016) | 0.736($\pm$0.010) | 0.674($\pm$0.165) | 0.797($\pm$0.005) | 0.772($\pm$0.011) | 0.742($\pm$0.007) |

Table 74: Mean and standard deviation (after $\pm$) of target classification accuracy on HHAR computed with domain adaptation method CMD.

