# OpenReview forum: "Minimax-Optimal Aggregation for Density Ratio Estimation"
_ICLR.cc/2026/Conference — ICLR 2026 Poster_

### Official Review · Reviewer_aJPm · 2025-10-27

**Soundness:** 3
**Presentation:** 3
**Contribution:** 3
**Rating:** 8
**Confidence:** 3

**Summary:**

This paper provides an aggregation method for solving density ratio estimation (DRE). Through this, it tackles both the issue of method selection as well as hyperparameter tuning by instead training multiple models with multiple hyperparameter setups and aggregating them instead of just selecting one. The authors provide minimax-optimal error bounds without smoothness assumptions for RKHS and convex settings. Thorough empirical validation of the method against cross-validation baselines is done on models beyond just RKHS settings including neural networks.

**Strengths:**

* Proposes a novel and effective strategy for dealing with model and hyperparameter aggregation for density ratio estimation.
* The proposed closed form solution beats every candidate model instead of just selecting a single one.
* Theoretical minimax guarantees for RKHS and convex settings.
* Empirical results also show significant improvements over existing selection baselines in different data settings and using models like neural networks.

**Weaknesses:**

* It is unclear if the theory covers all the empirical settings like the case where models are neural networks, though this is a tough ask.
* The method requires empirical computation of a large Gram matrix of size ~ number of models X hyperparameter settings. This could be numerically unstable for various real world applications.

**Questions:**

As per the weakness section,

1. Does the theory extend to all your empirical evaluations?
2. Could the authors comment briefly on the numerical stability of computing the Gram matrix in real applications, and also comment on the costs of computing Hessians in Alg 1?
3. Are there guarantees for the aggregated DRE to always be non-negative?

---

> ### Author Response · Authors · 2025-11-20
>
> Thank you for reviewing our manuscript and for your valuable feedback. Please find our detailed responses below.
>
> **On the settings covered by our theory**:  Thank you for raising this question. Our theoretical guarantees apply to the broad class of DRE models optimized via the Bregman objective in equation (1). When using neural networks, our method serves as a heuristic as is typical for neural-network–based estimators. Results are shown in Ablation 1 (Table 1). Similarly, we use this heuristic to tune the kernel bandwidth parameter in Kernel Mean Matching where CV cannot be used (Ablation 2; Table 1). In both cases, our approach consistently improves experimental results upon baseline DRE methods.
> All other experiments employ DRE models directly based on the Bregman objective, and therefore fall fully within the scope of our theoretical analysis. We now clarify this distinction explicitly in the manuscript as can be seen in the revised pdf.
>
> **On numerical stability, and Hessian computation**: Standard tools are available to mitigate numerical instabilities arising from ill-conditioned matrices. One effective approach is an eigenvalue regularization based on a threshold value for small eigenvalues when computing the pseudoinverse via eigendecomposition of the Gram matrix (Strang, 1980). Using this procedure, we did not encounter numerical issues in any of our real-world experiments.
> Regarding computational cost: in Algorithm 1, the Hessians required by our aggregation method have closed-form expressions, making their evaluation computationally negligible. We provide a complete worked example for the logistic-regression loss in Appendix B.
>
> **On the non-negativeness of DRE**: The non-negativeness of the aggregated estimator depends on the properties of the individual base DRE methods. Some estimators e.g., logistic-regression–based DREs are inherently non-negative due to their exponential output parameterization. Others, such as KuLSIF, can in principle yield negative estimates, although constrained variants exist that enforce non-negativeness. In practice, methods like KuLSIF are effectively non-negative within their statistical error, and our aggregation scheme further improves their convergence rates to achieve minimax-optimal performance.
>
> We thank you again for your time and effort used to review our work. If you are satisfied with our answers, we kindly ask you to take this into account in your final decision.
>
> G. Strang. Linear algebra and its applications. Orlando, FL, Academic Press, Inc., 1980

---

### Official Review · Reviewer_HR3j · 2025-10-28

**Soundness:** 3
**Presentation:** 2
**Contribution:** 2
**Rating:** 8
**Confidence:** 3

**Summary:**

This paper proposes a method for training multiple models under different hyperparameter configurations and optimally aggregating their outputs. The proposed approach demonstrates the ability to achieve minimax-optimal convergence rates without requiring any prior knowledge about the smoothness of the density ratios. Experimental results also demons	trate that the proposed method outperforms cross-validation, Bayesian model averaging, and Super Learner.

**Strengths:**

* The method achieves minimax-optimal convergence rates under reasonable assumptions.
* The proposed algorithm is theoretically guaranteed and practically computable.  The proposed algorithm is a simple least-squares method that is intuitively easy to understand.
* Through various density ratio estimation methods, they demonstrate the superiority of the proposed approach over the baseline methods.

**Weaknesses:**

* The following descriptions of the experiments are insufficient.
  * The tuning method for the hyperparameter, lambda
  * The details of BMA:  It is better to write which algorithm in Fragoso et al. (2018) is used.

* Since cross-validation requires only inference from a single selected density ratio estimator during inference, it has the computational advantage of requiring less processing time than the proposed method. Therefore, I consider the superiority of the proposed method to be limited to only certain applications.

* Using $f$ both for DRE models in equation (1) and for DA models in equation (13).   The use of the symbol may confuse readers.

**Questions:**

* How did you choose the regularization parameter value, lambda?
* Could you discuss how the lambda selection affected the computational complexity in Fig. 2? In my understanding, the trained models and empirical Hessians can be shared across different lambda values, so there is little impact on overall training time. When you tried 10 different lambda values, $\{10^{−6},10^{−5},...,10^{4}\}$ in the experiments, how much did the whole training time increase compared with the single lambda value?

---

> ### Author Response · Authors · 2025-11-20
>
> Thank you for reviewing our manuscript, your detailed feedback, and your valuable suggestions. Please find below our detailed answers.
>
> **On the description of experiments**: Thank you for your helpful comments. Following your suggestions, we have added the requested experimental details, which can also be seen in the revised version of our pdf.
>  - Tuning method for lambda: For model selection, we conduct 10-fold cross-validation which performs grid-search to tune the hyperparameter lambda from a range $10^{−6} , 10^{−5} , \ldots , 10^4$. For model aggregation, we do not select a single lambda parameter; instead, we aggregate models corresponding to all lambda values in the grid.
>  - Details on BMA: We now clarify in the manuscript that we use the BMA procedure described in Fragoso et al. (2018), which builds upon Yeung et al. (2005). As in standard BMA, individual models are weighted by their posterior probabilities.
>
> **On inference compute**: Our method consistently outperforms cross-validation in terms of approximation error. In most DRE applications, inference-time computation is not the main bottleneck; training dominates the computational cost. When an inference-time tradeoff between accuracy and compute is needed, one can simply use a subset of the previously trained ensemble. This is consistent with our observations in Figure 3 (Left) of the Appendix, where our method assigns higher weights to more accurate models, making such pruning straightforward.
>
> **On notation**: Thank you for pointing out the confusion in equation (13). We now use a distinct symbol for the domain-adaptation models, as reflected in the revised manuscript.
>
> **On the effect of lambda selection on the computational complexity**: For our aggregation method, we do not select a single lambda; all trained models (each corresponding to a different lambda value) are aggregated. Thus, while the empirical Hessians can indeed be reused across lambda values, as you correctly noted, the overall training time remains dominated by the cost of training the individual models. The Hessians, like the aggregation weights, can be computed analytically, adding negligible computational overhead. Consequently, exploring 10 different lambda values increases training time by approximately a factor of 10 compared with training a single model. In contrast, the $n$-fold cross-validation baseline must train models for each of the $n$ folds and each lambda value, increasing training time by an additional factor of $n$ relative to our aggregation approach, as can be seen in Figure 2 (Left).
>
> We thank you again for your time and effort used to review our work. If you are satisfied with our answers, we kindly ask you to take this into account in your final decision.
>
> Yeung et al.. Bayesian model averaging: development of an improved multi-class, gene selection and classification tool for microarray data. Bioinformatics, 2005

---

> > ### Comment · Reviewer_HR3j · 2025-11-21
> > **Thank you for your response.**
> >
> > > Tuning method for lambda: For model selection, we conduct 10-fold cross-validation which performs grid-search to tune the hyperparameter lambda from a range $10^{−6} , 10^{−5} , \ldots , 10^4$. For model aggregation, we do not select a single lambda parameter; instead, we aggregate models corresponding to all lambda values in the grid.
> >
> > Could you provide a bit more detailed explanation?
> > For model aggregation, Algorithm 1 states $\hat{\beta}(x) = g\left( \sum_k \alpha_{k} f_k(x) \right)$ , but does this actually mean $\hat{\beta}(x)=g\left( \sum_{\lambda} \sum_k \hat{\alpha}_{k,\lambda} f_k(x) \right)$  ?
> > Also, regarding the model selection, I understood you're selecting one from $\{f_k\}_k^K$ using the cross-validation - why does $\lambda$ come into play here?
> > Please forgive me if I've overlooked them, but these are important details for reproducibility that should be included in the paper.

---

> > > ### Author Response · Authors · 2025-11-21
> > >
> > > Thank you  for the opportunity to further elaborate on our approach. To address your comments, we provide a more detailed explanation of our algorithm, and the model selection procedure.
> > >
> > > Algorithm 1 presents the general formulation of our aggregation approach. In this framework, each $f_k$​ denotes a model trained under a distinct hyperparameter choice. In many of our experiments, these configurations correspond specifically to different values of the regularization parameter $\lambda$ used in the DRE methods, as can be seen, e.g., in equation 8, and below where $\widehat{\beta}$ is defined. In that setting, Algorithm 1 instantiates as
> > > $$
> > > \widehat{\beta}(x) = g\left(\sum_{k=1}^K \widehat{\alpha}_k f^{\lambda_k}(x)\right)
> > > $$
> > > where the $\lambda_k$​ values range over $\{10^{-6}, 10^{-5}, \ldots, 10^4\}$.
> > >
> > > We use the generic notation $f_k$ in Algorithm 1 deliberately: our aggregation procedure is not restricted to varying the regularization parameter, but is compatible with any choice of hyperparameters. Ablations 1 and 2 demonstrate this generality by applying the method to model families generated from different types of hyperparameter variations.
> > >
> > > As you correctly observed, for model-selection, one element from $\{f_k\}_{k=1}^K$ is selected by using cross-validation, where each $f_k$​ is indeed a model trained with a distinct $\lambda_k$​ in the experiments focused on regularization.
> > >
> > > Thank you again for your discussion, we will include this explanation in the final version directly below the description of Algorithm 1.

---

> ### Comment · Reviewer_HR3j · 2025-11-22
> **damping term of matrix inverse**
>
> Since you calculate the inverse of G, which would usually utilize a damping term for numerical stability, I mistakenly believed you were using lambda when computing both G and r. Since you're actually computing the inverse without the damping term, does this imply you're assuming G is always positive definite? Would not this be a problem in practice? Additionally, the notation "as in equation 11" in Algorithm 1 also contributed to my misunderstanding.
> It would be helpful to clarify why Algorithm 1 ignores the lambda introduced in Section 6.1.

---

> > ### Author Response · Authors · 2025-11-22
> >
> > Thank you for continuing this valuable and engaging discussion.
> >
> > Regarding the Gram matrix, it is by construction positive semidefinite. In practice, we can safely assume that it is invertible and thus positive definite because otherwise some models would be effectively redundant i.e., can be excluded from consideration.
> > Standard techniques exist to address potential numerical instabilities arising from ill-conditioned matrices. One effective approach is an eigenvalue regularization based on a threshold value for small eigenvalues when computing the pseudoinverse via eigendecomposition of the Gram matrix (Strang, 1980). Using this approach, we did not encounter numerical issues in any of our experiments. We will include this discussion in the final version of the paper.
> >
> > Concerning the phrase “as in equation 11” in Algorithm 1: this refers specifically to the computation of the second derivative $\nabla^2 \ell (y, f(x))$. We will clarify this reference as well as the precise role of lambda in the final version of the paper.
> >
> > G. Strang. Linear algebra and its applications. Orlando, FL, Academic Press, Inc., 1980

---

> > > ### Comment · Reviewer_HR3j · 2025-11-25
> > > **Thank you for the reply**
> > >
> > > > Regarding the Gram matrix, it is by construction positive semidefinite.
> > >
> > > For non-linear models like DeepLR, its Hessian can be indefinite, but assuming the model converges to a local minimum, I understand it to be positive semi-definite, so the Gram matrix is positive semi-definite.

---

> > > > ### Author Response · Authors · 2025-11-25
> > > >
> > > > Thank you for continuing our discussion.
> > > >
> > > > DeepLR is likewise based on the logistic-regression loss $\ell (y, f(x))$ (Example 2, 4.). As with the other losses considered, this loss is convex under Assumption 4. Its Hessian is taken with respect to $f$, more precisely $\nabla_f^2 \ell (y, f(x))$, and is therefore positive semi-definite. We provide a complete worked example for the logistic-regression loss in Appendix B. Because the Gram matrix is defined through these Hessian-weighted inner products, it is consequently positive semi-definite as well.
> > > >
> > > > We will expand this discussion further in the final version of the paper.

---

> > > > > ### Comment · Reviewer_HR3j · 2025-11-25
> > > > > **Thank you**
> > > > >
> > > > > I forgot it was a functional derivative. Your explanation helped me understand it clearly.

---

### Official Review · Reviewer_96xS · 2025-10-31

**Soundness:** 3
**Presentation:** 3
**Contribution:** 3
**Rating:** 6
**Confidence:** 3

**Summary:**

The paper find suitable weights that is used to aggregate results from multiple models trained with different hyperparameter to predict density ratio estimation(DRE). The weights are obtained by minimizing the upper bound on the Bregman divergence.

**Strengths:**

1. There is a strong mathemtical proof
2. There a many benchmarks across various task and domains

**Weaknesses:**

1. The paper could provide analysis on the number of models trained and results obtained through this method?
2. The paper could also show analysis of the distribution of the hyper-parameters used.

**Questions:**

1. Does the variance of the models output affect the bounds?
2. How would noise within the data affect the results analytical?

---

> ### Author Response · Authors · 2025-11-20
>
> Thank you for reviewing our manuscript, your constructive feedback, and valuable suggestions which have helped improve the quality of our work. Please find our detailed responses below.
>
> **On analysis of the number of trained models**: Thank you for bringing this up for discussion.
> In Ablation 4 (Figure 2, Right), we investigate how performance changes as we increase the number of trained models. Our method remains robust even as additional, potentially suboptimal, models are included. In contrast, established ensembling and model-averaging techniques such as Raza & Samothrakis (2019), Bayesian Model Averaging (Fragoso et al., 2018), and Super Learner (Wu & Benkeser, 2024) exhibit instabilities and substantial performance degradation under the same conditions. This difference underscores a key strength of our approach: by explicitly optimizing aggregation weights, it achieves minimax-optimal convergence rates and handles increasing model sets more gracefully. We now highlight this more clearly in our manuscript which can be seen in the revised pdf.
>
> **On the distribution of the hyper-parameters**: We are not entirely sure what you mean by “the distribution of the hyper-parameters used”.  In Figure 3 (Left) in the Appendix we examine how our method assigns weights to models with different hyper-parameters and can see that more accurate models get assigned larger weights. For the model selection baseline which uses cross-validation, always the single best hyper-parameter is selected on a validation set. If this does not fully address your concern, we would be grateful for further clarification.
>
> Thank you for raising insightful questions about the potential effects of model variance and data noise. Our detailed responses are provided below, and we have incorporated the corresponding discussion into the revised manuscript, as reflected in the updated pdf.
> **On model output variance/noise**: It is handled by the least squares construction of the weights $\boldsymbol{\widehat{\alpha}} = \mathbf{\widehat{G}}^{-1} \mathbf{\widehat{r}}$: Consider a noisy model $f^{\lambda_k}$. Its self-correlation $\widehat{G}_{kk}$, i.e., the $k$-th diagonal element of the empirical Gram matrix, becomes large, while its alignment with the target function, $\widehat{r}\_k = \langle f\_k, f\_{\mathcal{H}} \rangle$, remains small because noise does not correlate with the true signal. This is desirable because the optimal weights satisfy $\boldsymbol{\widehat{\alpha}} = \mathbf{\widehat{G}}^{-1} \mathbf{\widehat{r}}$, and, in particular, the contribution of model $k$ is governed by the ratio $\widehat{r}\_k / \widehat{G}\_{kk}$ (in the diagonal approximation, intuitively $\widehat{\alpha}\_k \approx \widehat{r}\_k / \widehat{G}\_{kk}$). Hence, a large denominator and small numerator force $\widehat{\alpha}_k$ to be small, ensuring that noisy or unstable models are down-weighted in the aggregation.
>
> **On data noise**: Data noise appears in two places:
> First, in Lemma 5, the deviation between the population Hessian $\mathbf{H}$ and the empirical Hessian
> $\widehat{\mathbf{H}}$ is controlled exactly as in classical uniform per-sample operator-trace bounds
> (e.g., matrix Bernstein), through the constant $B_2^\ast$, see Eq.(20)-(22).
> Second, in Lemma 4, the deviation measure $\mathbf{t}(f_{\mathcal H} - \widehat{f}^{\lambda_0})$ is
> controlled in the standard way used in self-concordant and kernel-learning analyses, via the
> bounded kernel norm $R$, the bounded derivative $|\nabla \ell_z(0)|$, and the Hessian trace
> bounds $B_1^\ast$ and $B_2^\ast$.
> Summarizing, these quantities affect the concentration bounds and therefore the multiplicative constant $C$ in the final bound (12). Larger values increase the deviation terms controlling the empirical–population differences and enlarge the sample size required for the guarantees to hold. Importantly, however, they do not influence the non-asymptotic rate in $M+N$, which remains unchanged.
>
> We thank you again for your time and effort used to review our work. If you are satisfied with our answers and extensions, we kindly ask you to take this into account in your final decision.

---

> > ### Author Response · Authors · 2025-11-26
> >
> > Thank you once again for your time and thoughtful review of our work. We believe we have fully addressed the concerns you raised. If you have no further questions, please take this into account in your final scoring.

---

### Official Review · Reviewer_kUnW · 2025-10-31

**Soundness:** 3
**Presentation:** 3
**Contribution:** 3
**Rating:** 8
**Confidence:** 2

**Summary:**

The paper propose a new algorithm for Density Ratio Estimation (DRE) that aggregates multiple models with different hyper parameter settings. The empirical experiments focus on domain adaptation, as well as comparisons with cross validation for several different loss objectives. The authors prove that the aggregation achieves minimax-optimal error convergence under very mild conditions on the density ratio.

**Strengths:**

Overall I think this is a good paper. It's written well and proposes a novel method for DRE estimation that is more computationally efficient than simple cross validation, and improves performance over several other baselines, on several relevant tasks.
The method is supported by a provably minimax-optimal error convergence.
Several ablations are done to show the robustness of the method.

**Weaknesses:**

Slightly more detail about the dataset for the top part of Table 1 would have been useful. What does "c3,d1.70" stand for instance, and how many samples were used in the dataset "c3,d1.70".

Minor:
I think the left part of figure two can be improved. If my understanding is correct, the number of models needed is reduced by a factor of the number of k-folds. By plotting it out instead of just stating the mechanical relationship, people may get confused and look for some patterns in the figure.

**Questions:**

Can the method also be applied when instead of having different hyperparameters in the same objective, we have multiple models from different objectives (KuLSIF, Exp, etc.)?

---

> ### Author Response · Authors · 2025-11-20
>
> Thank you for reviewing our manuscript, your constructive feedback, and your valuable suggestions. Please find below our answers.
>
> **On dataset details for known density ratios**: Thank you for pointing this out. The notation “c3, d1.70” serves as an identifier for one of the 10 Gaussian–mixture–based datasets used in our evaluation. Here, “c3” indicates that the denominator distribution of the density ratio has 3 mixture components, and “d1.70” denotes the minimum separation (1.70) between any pair of modes across both the numerator and denominator distributions. The remaining datasets follow the same naming convention, and each contains 2,500 samples. We clarified these details as can be seen in our revised pdf.
>
> **On Figure 2**: Thank you for the suggestion. You are correct, the number of models that must be trained is reduced by a factor equal to the number of folds used in cross-validation. We have expanded the explanation in the revised manuscript to make this clearer.
>
> **On extending aggregation**: Thank you for this insightful question. This would be an interesting heuristic beyond minimax error guarantees in Theorem 1. Aggregation would have to be done using a norm different from (10) which we consider an interesting avenue for future work.
>
> We thank you again for your time and effort used to review our work. If you are satisfied with our answers, we kindly ask you to take this into account in your final decision.

---

### Author Response · Authors · 2025-11-20
**General Response**

We sincerely thank all reviewers for their time, effort, and constructive feedback, as well as for their positive assessment of our work. We are pleased that the theoretical minimax-optimal result was well received, and that the reviewers recognized the strength of our experimental evaluation, which demonstrates consistent improvements over baseline methods across diverse tasks, benchmarks, and domains. Below, we provide detailed, point-by-point responses to each individual review. We have also uploaded an updated version of the manuscript that incorporates the reviewers’ suggestions.

---

### Meta-Review · Area_Chair_oXj3 · 2026-01-08

**Summary:**

This paper proposes a way to perform density ratio estimation (DRE) by optimally aggregating multiple models. These models are trained separately and the method provides a simple yet practical way to aggregate. While the least squares perspective is simple, the method comes with minimax results and reviewers agree these provide a worthwhile theoretical element. Importantly, the validation is fairly thorough, including domain adaptation tasks and neural‑networks, and, is compared with reasonable baselines. Overall, it is convincing that this method may be a good alternative to classic choices such as CV or BMA.

**Reviewer Concerns:**

There were some concerns that were raised including implementation details, numerical points, etc, that were mostly answered adequately during reviewer response period. The authors provided clarification on data, tuning, and various computations involved and provided clarification on other questions related to the data and scope of guarantees.

**Reviewer Scores:**

The scores look consistent with the content of the reviews

---

### Decision · Program_Chairs · 2026-01-26

Accept (Poster)